# A Closed-Form Persistence-Landmark Pipeline for Certified Point-Cloud and Graph Classification

## Abstract

We introduce **PLACE** (**P**ersistence-**L**andmark **A**nalytic **C**lassification **E**ngine), a closed-form pipeline for classifying point clouds and graphs through their persistent-homology signatures. Three quantitative guarantees—a margin-based excess-risk rate, a closed-form descriptor-selection rule, and a per-prediction certificate—are derived from training labels alone, with no learned weights or held-out calibration. The embedding sums Mitra–Virk single-point coordinate functions over a sparse landmark grid; the closed-form weight rule $w_k^2 \propto (d_{k+1}^2 - d_k^2)/R_k^2$ maximizes the distortion slope in Mitra–Virk's affine certificate under $\nu$-coherence. The guarantees take the following form. *(i)* An $O(kR/(\Delta\sqrt{m_{\min}}))$ bound on the excess risk over the empirical $\rho$-margin loss, driven by class-mean separation $\Delta$ and embedding radius $R$, paired in the sample-starved regime $m \lesssim R/\Delta$ with a Le Cam two-point identifiability lower bound ($m = \Omega(R/\Delta)$ needed to tell two $\Delta$-separated populations apart). *(ii)* The Mahalanobis margin under Ledoit–Wolf-shrunk covariance is the empirically strongest closed-form ranker on a heterogeneous 64-descriptor chemical-graph pool (mean Spearman $\rho = +0.56$ across 11 benchmarks, positive on 10 of 11); the isotropic surrogate $\Delta/\sqrt{\ell}$ admits a closed-form selection-consistency rate on the homogeneous protein/social pools. *(iii)* A training-time-decided *agreement diagnostic* for the nearest-centroid rule, with no per-prediction overhead, in three concrete radii (Pinelis, Gaussian plug-in, and variance-aware Pinelis–Bernstein); it bounds when the empirical and population nearest-centroid rules provably coincide, not whether a prediction is correct. Empirically, PLACE is the strongest diagram-based method on Orbit5k; its closed-form descriptor selector runs a mean 2.2 pp below the in-pool oracle across the graph benchmarks, and that oracle matches the strongest topology-based baseline within statistical noise on MUTAG and COX2. Remaining gaps fall into two diagnosable regimes (descriptor blindness on NCI1/NCI109; pool-coverage limits elsewhere). Carried with its full finite-sample Bernstein term and honest firing condition, the variance-aware radius clears only three benchmarks (NCI1, NCI109, DD); on the others it does not, though on MUTAG the empirical and population nearest-centroid rules nonetheless agree on every one of 940 held-out test predictions.

**Keywords:** persistent homology; topological data analysis; landmark embedding; kernel methods; classification certificates; minimax lower bound; descriptor selection; closed-form learning.

## 1 Introduction

Persistent homology produces a canonical topological signature of structured data—graphs, point clouds, shapes—called the *persistence diagram*: a finite multiset of points in the half-plane above the diagonal, augmented by a formal diagonal point $*$ (Figure 1). Stability under perturbation is well-understood (Cohen-Steiner et al., 2007; Chazal et al., 2009; 2016), but the varying cardinality and non-Hilbertian geometry of diagrams make them incompatible with standard machine learning. Existing vectorizations—persistence images (Adams et al., 2017), landscapes (Bubenik, 2015), kernels (Kusano et al., 2016; Carrière et al., 2017), and learned weights (Zhao & Wang, 2019)—all offer Lipschitz *upper* bounds on embedding distortion but no *lower* bound with explicit constants, so there is no guarantee that bottleneck-separated diagrams

remain separated after vectorization. Each method further carries hyperparameters—kernel bandwidth, image resolution, landscape level count, learned weight function—whose selection requires held-out data, so any downstream accuracy claim inherits the dependence on a validation split. Despite a decade of work, there is no way to inspect a trained persistence-diagram classifier and certify, before seeing test data, whether its predictions will be correct.

A second gap concerns how the input $X$ is turned into a persistence diagram in the first place. For graphs, this means choosing a *descriptor function* $f : X \to \mathbb{R}$—degree, centrality measures, curvature, or heat-kernel signatures of various scales (Ollivier, 2009)—whose sublevel sets define the filtration. For point clouds, the analogous choice is among filtration constructions parameterized by a radius (e.g., Vietoris–Rips or alpha complex). Different choices produce different diagrams and different downstream accuracy, with swings of 5–15 percentage points across our 12 benchmarks. Zhao & Wang (2019) highlight the effective use of multiple descriptor functions as a key open problem; in practice, the choice is made by trial-and-error against held-out labels, embedding a label-consuming hyperparameter selection into every reported accuracy number. There is no closed-form rule that ranks descriptors directly from training data.

This paper introduces **PLACE**, a persistence-based classifier with provable accuracy guarantees and a training-time agreement diagnostic for its nearest-centroid predictions—certifying, when it fires, that the empirical and population nearest-centroid rules coincide, not that a prediction is correct (Section 5). Our starting point is the *persistence landmark embedding* of Mitra & Virk (2024): the only *explicit* coarse embedding of $\mathcal{D}_n$ into a finite-dimensional Euclidean space with known distortion constants. The earlier work Mitra & Virk (2021) established existence via an asymptotic-dimension argument ($\mathrm{asdim}(\mathcal{D}_n, d_{\mathcal{B}}) = 2n$, linear in $n$); the 2024 construction makes that existence concrete, placing $M$ landmarks on a lattice in the birth-death plane at $N$ geometrically spaced scales, assigning each landmark a compactly supported hat function as a coordinate, and assembling an $n$-fold composition for $n$-point diagrams. The composition's $n$-fold structure is what makes the lower distortion bound $\rho_-$ *unconditional* on $\{d_{\mathcal{B}} \geq R_1\}$: every cross-pair gets a dedicated coordinate, so cancellation between hat-function contributions cannot occur. The explicit composition has dimension $M^n$ per scale, growing exponentially in the diagram cardinality $n$.

We replace it with a summation that evaluates the single-point coordinate at every point of the diagram and adds, dropping the embedding dimension to $\ell = O(MN)$ overall—linear in the grid size and the number of scales. The summation pooling is the source of computational tractability and of a structural trade-off: by collapsing the $M^n$ cross-pair coordinates of the $n$-fold form into $M$ summed coordinates, summation enables a cancellation construction (Remark 2.1) that the $n$-fold form rules out automatically. The lower bound is therefore *conditional* on $\nu$-coherence—a matching-free per-scale block-norm floor on the embedding difference (Proposition 2.1; holding on $\geq 99.7\%$ of pairs in the Section 6 audit). Linear complexity is bought at the price of a conditional lower bound, with the conditional close to universal in practice. Summation pooling additionally keeps the embedding *linear* in the empirical diagram measure—a property that max-pooled or order-statistic alternatives lack. The distortion constant depends on the per-scale terms $w_k^2 R_k^2$, whose optimization (Section 4) yields a closed-form weight rule (equation (2.13)) in one step. For downstream classification, the per-pair distortion is replaced by a data-dependent class-mean separation $\Delta$, which drives the theory in Section 3.

Three results establish the theory. First, with $R$ denoting the embedding radius, the excess risk of a linear SVM on the embedded features is $O\big(kR/(\Delta\sqrt{m_{\min}})\big)$ in the per-class training-set size (Theorem 3.1; the factor $k - 1$ comes from the one-vs-one reduction used in the proof); a Le Cam two-point argument shows that in the sample-starved regime $m \lesssim R/\Delta$ two $\Delta$-separated diagram populations cannot be reliably told apart (testing error $\geq 1/4$; Theorem 3.2), so $m = \Omega(R/\Delta)$ samples are necessary; the threshold depends on $\Delta$, not on the worst-case bottleneck separation.

Second, descriptor selection is itself closed-form: the Mahalanobis margin[1] $\hat{\rho}_{\mathrm{Mah}}$ between empirical class means under a Ledoit–Wolf-shrunk pooled covariance (Ledoit & Wolf, 2004)—the LDA Bayes-margin form

---

[1]Prasanta Chandra Mahalanobis (1893–1972) was an Indian statistician who founded the Indian Statistical Institute in 1931 and pioneered the use of large-scale sample surveys in national planning; his 1936 paper *On the Generalised Distance in Statistics* (Mahalanobis, 1936) introduced the covariance-aware distance that bears his name, the LDA Bayes-margin form of the Fisher discriminant ratio, which Section 4.1 of this paper adopts as the descriptor-selection rule on heterogeneous pools. We dedicate this work to his memory.

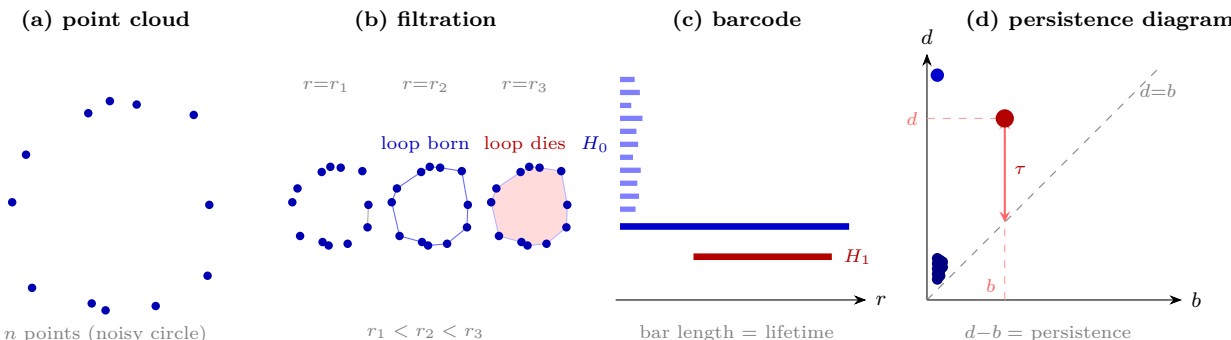

Figure 1: **From point cloud to persistence diagram. (a)** Noisy sample from a circle. **(b)** A growing complex (schematic) at radii $r_1 < r_2 < r_3$: as $r$ increases the complex fills in, so the 1-cycle born at $r_2$ dies at $r_3$. **(c)** Barcode; bar length equals feature lifetime. **(d)** Persistence diagram; each feature becomes a point $(b, d)$, with distance $\tau = d - b$ to the diagonal measuring topological significance. The 0-dim (blue) and 1-dim (red) points are shown together.

of the Fisher discriminant ratio—is computable from training labels without any held-out validation or learned weights, and rank-correlates with linear-SVM accuracy across 11 benchmarks with mean Spearman $\rho = +0.56$ (range $-0.24$ to $+0.89$, positive on 10 of 11; Section 6.3). Its simpler isotropic surrogate $\Delta/\sqrt{\ell}$ is ranking-consistent under a separation gap (Proposition 4.1, Corollary 4.1) and provides an upper-bound interpretation tied directly to Theorem 3.1, with Mahalanobis as the appropriate selector when structural homogeneity fails—the regime in which heterogeneous descriptor pools live (Remark 4.1).

Third, a scalar check $r_m < \frac{1}{2}\Delta$ certifies agreement between the empirical and population nearest-centroid classifiers on every input, with probability $\geq 1 - \alpha$ (Theorem 5.1). The bound provides three complementary radii: a non-asymptotic Pinelis form (dimension-free but $L^2$-bounded), an asymptotic Gaussian plug-in form (dimension penalty $\sqrt{\chi_\ell^2}$), and a non-asymptotic variance-aware Pinelis–Bernstein form that combines the dimension-freeness of the first with the operator-norm refinement of the second. Carried honestly—with its finite-sample linear term and the empirical firing condition $r_m < \frac{1}{4}\hat{\Delta}$ (Proposition 3.2)—the Pinelis–Bernstein radius clears the condition on only 3 benchmarks (NCI1, NCI109, DD; Table 6), and we read firing as a heuristic diagnostic of empirical/population nearest-centroid agreement rather than a per-prediction correctness certificate (Remark 5.3). The check is performed once from training statistics, has no per-prediction overhead, requires no calibration split—unlike conformal methods (Vovk et al., 2005)—and applies to individual point predictions rather than to sets; what it attests, when it fires, is agreement of the empirical nearest-centroid rule with the population rule, not correctness of the label (Section 5). Figure 2 gives the end-to-end view: raw graph or point cloud enters on the left, a label with its agreement diagnostic exits on the right, and every ingredient along the way is fixed analytically from the compact support size $L$ and the training labels.

The shift from worst-case distortion to class-mean separation also resolves a puzzle in the empirical literature: descriptors with vanishing pairwise separation (e.g., degree) can match the accuracy of descriptors with large separation (e.g., Ricci curvature), because $\Delta$ captures the aggregate distributional signal the classifier actually uses. Descriptor choice—not mass tuning or scale optimization—is the primary accuracy driver, and two closed-form selectors identify the right one without held-out data: the Mahalanobis margin $\hat{\rho}_{\mathrm{Mah}}$ on heterogeneous pools (the LDA Bayes-margin form of the Fisher discriminant ratio), and its isotropic surrogate $\hat{\Delta}/\sqrt{\ell}$ on homogeneous pools, computable directly from the raw input without a separate diagram-level analysis. This addresses an open question raised by Zhao & Wang (2019), who identified the effective use of multiple descriptor functions as a key challenge; the Mahalanobis-plus-surrogate pair provides a principled, closed-form answer to the descriptor-selection part of that challenge (Section 4). Across 12 benchmarks (Section 6), PLACE is the strongest diagram-based method on Orbit5k, its in-pool oracle matches the strongest topology-based baseline within statistical noise on MUTAG and COX2 (the closed-form selector running a mean 2.2 pp below that oracle), and it exhibits quantitative gaps on the remaining graph datasets,

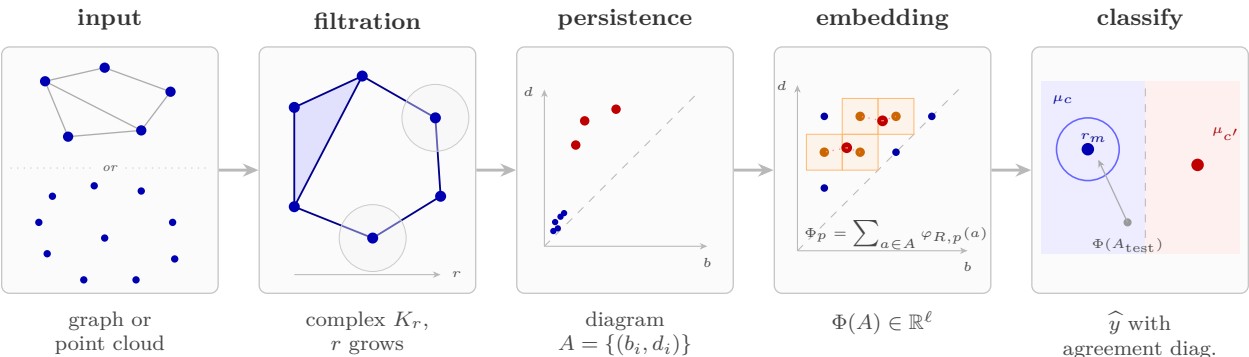

| input | filtration | persistence | embedding | classify |
|---|---|---|---|---|
| graph or point cloud | complex $K_r$, $r$ grows | diagram $A = \{(b_i, d_i)\}$ | $\Phi(A) \in \mathbb{R}^\ell$ | $\widehat{y}$ with agreement diag. |

Figure 2: **The PLACE pipeline.** A point cloud or graph (left) is converted to a persistence diagram through a filtration—a growing sequence of simplicial complexes—then embedded to $\mathbb{R}^\ell$ by summing hat-function coordinates over a landmark grid: each diagram point (red) contributes to the coordinates indexed by the landmarks (orange) whose $d_\mathcal{B}$-cover squares it falls within, via $\Phi_p(A) = \sum_{a \in A} \varphi_{R,p}(a)$. The embedded vector is then classified by a linear rule; the nearest-centroid variant additionally carries the training-time agreement diagnostic of Section 5 (it attests, when it fires, that the empirical and population nearest-centroid rules coincide, not that the label is correct). Every embedding ingredient—the descriptor choice, the grid scales $R_k$, the weights $w_k$, and the diagnostic threshold—is fixed analytically from training labels alone, with no held-out calibration; only the linear classifier's regularization is set by cross-validation.

all with no held-out calibration—the descriptor, scales, and weights are fixed analytically and only the linear classifier's regularization is cross-validated. The method's principled failures—e.g., on NCI1/NCI109, where classes are distinguished by discrete node labels that our continuous descriptors cannot access—are diagnosed by the same statistic, suggesting the descriptor pool, rather than the embedding machinery, is the bottleneck.

## 1.1 Our Contribution and Organization

Table 1: The four contributions of PLACE and where each is established.

| | Contribution | Guarantee / location |
|---|---|---|
| (i) | Linear-size summation embedding | lower-distortion floor under $\nu$-coherence (Prop. 2.1, Cor. 2.1) |
| (ii) | Margin excess-risk rate | $O(kR/(\Delta\sqrt{m}))$ + matching Le Cam lower bound (Thms 3.1–3.2) |
| (iii) | Closed-form descriptor selection | Mahalanobis margin; consistency rate (Prop. 4.1) |
| (iv) | Training-time NC agreement diagnostic | three concentration radii, per-dataset firing (Thm 5.1, Tab. 6) |

We make four contributions—all closed-form, computationally efficient, and validated on 12 benchmarks; Table 1 summarizes them.

**(i) A linear-size summation embedding with a lower-distortion certificate.** A summation embedding of dimension $\ell = O(MN)$ specializing the $n$-fold construction of Mitra & Virk (2024) to linear-in-grid complexity, with an explicit constant-floor lower bound on the bottleneck metric $\mathcal{D}_n$ holding under $\nu$-coherence (Proposition 2.1). The summation specialization trades the unconditional but exponential ($M^n$) lower bound of Mitra & Virk (2024) for a conditional but linear ($MN$) one; the matching-free $\nu$-coherence hypothesis holds on $\geq 99.7\%$ of cross-class pairs in the Section 6 audit. The closed-form scale weights $w_k^2 \propto (d_{k+1}^2 - d_k^2)/R_k^2$ (equation (2.13), Section 2) are the unique maximizer of the distortion slope $\lambda(\nu)$ in Mitra–Virk's affine certificate of Corollary 2.1. Beyond bi-Lipschitz stability, $\lambda(\nu)$ bridges geometry and statistics: it controls how the class-mean separation $\Delta$ in (ii) inherits from diagram-level separation (Proposition 3.1). $\nu$-coherence is the proof's actual mechanism (a per-scale block-norm floor on $\Phi(A) - \Phi(B)$); empirically it holds on $\geq 99.7\%$ of pairs in the Section 6 audit (100% on three of four benchmarks). The empirical rate of (ii) flows through $\Delta > 0$ directly via Theorem 3.1, independent of any pairwise condition.

**(ii) A margin excess-risk rate with a matching lower bound.** A margin-based excess-risk rate $O\big((k-1)R/(\Delta\sqrt{m_{\min}})\big)$, driven by class-mean separation $\Delta$ alone and independent of the cross-pair hypothesis of (i), with a matching Le Cam sample-starved two-point identifiability lower bound (two $\Delta$-separated populations are unidentifiable for $m \lesssim R/\Delta$); the upper rate uses no tunable parameters beyond the closed-form pipeline (Theorems 3.1–3.2, Section 3).

**(iii) A closed-form descriptor-selection rule.** A closed-form descriptor-selection rule given by the Mahalanobis margin under Ledoit–Wolf shrinkage—the LDA Bayes-margin form of the Fisher discriminant ratio—empirically the strongest closed-form ranker on a heterogeneous 64-descriptor chemical-graph stress test (mean Spearman $\rho = +0.56$ across 11 benchmarks, range $-0.24$ to $+0.89$, positive on 10 of 11), with the isotropic surrogate $\Delta/\sqrt{\ell}$ admitting a closed-form selection-consistency rate (Proposition 4.1, Corollary 4.1, Remark 4.1, Section 4).

**(iv) A training-time prediction certificate.** A certificate for individual point predictions, decided once at training time from $\hat{\Delta}$ and $r_m$, with no per-prediction overhead, in three complementary forms—a non-asymptotic Pinelis radius, an asymptotic Gaussian plug-in radius, and a non-asymptotic variance-aware Pinelis–Bernstein radius—with per-dataset firing-rate diagnostics across all 12 benchmarks (Theorem 5.1, Table 6, Section 5); unlike conformal prediction this requires neither a calibration split nor an inflation factor, only $\hat{\Delta}$ and (for the variance-aware forms) the empirical covariances $\hat{\Sigma}_c$. Empirically (Section 6), PLACE is the strongest diagram-based method on Orbit5k; its in-pool oracle matches the strongest topology-based baseline within statistical noise on MUTAG and COX2, with the closed-form selector a mean 2.2 pp below the oracle, and it exhibits quantitative gaps on the remaining graph datasets that fall into two diagnosable regimes (descriptor-blindness or pool-coverage limits; Section 6.3).

The remaining degree of freedom not analytically pinned by contribution (i)—the *positions* of the landmarks, which change the grid combinatorially—is left to future work.

## 1.2 Related Work

We situate PLACE at the intersection of three lines of work: certified machine learning, persistence diagram vectorizations and their neural extensions, and the Mitra–Virk landmark embedding (Mitra & Virk, 2021; 2024).

**Certified machine learning.** Three families of methods attach correctness guarantees to classifier predictions. *Conformal prediction* (Vovk et al., 2005; Lei et al., 2018; Vovk, 2013; Angelopoulos & Bates, 2023) constructs prediction *sets* $\hat{C}(x)$ with marginal coverage $\mathbb{P}(y \in \hat{C}(x)) \geq 1 - \alpha$ using a held-out calibration set; the guarantee is distribution-free but applies to the set, not to any single label, and requires data splitting. *Selective classification* (Chow, 1970; Geifman & El-Yaniv, 2017; 2019) and learning with rejection (Bartlett & Wegkamp, 2008; Cortes et al., 2016) let the classifier abstain on low-confidence inputs but provide no probabilistic correctness guarantee for accepted predictions. PLACE differs from both families: its certificate applies to individual point predictions, requires no calibration data, and is decided once at training time (Section 5).

**Persistence diagram vectorizations.** Persistence landscapes (Bubenik, 2015) embed a diagram as a sequence of piecewise linear functions, stable under the bottleneck distance but potentially of high dimension for peaked diagrams. Persistence images (Adams et al., 2017) discretize a weighted Gaussian mixture supported on the diagram onto a fixed grid; choice of weighting function and bandwidth are free parameters. Sliced Wasserstein and persistence scale-space kernels (Kusano et al., 2016; Carrière et al., 2017) avoid an explicit finite-dimensional feature map in favor of a positive-definite kernel over diagrams. Weighted kernels (WKPI) of Zhao & Wang (2019) learn the Gaussian weight function via metric learning on held-out labels. Neural extensions—PersLay (Carrière et al., 2020) and Persformer (Reinauer et al., 2021)—learn the vectorization end-to-end. A common feature of all the above is a Lipschitz *upper* bound on embedding distortion under bottleneck perturbation and the absence of any *lower* bound: bottleneck-distant diagrams may collapse to identical features. None of the above constructions carries a per-prediction correctness certificate.

See the recent survey of Ali et al. (2023) for a taxonomy of these vectorizations. Table 2 summarizes the comparison.

**Topology with neural networks.** A parallel line of work couples persistence with deep learning more tightly than a frozen vectorization. Hofer et al. (2017) first embedded persistence diagrams through a learnable layer trained end-to-end with a downstream network. Brüel Gabrielsson et al. (2020) expose filtration and persistence as a differentiable topology layer over general simplicial complexes, permitting task-driven filtrations. On graphs specifically, topological GNNs (Horn et al., 2022) inject persistence features as channels into message-passing architectures, reusing the inductive bias of standard GNN backbones (Xu et al., 2019; Zhang et al., 2018) while augmenting them with global topological summaries; the Euler characteristic transform of Röell & Rieck (2024) is a lightweight alternative that captures shape structure at GNN-competitive cost. These neural/GNN approaches typically reach higher empirical accuracy on label-rich datasets (NCI1, NCI109) by learning filtration and vectorization jointly on held-out data; the distinguishing contribution of PLACE is orthogonal—an analytically fixed embedding that admits a closed-form descriptor-selection criterion and a training-time prediction-agreement certificate (Section 5), neither of which, to our knowledge, has been demonstrated for the learned families above.

Table 2: Persistence diagram vectorizations at a glance. **Lipschitz:** upper stability under the bottleneck distance. **Lower dist.:** an explicit lower bound on $\|\Phi(a) - \Phi(a')\|$ (multiplicative or constant-floor), on the indicated subspace of diagrams. **Poly. dim:** embedding dimension polynomial in the grid size $M$. **Analytic embed.:** all embedding parameters (scales, weights, kernel width, grid resolution) are fixed analytically—no held-out calibration and no learned weights *for the embedding* (the downstream linear-SVM regularization is still cross-validated; the pipeline is not tuning-free end to end). **Cert.:** a training-time prediction certificate of some form. Mitra–Virk's lower modulus $\rho_-$ certifies metric *distortion* (topological differences survive embedding); PLACE's $\lambda(\nu)$ certifies the same and additionally drives a *classification-agreement* guarantee (the empirical prediction matches the population prediction with probability $\geq 1 - \alpha$, Theorem 5.1). Among the tabulated methods, PLACE is the only one with an entry in every column; the comparison is over these criteria and this method set, not a claim of universal uniqueness.

| Method | Lipschitz | Lower dist. | Poly. dim | Analytic embed. | Cert. |
|---|---|---|---|---|---|
| Landscapes (2015) | ✓ | — | ✓ | ✗ (levels) | ✗ |
| Persistence images (2017) | ✓ | — | ✓ | ✗ ($\sigma$, grid, weight) | ✗ |
| SW / PSS kernels (2016; 2017) | ✓ | — | implicit | ✗ (bandwidth) | ✗ |
| WKPI (2019) | ✓ | — | ✓ | ✗ (learned $w$) | ✗ |
| PersLay / Persformer (2020; 2021) | learned | — | ✓ | ✗ (end-to-end) | ✗ |
| Mitra–Virk asdim (2021) | ✓ | existential (asdim $= 2n$) | — | — | metric only |
| Mitra–Virk $n$-fold (2024) | ✓ | ✓ $\rho_-$ on $\{d_\mathcal{B} \geq R_1\}$, unconditional | ✗ ($NM^n$) | ✓ | metric only |
| **PLACE (ours)** | ✓ | ✓ $\rho_-$ on $\mathcal{D}_n$, conditional ($\nu$-coherent, $\geq 99.7\%$) | ✓ ($MN$) | ✓ | classification |

**Mitra–Virk landmark embedding and why we build on it.** Mitra & Virk (2021) establish the coarse embedding $\mathcal{D}_n \hookrightarrow \ell^2$ existentially via proving $\mathrm{asdim}(\mathcal{D}_n, d_\mathcal{B}) = 2n$, without making the coarse embedding explicit. Mitra & Virk (2024) give the first explicit such coarse embedding with computable distortion constants $\rho_-$ on $\{d_\mathcal{B} \geq R_1\}$; their construction is unconditional—no matching or coherence hypothesis is required—at the cost of $M^n$ coordinates per scale, exponential in the diagram cardinality. Their work is purely metric-theoretic and does not address classification or downstream learning tasks. Three geometric features of their construction produce that bound, and PLACE is designed to retain them. *(i) Compactly supported ramp coordinates.* Each $\varphi_{R,p}$ is a 1-Lipschitz hat function with fixed peak $3R/2$ and bounded support in bottleneck distance; pointwise changes translate into bounded, traceable changes in the embedded vector. Gaussian kernels used by persistence images spread mass across all landmarks, so no single coordinate is responsible for a local displacement and the constants in any lower bound degrade with the bandwidth. *(ii) Cover structure of the grid.* The landmark lattice is designed so every diagram point lies within $3R/2$ of some landmark; this is what lifts the pointwise Lipschitz property into a bi-Lipschitz embedding (Proposition 2.1). Order-statistic constructions such as persistence landscapes (Bubenik, 2015) are nonlinear in the empirical diagram measure, and kernel-based vectorizations (SW, PSS) work implicitly and admit no finite-dimensional grid with this property. *(iii) Analytically optimal weights and scales.* The dis-

tortion slope $\lambda(\nu)$ of Corollary 2.1 admits a unique weight-vector argmax in closed form (equation (2.13) in Section 4), eliminating the hyperparameter search (bandwidth, resolution, learned weighting) that PI and WKPI require.

PLACE retains these three ingredients but replaces the $n$-fold composition—$M^n$ coordinates per scale, unconditional lower bound—with a summation over single-point evaluations (Section 2), reducing the per-scale dimension to $M$ and the total to $\ell = O(MN)$. Summation pooling collapses the $M^n$ cross-pair coordinates into $M$, which is what enables the cancellation construction of Remark 2.1; the resulting lower bound is conditional on $\nu$-coherence rather than unconditional. The trade is *exponential-unconditional* (Mitra & Virk, 2024) for *linear-conditional* (PLACE), with empirical near-universality of $\nu$-coherence ($\geq 99.7\%$; Section 6) making the conditional close to operational. At the same time, PLACE trades Mitra and Virk's individual-pair $\rho_-$ guarantee for a data-dependent class-mean separation $\Delta$ that drives the classification theory of Section 3. The closed-form specification is what lets PLACE deliver a classifier, a descriptor ranking, and a per-prediction certificate from training labels alone; stripped of learned weights, WKPI (Zhao & Wang, 2019) reduces to an ordinary persistence image, and the WKPI numbers in Table 10 are achievable only with the learned weight function, not with the underlying vectorization.

## 2 Persistence Landmark Embedding

A *persistence diagram* $A = \{a_1, \ldots, a_n\}$ is a finite multiset of $n$ points $(b_i, d_i)$ with $d_i > b_i \geq 0$ (Figure 1); we write $\mathcal{D}_n$ as the space of all diagrams on $n$ points, and $\mathcal{D} = \bigcup_n \mathcal{D}_n$ for the space of all such finite diagrams. The *bottleneck distance* $d_{\mathcal{B}}(A, B) = \min_\sigma \max_i d_\infty(a_i, b_{\sigma(i)})$ is the optimal matching cost (Cohen-Steiner et al., 2007), where the matching $\sigma$ pairs points of $A$ with points of $B$ and matches any unmatched points of $A$ or $B$ to the formal diagonal $*$, with matching cost $d_\infty(a, *) = (d - b)/2$ for $a = (b, d)$. On single-point diagrams $\mathcal{D}_1$, $d_{\mathcal{B}}(p, p') = \min\{d_\infty(p, p'), \max\{d_\infty(p, *), d_\infty(p', *)\}\}$.

We embed diagrams into $\mathbb{R}^\ell$ by specializing the landmark construction of Mitra & Virk (2024); all diagrams are assumed to be supported in the compact region $\mathcal{T}_L := \{(b, d) : 0 \leq b < d \leq L\}$ and to have cardinality bounded by some $N_{\max} < \infty$ (the specific value used in experiments is given in Section 6). The cardinality bound is structurally necessary, not merely a practical convenience:

- Carrière & Bauer (2019) show that even on bounded-cardinality diagrams $\mathcal{D}_n$ (with $n$ fixed), no bi-Lipschitz embedding into any Hilbert space exists.

- Mitra & Virk (2021) and Bubenik & Wagner (2020) show that even on the union $\mathcal{D} = \bigcup_n \mathcal{D}_n$ of all finite diagrams, no coarse embedding into any Hilbert space exists.

- Zava (2025) extends the impossibility to the unbounded-cardinality Gromov–Hausdorff space (whose 1D Euclidean–Hausdorff specialization contains $(\mathcal{D}, d_{\mathcal{B}})$ as a subspace), ruling out coarse embeddings into any uniformly convex Banach space and hence into any Hilbert space.

The matching positive direction in the bounded regime is established by Mitra & Virk (2021), who prove $\mathrm{asdim}(\mathcal{D}_n, d_{\mathcal{B}}) = 2n$ existentially. Mitra & Virk (2024) make the existence concrete: their $n$-fold landmark composition gives the first explicit coarse embedding of $\mathcal{D}_n$ with computable distortion constants $\rho_-$ on $\{d_{\mathcal{B}} \geq R_1\}$, at $M^n$ coordinates per scale. The construction (2.3) below specializes that $n$-fold form to a summation, reducing complexity to $\ell = O(MN)$ at the cost of a $\nu$-coherence conditional on the lower bound (Proposition 2.1). For the analysis below, each diagram is padded with diagonal points $*$ to cardinality exactly $N_{\max}$, so all diagrams lie in $\mathcal{D}_{N_{\max}}$; this leaves $d_{\mathcal{B}}$ unchanged and adds zero contribution to every non-diagonal landmark coordinate, so the embedding $\Phi$ of (2.3) below is unaffected on the original points. Fix a scale $R > 0$. The *landmark grid* $\mathbb{G}_R$ is the finite set of single-point diagrams in $\mathcal{D}_1$ whose single point, called a *landmark*, lies in the lattice

$$\mathbb{G}_R = \big\{ (mR, nR) : m \in \{1, 3, 5, \ldots\}, \ n \in \{4, 6, 8, \ldots\}, \ n \geq m + 3 \big\} \cap [0, L]^2. \tag{2.1}$$

We write $\mathbb{G}_R^+ := \mathbb{G}_R \cup \{*\}$, adjoining the formal diagonal landmark $*$. The parity condition makes the $d_{\mathcal{B}}$-balls of radius $\frac{3R}{2}$ centered at the points of $\mathbb{G}_R^+$ cover $\mathcal{D}_1$ with multiplicity at most four (Mitra & Virk, 2024, Lemma 3.5) (Figure 3).

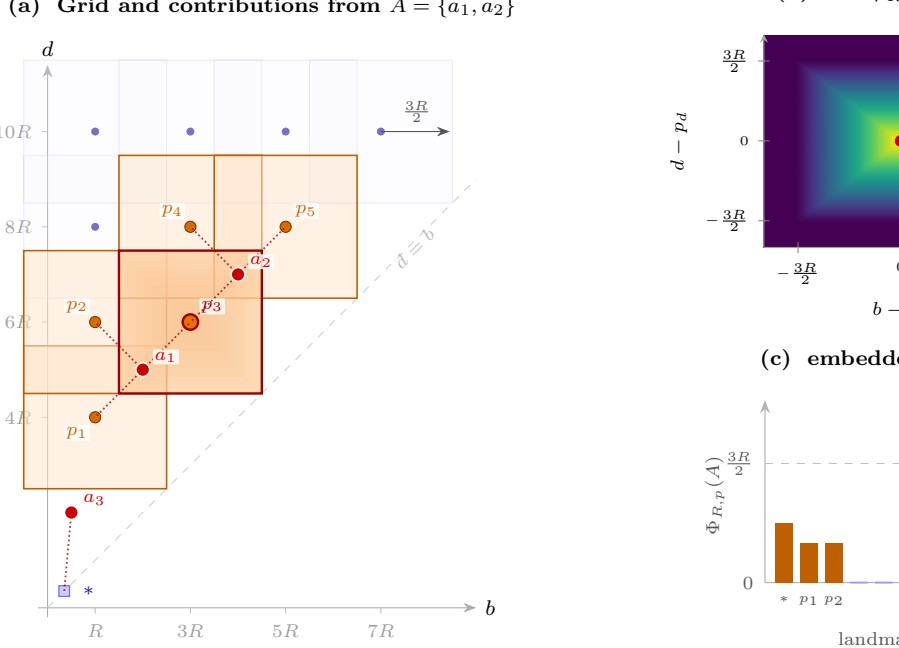

Figure 3: **Landmark grid, hat coordinate, and summation embedding. (a)** Grid $\mathbb{G}_R$ (odd $m$, even $n$, $n \geq m + 3$) with $d_{\mathcal{B}}$-cover squares of radius $\frac{3R}{2}$; diagram $A = \{a_1, a_2, a_3\}$ (red): $a_1, a_2$ each fall in three lattice landmarks (with $p_3$ shared—the summation site), while the low-persistence point $a_3$ contributes only to the diagonal landmark $*$. **(b)** Hat $\varphi_{R,p}(x) = \max\{\frac{3R}{2} - d_{\mathcal{B}}(p, x), 0\}$: a $d_{\infty}$-pyramid peaking at $p$; its level sets are previewed by the concentric shading on $p_3$ in (a). **(c)** Embedded vector $\Phi_R(A)$ with one coordinate per landmark; $p_3$ receives the sum of two contributions as a stacked bar and $*$ receives a single contribution from $a_3$. Multiscale $\Phi$ concatenates such blocks at scales $R_1 < \cdots < R_N$ under the scale configuration $\nu = \{(R_k, w_k)\}_{k=1}^N$.

To each landmark $p \in \mathbb{G}_R^+$ we attach the compactly supported coordinate function $\varphi_{R,p} : \mathcal{D}_1 \to [0, 3R/2]$ given by $\varphi_{R,p}(x) = \max\{\frac{3R}{2} - d_{\mathcal{B}}(p, x), 0\}$, a hat function of height $\frac{3R}{2}$ supported in the $d_{\mathcal{B}}$-ball of radius $\frac{3R}{2}$ around $p$ (Figure 3(b)). Stacking the coordinate functions into a map $\varphi_R : \mathcal{D}_1 \to \mathbb{R}^M$, $M := |\mathbb{G}_R^+|$, by $\varphi_R(x) = (\varphi_{R,p}(x))_{p \in \mathbb{G}_R^+}$ produces a $2\sqrt{2}$-Lipschitz embedding of single-point diagrams into $\mathbb{R}^M$ (Mitra & Virk, 2024, Lemma 3.8).

For a diagram $A = \{a_1, \ldots, a_n\} \in \mathcal{D}_n$, we evaluate each coordinate on each point and sum, defining the *single-scale summation embedding*

$$\Phi_R(A) = \left( \sum_{a \in A} \varphi_{R,p}(a) \right)_{p \in \mathbb{G}_R^+} \in \mathbb{R}^M. \tag{2.2}$$

This replaces Mitra–Virk's $n$-point bottleneck evaluation (which requires $M^n$ coordinates) with $|A|$ single-point evaluations at a fixed single-scale grid, and preserves linearity in the empirical diagram measure—properties our classification theory (Section 3) relies on. We use *embedding* in the feature-map sense throughout: $\Phi_R$ is Lipschitz-stable (2.5) and, aggregated across scales, carries the lower-distortion certificate of Proposition 2.1 under $\nu$-coherence; it is not claimed injective on all of $\mathcal{D}_n$, since incoherent pairs can collapse (Remark 2.1).

A single scale $R$ yields a lower distortion bound only for pairs with $d_{\mathcal{B}} \geq 3R$ (Mitra & Virk, 2024, Lemma 3.18), so a coarse scale misses close pairs entirely. A very fine scale would cover all distances, but the guaranteed separation is only $R\sqrt{2}/8$, which vanishes with $R$. Following Mitra & Virk (2024, Section 4), we

compose embeddings across multiple scales: fine scales supply a lower bound for close pairs, coarse scales for distant ones, and each scale contributes a separation proportional to its own $R$. Fix $0 < R_1 < \cdots < R_N \leq L$ and weights $\{w_k\}_{k=1}^{N}$ with $\sum_k w_k^2 = 1$; the *multiscale landmark embedding* $\Phi : \mathcal{D} \to \mathbb{R}^\ell$ concatenates the single-scale embeddings with block weights,

$$\Phi(A) \;=\; \left( w_k\, 2^{-3/2}\, \Phi_{R_k}(A) \right)_{k=1}^{N} \;\in\; \mathbb{R}^\ell, \qquad \ell = \sum_{k=1}^{N} |\mathbb{G}_{R_k}^{+}|, \tag{2.3}$$

where the factor $2^{-3/2}$ renormalizes each block to be 1-Lipschitz (given the $2\sqrt{2}$-Lipschitz per-block bound) and the weights $w_k$ balance scales' contributions. We collect the embedding parameters as the *scale configuration*

$$\nu \;:=\; \left\{ (R_k, w_k) \right\}_{k=1}^{N}, \tag{2.4}$$

and write $\Phi(A; \nu)$ when these parameters need emphasis. As shown in Proposition 2.1 and Corollary 2.1, the per-scale combinations $w_k^2 R_k^2$ drive the sharp certificate (2.8) and the distortion slope $\lambda(\nu)$ in (2.11), with each such term measuring scale $k$'s contribution to the bi-Lipschitz guarantee. Because $\Phi$ sums $|A|$ single-point evaluations, its Lipschitz constant depends on the diagram cardinality. For any $A, B \in \mathcal{D}$ with $\max(|A|, |B|) \leq N_{\max}$,

$$\|\Phi(A) - \Phi(B)\|_{\ell^2} \;\leq\; N_{\max}\, d_{\mathcal{B}}(A, B). \tag{2.5}$$

The constant $N_{\max}$ follows from four steps.

*(i)* **Per-point Lipschitz.** The single-scale map $\varphi_{R_k} : \mathcal{D}_1 \to \mathbb{R}^{|\mathbb{G}_{R_k}^{+}|}$ is $2\sqrt{2}$-Lipschitz in $d_{\mathcal{B}}$ by (Mitra & Virk, 2024, Lemma 3.8).

*(ii)* **Summation over matched pairs.** Fix an optimal matching $\sigma$ realizing $d_{\mathcal{B}}(A, B)$. Summing per-point displacements and applying the triangle inequality coordinate-wise in $\mathbb{R}^{|\mathbb{G}_{R_k}^{+}|}$,

$$\left\| \Phi_{R_k}(A) - \Phi_{R_k}(B) \right\|_{\ell^2} \;\leq\; \sum_{i=1}^{N_{\max}} \left\| \varphi_{R_k}(a_i) - \varphi_{R_k}(b_{\sigma(i)}) \right\|_{\ell^2} \;\leq\; 2\sqrt{2}\, N_{\max}\, d_{\mathcal{B}}(A, B).$$

*(iii)* **Per-block normalization.** The block prefactor $w_k \cdot 2^{-3/2}$ cancels the $2\sqrt{2} = 2^{3/2}$ from step (ii):

$$\left\| w_k\, 2^{-3/2} \left( \Phi_{R_k}(A) - \Phi_{R_k}(B) \right) \right\|_{\ell^2} \;\leq\; w_k \cdot 2^{-3/2} \cdot 2^{3/2}\, N_{\max}\, d_{\mathcal{B}}(A, B) \;=\; w_k\, N_{\max}\, d_{\mathcal{B}}(A, B).$$

*(iv)* **$\ell^2$-concatenation over scales.** Squaring per-block bounds, summing over $k$, and using the normalization $\sum_{k=1}^{N} w_k^2 = 1$,

$$\|\Phi(A) - \Phi(B)\|_{\ell^2}^2 = \sum_{k=1}^{N} \left\| w_k\, 2^{-3/2} (\Phi_{R_k}(A) - \Phi_{R_k}(B)) \right\|_{\ell^2}^2$$

$$\leq N_{\max}^2\, d_{\mathcal{B}}(A, B)^2 \sum_{k=1}^{N} w_k^2 \;=\; N_{\max}^2\, d_{\mathcal{B}}(A, B)^2.$$

Taking square roots gives (2.5). Under the top-$N_{\max}$ persistence filter, the stability constant is thus a fixed multiple of $d_{\mathcal{B}}(A, B)$ and the embedding remains Lipschitz on the truncated diagram space.

The upper bound (2.5) guarantees that close diagrams remain close after embedding. The following proposition establishes a converse: under a minimum-distance condition, the embedding does not collapse distinct diagrams, yielding a constant-floor lower bound. The lower bound requires a structural assumption on the cross-pair geometry of the embedding-difference contributions, which we record next.

**Definition 2.1** ($\nu$-coherence [Empirical]). *Fix a scale configuration $\nu = \{(R_k, w_k)\}_{k=1}^{N}$ as in (2.4), with the corresponding scale-block embedding $\Phi_{R_k}$ of Section 2. For a pair $(A, B) \in \mathcal{D}_n \times \mathcal{D}_n$, call an index $k \in \{1, \ldots, N\}$ an* active scale *for $(A, B)$ if $3R_k \leq d_{\mathcal{B}}(A, B)$. We say $(A, B)$ is $\nu$-coherent if the per-scale block-norm satisfies the floor*

$$\left\| \Phi_{R_k}(A) - \Phi_{R_k}(B) \right\|_{\ell^2}^2 \;\geq\; \tfrac{R_k^2}{32} \tag{2.6}$$

*at every active scale $k$ for $(A, B)$.*

The floor constant $R_k^2/32$ is inherited from the single-point Mitra–Virk lemma (Lemma 3.18 of Mitra & Virk, 2024, with the single-point constant of Lemma 3.9 therein): for any $a, a' \in \mathcal{D}_1$ with $d_{\mathcal{B}}(a, a') \geq 3R_k$, $\|\varphi_{R_k}(a) - \varphi_{R_k}(a')\|_{\ell^2}^2 \geq R_k^2/32$, and the constant is sharp—an explicit worst-case configuration of $a, a'$ on the cubic landmark lattice $\mathbb{G}_{R_k}$ realizes equality. The $1/32$ is set by the geometry of the hat function $\varphi_{R_k,p}(x) = \max\{3R_k/2 - d_{\mathcal{B}}(x, p), 0\}$ and the multiplicity-4 lattice cover (Lemma 3.5 of Mitra & Virk, 2024); we adopt this constant as given. For multi-point $\mathcal{D}_n$ with $n \geq 2$, (2.6) promotes the single-pair geometry to the corresponding *block-norm* statement, which is no longer automatic from $d_{\mathcal{B}}(A, B) \geq 3R_k$ but instead constrains how matched and unmatched point contributions add at scale $R_k$. The condition is thus a per-scale *non-cancellation* hypothesis: it fails precisely when the summed contributions of $A$ and $B$ destructively interfere at a scale (Remark 2.1), and it holds automatically under a bound-free geometric sufficient condition (Lemma 2.1 below). Crucially, $\nu$-coherence is a hypothesis on the *per-scale block norms*, whereas its consequence (2.8) is a statement about the *aggregate* embedding norm; the two are not equivalent. The implication is strict: there exist pairs for which (2.8) holds while $\nu$-coherence fails at an active scale (the compensation regime, Remark 2.2), so (2.8) does not imply $\nu$-coherence and the condition is not a restatement of the bound it certifies.

**Lemma 2.1** (Disjoint-activation sufficient condition). *Let $(A, B) \in \mathcal{D}_n \times \mathcal{D}_n$ with $d_{\mathcal{B}}(A, B) \geq 3R_1$. Fix an optimal bottleneck matching $\sigma$ realizing $d_{\mathcal{B}}(A, B)$. If, at every active scale $k$, each matched pair $(a_i, b_{\sigma(i)})$ activates a set of landmarks disjoint from those activated by every other matched pair, then $(A, B)$ is $\nu$-coherent (Definition 2.1).*

*Proof.* Fix an active scale $k$. Under disjoint activation the per-pair displacements $\varphi_{R_k}(a_i) - \varphi_{R_k}(b_{\sigma(i)})$ have pairwise-disjoint coordinate supports, so they add in quadrature: $\|\Phi_{R_k}(A) - \Phi_{R_k}(B)\|_{\ell^2}^2 = \sum_i \|\varphi_{R_k}(a_i) - \varphi_{R_k}(b_{\sigma(i)})\|_{\ell^2}^2$. At least one matched pair realizes the bottleneck—an off-diagonal pair $(a_i, b_{\sigma(i)})$, or a diagonal projection $(a_i, *)$ when the realizer is a point matched to the diagonal. In either case that pair's single-point bottleneck distance equals $d_\infty(a_i, b_{\sigma(i)}) = d_{\mathcal{B}}(A, B) \geq 3R_k$: for an off-diagonal pair, optimality of $\sigma$ rules out $d_\infty$ exceeding both diagonal costs $\max\{d_\infty(a_i, *), d_\infty(b_{\sigma(i)}, *)\}$ (else rematching both partners to $*$ would strictly lower the bottleneck), so the single-point distance is $d_\infty$ itself; for a diagonal projection it is immediate. The single-point floor (Lemma 3.18 of Mitra & Virk, 2024, single-point constant of Lemma 3.9 therein) then bounds that summand below by $R_k^2/32$; the remaining summands are non-negative. Hence the block norm meets the floor (2.6) at scale $k$, and, $k$ being arbitrary among active scales, $(A, B)$ is $\nu$-coherent. $\qquad\square$

Aggregating these per-scale floors across active scales yields the Lipschitz lower bound below.

**Proposition 2.1** (Distortion bounds on $\mathcal{D}_n$). *Let $A, B \in \mathcal{D}_n$ with $n \geq 1$ points each.*

(a) **Stability.** *Unconditionally,*

$$\|\Phi(A) - \Phi(B)\|_{\ell^2} \;\leq\; n\, d_{\mathcal{B}}(A, B). \tag{2.7}$$

(b) **Sharp certificate.** *Suppose $d_{\mathcal{B}}(A, B) \geq 3R_1$ and $(A, B)$ is $\nu$-coherent (Definition 2.1). Then*

$$\|\Phi(A) - \Phi(B)\|_{\ell^2} \;\geq\; \tfrac{1}{16} \sqrt{\sum_{k:\, 3R_k \leq d_{\mathcal{B}}(A,B)} w_k^2 R_k^2}. \tag{2.8}$$

*The right-hand side is the sharpest bound deducible from $\nu$-coherence under the orthogonal scale-decomposition of $\Phi$: given only the per-scale floors of Definition 2.1, no larger lower bound on $\|\Phi(A) - \Phi(B)\|_{\ell^2}$ follows, since the aggregation $\|\Phi(A) - \Phi(B)\|_{\ell^2}^2 = \sum_k (w_k^2/8)\, \|\Phi_{R_k}(A) - \Phi_{R_k}(B)\|_{\ell^2}^2$*

*is minimized when each active block meets its floor. Each floor is individually sharp, realized scale-by-scale by a (scale-dependent) worst-case pair on $\mathbb{G}_{R_k}$; we do not claim a single pair saturates all active scales simultaneously (Remark 2.2).*

*Proof.* *(a)* **Stability.** Specializing (2.5) with $N_{\max} = n$ gives (2.7).

*(b)* **Sharp certificate.** The full embedding difference decomposes orthogonally across scales as $\Phi(A) - \Phi(B) = \left( w_k \cdot 2^{-3/2} \left( \Phi_{R_k}(A) - \Phi_{R_k}(B) \right) \right)_{k=1}^{N}$, so

$$\|\Phi(A) - \Phi(B)\|_{\ell^2}^2 = \sum_{k=1}^{N} \frac{w_k^2}{8} \left\| \Phi_{R_k}(A) - \Phi_{R_k}(B) \right\|_{\ell^2}^2.$$

Scales $k$ with $3R_k > d_{\mathcal{B}}(A, B)$ contribute non-negatively; under $\nu$-coherence (2.6), every active scale $(3R_k \leq d_{\mathcal{B}}(A, B))$ contributes at least $w_k^2/8 \cdot R_k^2/32 = w_k^2 R_k^2/256$. Summing across the active scales,

$$\|\Phi(A) - \Phi(B)\|_{\ell^2}^2 \geq \frac{1}{256} \sum_{k:3R_k \leq d_{\mathcal{B}}(A,B)} w_k^2 R_k^2, \tag{2.9}$$

and taking square roots yields (2.8).

*Per-scale sharpness.* The floor at each active scale is individually attained. Fix an active scale $k$ and take singletons $A = \{a\}$, $B = \{a'\}$ with $a, a'$ in the Mitra–Virk single-point worst-case configuration on $\mathbb{G}_{R_k}$ (Lemma 3.18 of Mitra & Virk, 2024, single-point constant of Lemma 3.9 therein), so that $\|\varphi_{R_k}(a) - \varphi_{R_k}(a')\|_{\ell^2}^2 = R_k^2/32$. Then the scale-$k$ contribution to (2.9) meets its floor exactly, showing the constant $1/256$ cannot be improved. When only one scale is active $(3R_1 \leq d_{\mathcal{B}}(A, B) < 3R_2)$, this pair realizes equality in (2.8). When two or more scales are active the bound (2.8) is not attained by any single pair: the worst-case configuration is a specific position on the lattice $\mathbb{G}_{R_k}$, whose spacing scales with $R_k$, so a pair extremal for $\mathbb{G}_{R_k}$ is non-extremal for the geometrically distinct grid $\mathbb{G}_{R_{k'}}$, $k' \neq k$. Concretely, once $d_{\mathcal{B}}(A, B) \gg 3R_j$ for a coarser active scale $R_j$, that block sits strictly above its floor. Equation (2.8) is therefore the sharp *deductive* consequence of the per-scale floors, tight scale-by-scale, rather than a bound attained simultaneously across active scales; this is consistent with the empirical ratios of Table 8, which exceed unity by one to three orders of magnitude on real pairs.

$\square$

Among existing persistence vectorizations, only the Mitra–Virk construction (Mitra & Virk, 2024) (of which our $\Phi$ is the summation specialization) carries such an explicit lower distortion bound. Each additional scale contributes a positive $w_k^2 R_k^2$ to the sum on the right-hand side of (2.8), so the lower bound strictly increases with $N$—a concrete benefit of the multiscale construction beyond coverage.

For downstream use—the classification theory of Section 3 requires a Lipschitz constant linear in $d_{\mathcal{B}}(A, B)$, not the step form—we record the corollary obtained by replacing the right-hand side of (2.8) with its largest linear lower bound through $(R_1, 0)$.

**Corollary 2.1** (Mitra–Virk affine certificate [Proved]). *Let $A, B \in \mathcal{D}_n$ with $3R_1 \leq d_{\mathcal{B}}(A, B) \leq L$, and assume $(A, B)$ is $\nu$-coherent (Definition 2.1) at every active scale. Then*

$$\|\Phi(A) - \Phi(B)\|_{\ell^2} \geq \rho_-(d_{\mathcal{B}}(A, B); \nu) = \lambda(\nu) \left( d_{\mathcal{B}}(A, B) - R_1 \right), \tag{2.10}$$

*where the slope*

$$\lambda(\nu) := \frac{1}{48} \min \left\{ \min_{2 \leq i \leq N} \frac{\sqrt{\sum_{k=1}^{i-1} w_k^2 R_k^2}}{R_i - R_1}, \frac{\sqrt{\sum_{k=1}^{N} w_k^2 R_k^2}}{L - R_1} \right\}. \tag{2.11}$$

*The slope $\lambda(\nu)$ coincides with the closed form obtained by Mitra & Virk (Theorem 5.1 of Mitra & Virk, 2024) for the $n$-fold construction; here it is derived directly from the summation embedding via Proposition 2.1(b).*

*Proof.* On the stated range $d_{\mathcal{B}}(A, B) \geq 3R_1$ at least the finest scale is active, and the affine bound follows directly from Proposition 2.1(b)'s step-form (2.8): the line through $(R_1, 0)$ in the scale-coordinate parametrization lies below the step, with worst-case correction factor $\frac{1}{3}$ ensuring it stays below every breakpoint. The resulting prefactor $\frac{1}{3 \cdot 2^{n+3}}$ specializes to $\frac{1}{48}$ at $n = 1$, matching the closed form of Mitra & Virk (2024, Theorem 5.1). □

**Remark 2.1** (Canonical incoherent case). *$\nu$-coherence fails in the cancellation construction in which the empirical measures of $A$ and $B$ destructively interfere across the landmark grid. The canonical example is $A = \{a_1, a_2\}$, $B = \{a_1 + \delta, a_2 - \delta\}$ with $\|\delta\|$ small. Then $d_{\mathcal{B}}(A, B) = \|\delta\| > 0$; if $a_1$ and $a_2$ share a covering landmark $p$, the hat-function contributions shift in opposite directions, $\varphi_{R,p}(a_1 + \delta) - \varphi_{R,p}(a_1) \approx -(\varphi_{R,p}(a_2 - \delta) - \varphi_{R,p}(a_2))$, so $\Phi_{R_k}(A)(p) \approx \Phi_{R_k}(B)(p)$ at every active scale activating $p$. The per-scale block-norm $\|\Phi_{R_k}(A) - \Phi_{R_k}(B)\|_{\ell^2}^2$ collapses below $R_k^2/32$ and the lower bound fails. $\nu$-coherence rules out exactly this collapse: the empirical measure of $A$ and $B$ disagree enough at the per-scale block level to preserve the single-pair floor. It holds automatically whenever pairs of points in $A$ and $B$ activate disjoint landmarks at every active scale, since the per-pair contributions then add in quadrature; this is the sufficient condition made precise in Lemma 2.1.*

**Remark 2.2** (Empirical scope and compensation slack). *$\nu$-coherence holds on $\geq 99.7\%$ of qualifying cross-class pairs across the four chemical benchmarks (Section 6, Table 7; 100% on three of them), and the certificate's conclusion (2.8) holds on 100% (Table 8). The residual $\sim 0.3\%$ gap on PTC—pairs where the aggregate certificate holds yet $\nu$-coherence fails at some active scale—is the compensation regime: a per-scale shortfall $\|\Phi_{R_{k^\star}}(A) - \Phi_{R_{k^\star}}(B)\|_{\ell^2}^2 < R_{k^\star}^2/32$ at some active $k^\star$ can be made up by overshoots at other active scales, since the aggregate $\|\Phi(A) - \Phi(B)\|_{\ell^2}^2 = \sum_k (w_k^2/8) \|\Phi_{R_k}(A) - \Phi_{R_k}(B)\|_{\ell^2}^2$ need not be per-scale tight. This exhibits the strict inequivalence noted after Definition 2.1: the aggregate certificate (2.8) can hold while $\nu$-coherence fails at an active scale, so (2.8) does not imply (2.6). Consequently $\nu$-coherence is strictly stronger than the bound it yields—it is a hypothesis on the per-scale blocks, not a restatement of the aggregate conclusion. The sharpness in Proposition 2.1(b) is at the per-scale level (each floor is individually attained), not a biconditional on the aggregate (2.8).*

**Closed-form scale weights.** Proposition 2.1(b) determines the canonical scale weights via an *equimarginal allocation* principle. At each activation $\delta = 3R_k^+$, the $k$-th scale contributes squared step height $\frac{1}{256} w_k^2 R_k^2$ to $\sigma^2(\delta)$, so $w_k^2 R_k^2$ is the scale's marginal certificate gain at activation. We allocate the budget $\sum_k w_k^2 = 1$ so that the cumulative pre-jump heights $S_i := \sum_{k<i} w_k^2 R_k^2$ track the squared scale-coordinate threshold $(R_i - R_1)^2$ through $(R_1, 0)$ collinearly:

$$S_i = c^2 d_i^2, \qquad d_i := R_i - R_1, \ d_{N+1} := L - R_1, \tag{2.12}$$

for a common slope $c$. Telescoping $S_{k+1} - S_k = c^2(d_{k+1}^2 - d_k^2)$ yields the closed form

$$w_k^2 \propto \frac{d_{k+1}^2 - d_k^2}{R_k^2} \qquad (k = 1, \ldots, N), \tag{2.13}$$

normalized so $\sum_k w_k^2 = 1$. Non-negativity is automatic since $d_{k+1} > d_k$ for all ordered scales, and $L$ enters only through the last weight $w_N^2 \propto [(L - R_1)^2 - (R_N - R_1)^2]/R_N^2$ via the trailing-edge term $d_{N+1}$.

*Tightness of the affine envelope.* Allocation (2.13) simultaneously maximizes the slope $\lambda(\nu)$ of Corollary 2.1: substituting $S_i = c^2 d_i^2$ into (2.11) saturates all $N$ ratios at the common value $c/48$, certifying joint optimality of this allocation against the concave max-min in $(w_k^2)_k$. Equivalently, under (2.13) the affine envelope of Corollary 2.1 is *tight at every step corner* of Proposition 2.1(b): the line $\lambda(\nu)(\delta - R_1)$ touches the lower envelope of the step at $\delta = 3R_i^-$ for every $i$ (and at $\delta = L$). The weights derived intrinsically from the sharp step certificate thus also realize the largest affine relaxation usable in the classification rate work of Section 3. This matches the closed-form choice of Mitra & Virk (2024) (Theorem 5.1), and we adopt it throughout. Scale *location* optimization (which changes $\ell$) is left to future work.

# 3  Classification Guarantees

This section develops the classification theory for the embedded features of Section 2. We first establish the key quantities—class-mean separation $\Delta$ and embedding radius $R$—then prove an excess-risk upper bound $O(kR/(\Delta\sqrt{m_{\min}}))$ (Section 3.1) with a matching Le Cam sample-starved lower bound (Section 3.2) and a ranking-consistent descriptor-selection criterion $\Delta/\sqrt{\ell}$ (Section 4); the per-prediction certificate then follows in Section 5.

**Evidence convention.**  Following a reviewer suggestion, each named result carries an evidence tag separating what is established by proof from what is observed on benchmarks: [Proved] marks a claim proved in full generality under its stated hypotheses; [Heuristic] a closed-form rule or approximation justified by a non-rigorous argument; [Empirical] a statement supported only by the experiments of Section 6. Hypotheses that hold empirically but are not guaranteed (e.g. $\nu$-coherence, low stable rank) are flagged as such at each use.

Let $(A, Y)$ be a random pair with joint distribution $\mathcal{P}$ on $\mathcal{D} \times [k]$, where $A$ is a finite persistence diagram and $Y \in [k] := \{1, \ldots, k\}$ is the class label. We associate to $\mathcal{P}$ two population quantities: the *class-conditional embedding mean* $\mu_c := \mathbb{E}[\Phi(A) \mid Y = c] \in \mathbb{R}^\ell$ and the *class-mean separation*

$$\Delta := \min_{c \neq c'} \|\mu_c - \mu_{c'}\|_{\ell^2}, \tag{3.1}$$

together with the *embedding radius* $R := \sup_A \|\Phi(A; \nu)\|_{\ell^2}$ (the supremum is taken over the support of $\mathcal{P}$ on $\mathcal{D}$, which is bounded by the top-$N_{\max}$ filter of Section 2). Note that $\Delta$ is a property of the embedding and the data distribution, not of the worst-case bottleneck distance: $\Delta > 0$ is possible even when some cross-class diagram pairs are bottleneck-close, because the embedding aggregates information from all diagram points into class means.

**Notation.**  Throughout Sections 3–5, $R$ denotes the embedding radius and $m$ denotes the training-sample size (Mohri convention); the single-scale radii of Section 2 are always written with a subscript as $R_1, \ldots, R_N$, and the cardinality of an individual diagram (the $n$ in $\mathcal{D}_n$ of Section 2) is bounded by $N_{\max}$, so there is no collision. The symbol $\delta$ is reserved for the confidence parameter $1 - \delta$ in concentration bounds; the geometric bottleneck separation between class supports is written $\delta_{cc'} := d_{\mathcal{B}}(\operatorname{supp} \mathcal{P}_c, \operatorname{supp} \mathcal{P}_{c'})$ for the pair $(c, c')$, with $\delta_* := \min_{c \neq c'} \delta_{cc'}$.

Given the population quantities above, we observe $m$ i.i.d. training samples $\{(A_i, y_i)\}_{i=1}^m \sim \mathcal{P}$, with per-class counts $m_c = |\{i : y_i = c\}|$, and form the empirical class means $\hat{\mu}_c = m_c^{-1} \sum_{y_i = c} \Phi(A_i)$. Because $\Phi$ is linear in the empirical diagram measure $\mu_A = \sum_{a \in A} \delta_a$, each $\hat{\mu}_c$ is an ordinary sample average of i.i.d. bounded $\mathbb{R}^\ell$-vectors, so standard concentration inequalities (CLT, Hoeffding, McDiarmid) apply directly; a full treatment including Berry–Esseen rates and functional CLTs is beyond the present paper's scope.

The embedding's distortion slope $\lambda(\nu)$ of Corollary 2.1 enters the classification theory through the following bridge, which ties the data-dependent separation $\Delta$ back to the geometry of the underlying persistence diagrams. It is the bridge through which $\lambda(\nu)$—inherited from the Mitra–Virk landmark construction's compact-support hats, multiplicity-4 lattice cover, and multi-scale aggregation—enters the downstream classification bounds; every later use of $\lambda(\nu)$ in this section ultimately invokes it. This is what makes the excess-risk rate and the certificate of Section 5 more than generic statements about an abstract bounded embedding.

**Proposition 3.1** ($\lambda$-separation bridge)**.**  *Let $D_c := \sup_{A:Y=c} \|\Phi(A) - \mu_c\|_{\ell^2}$ denote the within-class radius for class $c$. Suppose $\delta_* \geq 3R_1$ and every cross-class pair $(A, B)$ with $d_{\mathcal{B}}(A, B) \geq \delta_*$ is $\nu$-coherent (Definition 2.1); this hypothesis is mild empirically (Remark 2.2). Then*

$$\Delta \geq \lambda(\nu)(\delta_* - R_1) - 2\max_c D_c. \tag{3.2}$$

*Proof.* For any cross-class pair $A \in \operatorname{supp} \mathcal{P}_c$, $B \in \operatorname{supp} \mathcal{P}_{c'}$, the triangle inequality gives $\|\Phi(A) - \Phi(B)\|_{\ell^2} \leq \|\mu_c - \mu_{c'}\|_{\ell^2} + D_c + D_{c'}$. Applying Corollary 2.1 under $\nu$-coherence at separation $d_{\mathcal{B}}(A, B) \geq \delta_{cc'} \geq \delta_*$ gives

$\|\Phi(A) - \Phi(B)\|_{\ell^2} \geq \rho_-(d_{\mathcal{B}}(A, B); \nu) \geq \rho_-(\delta_*; \nu) = \lambda(\nu)(\delta_* - R_1)$ (monotonicity of $\rho_-$ in its first argument). Chaining and taking the minimum over $c \neq c'$, with $D_c + D_{c'} \leq 2 \max_c D_c$, yields (3.2). $\qquad\square$

**Remark 3.1** (Step-form sharpening). *Substituting Proposition 2.1(b)'s step certificate $\|\Phi(A) - \Phi(B)\|_{\ell^2} \geq \frac{1}{16}\sqrt{\sum_{k:\, 3R_k \leq \delta_*} w_k^2 R_k^2}$ for Corollary 2.1's affine form in the proof above gives a tighter $\Delta$ bound, $\Delta \geq \frac{1}{16}\sqrt{\sum_{k:\, 3R_k \leq \delta_*} w_k^2 R_k^2} - 2\max_c D_c$, by a factor of up to $\sim 3$ when multiple scales activate at $\delta_*$. We retain the affine form in (3.2) for clean substitution into Corollary 3.1; the step form is the sharper reading when the per-scale decomposition is of independent interest.*

Proposition 3.1 has three consequences. First, it propagates $\lambda(\nu)$ into the classification rate (Corollary 3.1 below). Second, it upgrades the interpretation of the Section 5 certificate: when the empirical condition $r_m < \frac{1}{2}\Delta$ fires, the proposition translates this back into an inequality on $\delta_*$—certifying that the class-conditional diagram distributions are genuinely bottleneck-separated, not merely that the embedding has concentrated empirical means. Third, it lifts the coarse-embedding property of Proposition 2.1 from points to first moments of class-conditional distributions: bottleneck-separated class supports remain Euclidean-separated in the mean, modulo the within-class spread $2D_{\max}$. A persistence vectorization without an explicit lower distortion bound (e.g., persistence images or landscapes) has no analogue of Proposition 3.1, and its $\Delta$ cannot be back-translated to bottleneck-level data geometry.

### 3.1 Classification Error Bound

We train a linear SVM $h$ on the embedded training data $\{(\Phi(A_i), y_i)\}_{i=1}^m$ and measure its quality by the *generalization* 0-1 *risk* $\mathcal{R}(h) := \mathbb{P}(h(A) \neq Y)$. For a margin parameter $\rho > 0$, the *empirical $\rho$-margin loss* $\widehat{\mathcal{R}}_\rho(h)$ is the fraction of training points whose signed margin under $h$ falls below $\rho$ (Mohri et al., 2018, Sec. 5.4); for our multiclass $h$, $\widehat{\mathcal{R}}_\rho$ is aggregated across the binary OvO sub-problems as made precise in the proof of Theorem 3.1.

**Theorem 3.1** (Classification error bound [Proved]). *Let $\{(A_i, y_i)\}_{i=1}^m$ be $m$ i.i.d. training samples from a distribution on finite persistence diagrams, with $k$ classes and $\Delta > 0$. Assume additionally $m_{\min} \geq 128R^2 \log(4k/\delta)/\Delta^2$, where $m_{\min} := \min_c m_c$ is the smallest per-class sample count (so that empirical class means concentrate at a scale below $\Delta/4$). Set $\rho := \Delta/4$. Then with probability $\geq 1 - \delta$, the linear SVM classifier $h$, trained via one-vs-one reduction with majority voting, satisfies*

$$\mathcal{R}(h) \leq \widehat{\mathcal{R}}_\rho(h) + \frac{8(k-1)R}{\Delta\sqrt{m_{\min}}} + O\left(\sqrt{\frac{\log(k/\delta)}{m_{\min}}}\right). \tag{3.3}$$

For balanced classes $m_c \asymp m/k$ the rate term is $O(k^{3/2}R/(\Delta\sqrt{m}))$ in the total sample $m$; the $\sqrt{k}$ overhead is the price of the OvO reduction, since each binary sub-problem trains on only $\Theta(m/k)$ samples.

*Proof.* For each unordered pair $\{c, c'\}$, the population class means are separated by margin $\gamma_{cc'} := \frac{1}{2}\|\mu_c - \mu_{c'}\| \geq \frac{1}{2}\Delta = 2\rho$.

Conditional on the per-class counts $\{m_c\}$, the centered vectors $\{\Phi(A_i) - \mu_c : Y_i = c\}$ are i.i.d. (since $\Phi$ is deterministic and centering by the constant $\mu_c$ preserves independence) with $\|\Phi(A_i) - \mu_c\| \leq 2R$ by the triangle inequality (using $\|\Phi(A_i)\| \leq R$ from the support bound and $\|\mu_c\| \leq R$ by Jensen). Pinelis's Hilbert-space Hoeffding inequality (Lemma B.1) and a union bound over the $k$ classes yield, with probability $\geq 1 - \delta/2$,

$$\varepsilon_m := \max_c \|\hat{\mu}_c - \mu_c\| \leq 2R\sqrt{\frac{2\log(4k/\delta)}{m_{\min}}}.$$

The sample-size hypothesis $m_{\min} \geq 128R^2 \log(4k/\delta)/\Delta^2$ gives $\varepsilon_m \leq \rho$, so by the reverse triangle inequality the empirical pairwise margin $\hat{\gamma}_{cc'} := \frac{1}{2}\|\hat{\mu}_c - \hat{\mu}_{c'}\| \geq \gamma_{cc'} - \varepsilon_m \geq \rho$ for every pair $c \neq c'$.

The OvO sub-problem between $c, c'$ trains on $m_c + m_{c'} \geq 2m_{\min}$ samples from the unit-norm linear hypothesis class $\mathcal{H} := \{x \mapsto w^\top x : \|w\| \leq 1\}$ with $\|x\| \leq R$. The margin-based generalization bound (Mohri et al., 2018, Cor. 5.11) at margin $\rho$ and confidence $\delta' := \delta/(2\binom{k}{2})$ yields, with probability $\geq 1 - \delta'$,

$$\mathcal{R}(h_{cc'}) \leq \widehat{\mathcal{R}}_\rho(h_{cc'}) + \frac{8R}{\Delta \sqrt{m_{\min}}} + O\left(\sqrt{\frac{\log(k/\delta)}{m_{\min}}}\right),$$

using $\log(2/\delta') = \log(\binom{k}{2}/\delta) = O(\log(k/\delta))$ and the conservative substitution $\sqrt{m_c + m_{c'}} \geq \sqrt{m_{\min}}$ (loose by at most $\sqrt{2}$).

A union bound over the $\binom{k}{2}$ OvO sub-problems at level $\delta/2$, combined with the $\delta/2$ budget for the class-mean concentration step, gives total coverage $\geq 1 - \delta$. The OvO majority-vote rule errs at $y = c$ only if some pairwise classifier $h_{cc'}$ ($c' \neq c$) misclassifies, so by the union bound $\mathcal{R}(h) \leq (k-1)\max_{c \neq c'} \mathcal{R}(h_{cc'})$. Substituting the per-pair bound and defining $\widehat{\mathcal{R}}_\rho(h) := (k-1)\max_{c \neq c'} \widehat{\mathcal{R}}_\rho(h_{cc'})$ yields (3.3). The $(k-1)$-max aggregation is conservative: a pairwise classifier with zero $\rho$-margin loss contributes nothing, so when most pairs separate cleanly, the aggregate is correspondingly small. □

**Remark 3.2** (PLACE-specific tightening). *The bound* (3.3) *uses the worst-case embedding radius $R$. On PLACE, the multiplicity-4 lattice cover (Lemma 3.5 of Mitra & Virk, 2024; see (2.1)) provably confines $\Phi(A)$ to at most $4|A|N$ nonzero coordinates out of $\ell$. This support bound caps but does not by itself determine the class-conditional variance $\|\hat{\Sigma}_c\|_{\mathrm{op}}$: the latter is small relative to $R^2$ only when the active mass does not spread isotropically across the support, which on our benchmarks we observe rather than derive (e.g., $\sim 37\times$ slack $R^2/\|\hat{\Sigma}_c\|_{\mathrm{op}}$ on MUTAG, median over configurations; the empirical stable rank $\mathrm{tr}(\hat{\Sigma}_c)/\|\hat{\Sigma}_c\|_{\mathrm{op}} \leq 1.17$, Remark 5.2). The same sparsity ingredient drives the non-vacuous Pinelis–Bernstein certificate of Section 5 (Theorem 5.1, radius* (iii)), *which replaces the norm bound $\|\Phi(A_i) - \mu_c\| \leq 2R$ in the Pinelis step by the variance proxy $\sigma_c^2 = \mathrm{tr}(\Sigma_c) \approx \|\Sigma_c\|_{\mathrm{op}}$ (since the empirical stable rank is within a factor* $1.17$ *of* $1$ *on our benchmarks), tightening the sample-size requirement by $4R^2/\|\Sigma_c\|_{\mathrm{op}} \approx 146\times$ on MUTAG (median-over-configuration $R^2 \approx 44.9$, $\|\hat{\Sigma}_c\|_{\mathrm{op}} \approx 1.23$).*

**Remark 3.3** (Sample-size hypothesis at experimental scales). *The hypothesis $m_{\min} \geq 128R^2\log(4k/\delta)/\Delta^2$ is the standard Rademacher–margin sufficient threshold; it is not met at the per-class sample sizes of the graph benchmarks in Section 6 (e.g., MUTAG has $m_{\min} = 57$ against a worst-case threshold of order $10^3$, even after the $4\times$ variance-aware tightening of Remark 3.2). Theorem 3.1 should therefore be read as a* rate statement *pairing with the matching sample-starved lower bound of Theorem 3.2, not as an operational certificate at our $m$. The empirical accuracies reported in Section 6 are obtained in a moderate-sample regime that lies between the necessary threshold $m \asymp R/\Delta$ (Theorem 3.2) and the sufficient threshold $m \asymp R^2/\Delta^2$ (Theorem 3.1), where neither bound is tight and a Mammen–Tsybakov margin condition or an Assouad/Fano construction would be needed to close the gap (Remark 3.5).*

**Corollary 3.1** ($\lambda$-anchored classification rate). *Suppose $\delta_* \geq 3R_1$, every cross-class pair is $\nu$-coherent (Definition 2.1), and $\lambda(\nu)(\delta_* - R_1) > 2\max_c D_c$. Then under Theorem 3.1's sample-size hypothesis with $\Delta$ replaced by $\lambda(\nu)(\delta_* - R_1) - 2\max_c D_c$, the linear SVM classifier $h$ satisfies, with probability $\geq 1 - \delta$,*

$$\mathcal{R}(h) \leq \widehat{\mathcal{R}}_\rho(h) + \frac{8(k-1)R}{\left(\lambda(\nu)(\delta_* - R_1) - 2\max_c D_c\right)\sqrt{m_{\min}}} + O\left(\sqrt{\frac{\log(k/\delta)}{m_{\min}}}\right). \tag{3.4}$$

*Proof.* Proposition 3.1 gives $\Delta \geq \lambda(\nu)(\delta_* - R_1) - 2\max_c D_c > 0$; substituting this lower bound on $\Delta$ into Theorem 3.1 yields (3.4). □

**Remark 3.4** (Empirical scope of Corollary 3.1). *The $\nu$-coherence hypothesis (Definition 2.1) on every cross-class pair holds on $\geq 99.7\%$ of pairs across the four chemical benchmarks (Table 7), and the $\rho_-$ certificate's conclusion holds on $100\%$ (Table 8). The corollary is therefore best read as a structural rate transferring the bottleneck-support separation $\delta_*$ to a classification rate via the $\lambda$-bridge of Proposition 3.1, underwritten in practice by $\nu$-coherence. The empirical rate reported in Section 6 follows from Theorem 3.1 directly, which depends only on $\Delta > 0$.*

Theorem 3.1 bounds the *excess risk* $\mathcal{E}(h) := \mathcal{R}(h) - R^*$ of the trained classifier, where the Bayes risk $R^* := \inf\{\mathbb{P}(f(A) \neq Y) \mid f : \mathcal{D} \to [k] \text{ measurable}\}$ is non-negative (so any bound on $\mathcal{R}(h)$ bounds $\mathcal{E}(h)$). The lower bound of Section 3.2 that follows is of a different kind—a two-point *testing / identifiability* bound—and pairs with Theorem 3.1 as a complementary sample-complexity statement: identifying two $\Delta$-separated populations already requires $m = \Omega(R/\Delta)$ samples, whereas $m \asymp R^2/\Delta^2$ suffices to drive the excess risk small.

## 3.2  A Matching Lower Bound, Consistency, and Linear Separability

The rate $R/(\Delta\sqrt{m_{\min}})$ of Theorem 3.1 is the standard Rademacher–margin rate; its sample-size hypothesis $m \gtrsim R^2/\Delta^2$ is sufficient for non-trivial accuracy. The two-point testing lower bound below (stated for $k = 2$, where $m_{\min} = m/2$ for balanced classes) shows that $m \gtrsim R/\Delta$ is *necessary*: below that scale two $\Delta$-separated diagram populations cannot be reliably distinguished, so no procedure depending on which population generated the data can succeed. The polynomial gap between the necessary $R/\Delta$ and sufficient $R^2/\Delta^2$ thresholds is the moderate-sample regime and would require an Assouad / Fano construction to close (Remark 3.5).

**Theorem 3.2** (Sample-starved two-point testing lower bound [Proved]). *Let $\mathcal{P}^{\mathrm{PD}}_{\Delta,R}$ denote the family of binary diagram laws $(Q_+, Q_-)$ on $\mathcal{D}$ whose pushforwards through the PLACE embedding $\Phi$ satisfy $\|\mathbb{E}_{Q_+}\Phi - \mathbb{E}_{Q_-}\Phi\| = \Delta$ and $\sup_{A\in\mathrm{supp}(Q_\pm)}\|\Phi(A)\| \leq R$. For a pair $(Q_+, Q_-)$ and a test $h : \mathcal{D}^m \to \{+, -\}$—deciding, from $m$ i.i.d. diagrams drawn from an unknown $Q_\theta$ ($\theta \in \{+, -\}$), which population generated them—write the two-point testing error*

$$\mathcal{E}_{\mathrm{test}}(h) := \max_{\theta\in\{+,-\}} \mathbb{P}_{A_1,\dots,A_m \overset{\mathrm{iid}}{\sim} Q_\theta}\big(h(A_1,\dots,A_m) \neq \theta\big).$$

*There is a constant $\beta \geq 0$ depending only on $\Phi$ (not on the embedding dimension $\ell$) and a threshold $\Delta_{\max} > 0$ such that for every separation $\Delta \in (0, \Delta_{\max}]$ there is a radius $R = R(\Delta)$ with $\Delta \leq 2R/3$ and a pair $(Q_+, Q_-) \in \mathcal{P}^{\mathrm{PD}}_{\Delta,R}$ for which*

$$\inf_{h:\mathcal{D}^m\to\{+,-\}} \mathcal{E}_{\mathrm{test}}(h) \geq \tfrac{1}{4} \qquad \text{whenever } m \leq c_0\, R/\Delta, \quad c_0 := \tfrac{1}{6(1+\beta)}. \tag{3.5}$$

*Consequently no test—and hence no procedure that identifies which of two $\Delta$-separated, $R$-bounded persistence-diagram populations generated a sample—can succeed from $m \lesssim R/\Delta$ diagrams, regardless of computational budget, model class, or embedding dimension. Because $R(\Delta)/\Delta \to \infty$ as $\Delta \to 0$, this identification requires $m = \Omega(R/\Delta)$ samples. (The bound is an identifiability/testing floor; it does not by itself assert a matching classification excess-risk rate—see Remark 3.5.)*

*Proof.* We exhibit a one-parameter sub-family of $\mathcal{P}^{\mathrm{PD}}_{\Delta,R}$ supported on single-pair diagrams whose pushforwards through $\Phi$ are 1-D uniform measures on $\mathbb{R}^\ell$; the minimax over $\mathcal{P}^{\mathrm{PD}}_{\Delta,R}$ is at least the minimax over this sub-family, which reduces to the dimension-one Hellinger calculation of Lemma B.2.

Fix a base point $p_0 \in \mathcal{D}_1$ and translate it along the birth axis $e_1 = (1, 0)$, defining the single-pair diagrams $A(t) := \{p_0 + t\, e_1\}$; only the birth coordinate moves, the death coordinate stays frozen. Each hat coordinate of (2.2),

$$\varphi_{R_k,p}(p_0 + t\, e_1) = \max\big\{\tfrac{3R_k}{2} - \|p_0 + t\, e_1 - p\|_\infty,\, 0\big\},$$

is a *piecewise-linear* function of the single scalar $t$ (the $\ell_\infty$ distance of a point moving along one axis is piecewise-linear in $t$, and $\max\{\cdot, 0\}$ adds at most one more breakpoint), with finitely many breakpoints—where a hat enters or leaves its support or crosses its $\ell_\infty$ ridge. Superimposing the breakpoints of all $\ell$ coordinates partitions $\mathbb{R}$ into finitely many intervals on each of which *every* coordinate is affine; fix $p_0$ and a half-width $B > 0$ so that $[-B, B]$ lies in the interior of one such interval. (Following review, $B$ is taken large enough to contain the *entire* support of the hard family constructed below, not merely a neighbourhood of $t = 0$.) On $[-B, B]$ every active hat—at *every* scale $R_k$ and landmark $p$ covering $p_0$, however many there are—is then affine, $\varphi_{R_k,p}(p_0 + t\, e_1) = c_{k,p} + \gamma_{k,p}\, t$, and every inactive coordinate is identically 0.

Because the multiscale embedding (2.3) is *linear* in the hat values, $\Phi$ composed with $t \mapsto A(t)$ is affine on the interval:

$$\Phi(A(t)) = c + t\, v, \qquad c := \big(w_k 2^{-3/2} c_{k,p}\big)_{(k,p)}, \quad v := \big(w_k 2^{-3/2}\gamma_{k,p}\big)_{(k,p)}, \tag{3.6}$$

with $v \neq 0$ (choose $p_0$ off the $\ell_\infty$ ridge of at least one covering hat, so some $\gamma_{k,p} \neq 0$). *It is this affinity—not the mere injectivity of $\Phi$—that places the family in $\mathcal{P}^{\mathrm{PD}}_{\Delta,R}$ with the prescribed constants*, the exact point raised in review. Because $t \mapsto \Phi(A(t))$ is affine, the embedded class mean moves linearly, so a separation in the $t$-means maps to an embedded-mean gap of *exactly* $\|v\|$ times it (tunable to $\Delta$), and the embedded norm $\|c + t\,v\|$ stays in an $\ell$-independent band along the segment (tunable to $R$). A nonlinear—even if injective—$\Phi$ would keep the points distinct yet distort both the mean gap and the norm, and neither class constraint could be certified. Several landmarks and scales are active here; the argument survives them precisely because each contributes its *own* affine coordinate and the block map (2.3) sums them linearly, so the composite $t \mapsto \Phi(A(t))$ is still a single affine map into the line $L := \{c + t\,v : t \in \mathbb{R}\}$.

Calibrate the two constants. With $s := \Delta/(2\|v\|)$ and a support half-width $w_t > 0$, set

$$Q_\pm := \mathrm{Unif}\big(\{A(t) : t \in [\pm s - w_t,\ \pm s + w_t]\}\big).$$

We require $s \leq w_t$ (so $\Delta = 2s\|v\| \leq 2w_t\|v\| = 2w$, the overlap Lemma B.2 needs) and $s + w_t \leq B$, so that *both* supports $[\pm s - w_t,\ \pm s + w_t]$ lie inside the single affine piece $[-B, B]$ on which (3.6) holds — the point raised in review: the affine identity $\Phi(A(t)) = c + tv$ must be valid on the actual support of $Q_\pm$, not merely near $t = 0$. Since $s = \Delta/(2\|v\|)$, this caps the realizable separation at $\Delta \leq 2\|v\|(B - w_t)$ (Remark 3.5). By (3.6) the pushforwards $\Phi_* Q_\pm$ are uniform on length-$2w_t\|v\|$ segments of $L$ centered at $c \pm s\,v$, so $\|\mathbb{E}\Phi_* Q_+ - \mathbb{E}\Phi_* Q_-\| = 2s\|v\| = \Delta$; setting $R := \sup_{|t| \leq s + w_t} \|c + t\,v\|$ gives $\Phi(\mathrm{supp}\, Q_\pm) \subset B(0, R)$, hence $(Q_+, Q_-) \in \mathcal{P}^{\mathrm{PD}}_{\Delta,R}$ by construction. The restriction $t \mapsto \Phi(A(t)) = c + t\,v$ is injective (affine with $v \neq 0$)—the *only* step that invokes injectivity—so the data-processing identity gives $H^2(Q_+^{\otimes m}, Q_-^{\otimes m}) = H^2((\Phi_* Q_+)^{\otimes m}, (\Phi_* Q_-)^{\otimes m})$.

The two pushforwards are one-dimensional uniform laws on $L$, translates of each other by $\Delta$ along $u := v/\|v\|$ with common half-width $w := w_t\|v\|$; Lemma B.2 at dimension one gives $H^2(\Phi_* Q_+, \Phi_* Q_-) \leq (\Delta/2)/w$ once $\Delta \leq 2w$. Writing $\beta := \|c\|/w$ for the $\ell$-independent geometry ratio of (3.6), the triangle inequality gives $R \leq \|c\| + s\|v\| + w = \|c\| + \Delta/2 + w$, so $w \geq (R - \Delta/2)/(1 + \beta) \geq \frac{2}{3}R/(1 + \beta)$ under $\Delta \leq 2R/3$ (the overlap $\Delta \leq 2w$ that Lemma B.2 needs is secured directly by the calibration choice $w_t \geq s$ above). Combining Le Cam's two-point bound (Tsybakov, 2009, Ch. 2.2, 2.4) with Hellinger tensorization $H^2(P_+^{\otimes m}, P_-^{\otimes m}) \leq m\, H^2(P_+, P_-)$ and TV $\leq \sqrt{2H^2}$ yields $\mathrm{TV}(Q_+^{\otimes m}, Q_-^{\otimes m}) \leq \sqrt{m\Delta/w} \leq \sqrt{\frac{3}{2}(1 + \beta)\, m\Delta/R}$. With $c_0 := \frac{1}{6(1+\beta)}$ we obtain $\mathrm{TV}(Q_+^{\otimes m}, Q_-^{\otimes m}) \leq 1/2$ for every $m \leq c_0 R/\Delta$. Le Cam's two-point testing bound (Tsybakov, 2009, Ch. 2.4), $\inf_h \max_\theta \mathbb{P}_{Q_\theta}(h \neq \theta) \geq \frac{1}{2}\big(1 - \mathrm{TV}(Q_+^{\otimes m}, Q_-^{\otimes m})\big)$, then gives $\mathcal{E}_{\mathrm{test}}(h) \geq \frac{1}{2} \cdot \frac{1}{2} = \frac{1}{4}$, which is (3.5). (No excess-risk factor enters: the quantity bounded is the two-point testing error, not a classification risk.) Since $\beta$ depends only on the fixed embedding geometry and not on the ambient dimension $\ell$, the constant $c_0$—and hence the whole bound—is dimension-free. $\qquad\square$

**Remark 3.5** (Scope of the lower bound). *The hard family is PD-realizable: single-pair diagrams displaced along the birth axis within one Mitra–Virk hat wedge, with classifiers acting on raw diagrams (not feature vectors). We use the MV hat-wedge geometry, rather than a PLACE-specific failure mode such as the cancellation construction of Remark 2.1, because Le Cam requires a one-parameter family of statistically close distributions in $\Phi$-space, and the displacement-along-birth-axis parameterization supplies one directly; cancellation produces single diagram pairs with small $\|\Phi(A) - \Phi(B)\|$ at fixed $d_\mathcal{B}(A, B)$, which lower-bounds distortion (the failure mode of $\nu$-coherence) rather than sample complexity. Tightening to a matching $\Omega(R/(\Delta\sqrt{m}))$ rate would similarly need PD-aware constructions—e.g., Assouad/Fano over a $d_\mathcal{B}$-packing of diagrams, exploiting the bottleneck-to-$\Phi$ distortion bound—rather than abstract sub-Gaussian packings. Theorem 3.1 delivers an upper rate of $O(R/(\Delta\sqrt{m}))$ for all $m$. Theorem 3.2 is a two-point testing (identifiability) lower bound. For the fixed embedding it exhibits a one-parameter family $(Q_+^\Delta, Q_-^\Delta) \in \mathcal{P}^{\mathrm{PD}}_{\Delta,R(\Delta)}$ ($\Delta \in (0, \Delta_{\max}]$), with $R$ tied to $\Delta$ by the construction rather than freely prescribable and $c_0 = \frac{1}{6(1+\beta)}$ geometry- (not universally-) valued; on it no test identifies the generating population from $m \lesssim R/\Delta$ diagrams (error $\geq \frac{1}{4}$). This is an identifiability floor, not a classification excess-risk rate: turning it into an excess-risk bound would require a labeled two-point family whose Bayes rules differ, and that route yields only the margin-scaled $\Omega(\Delta/R)$ excess-risk floor—a strictly weaker, different statement—so we state the result as the sample-complexity threshold it rigorously supports. Beyond the sample-starved regime the two-point Le Cam construction yields no information: for $m \gtrsim R/\Delta$, the specific two-point hypothesis pair used here drives $\mathrm{TV}(P_+^{\otimes m}, P_-^{\otimes m})$ to 1, making*

*the lower bound argument vacuous for that pair. A tighter lower bound in this regime would require: (i) an Assouad/Fano construction over $\Theta(\sqrt{m})$-spaced hypotheses (Tsybakov, 2009, Ch. 2.6–2.7) would tighten the lower bound to a matching $\Omega(R/(\Delta\sqrt{m}))$ rate across all $m$; (ii) a Mammen–Tsybakov margin condition would instead tighten the upper bound to a faster $O(1/m)$ rate, dropping the sufficient threshold from $R^2/\Delta^2$ to $R/\Delta$ in line with Theorem 3.2's necessary threshold. We leave both to future work. The practical takeaway is the sample-starved threshold $m = \Omega(R/\Delta)$: below it the two $\Delta$-separated populations cannot be told apart, so no procedure that depends on which one generated the data can succeed.*

The classification rate of Theorem 3.1 depends on the population separation $\Delta$; for that rate to be operationally useful, $\Delta$ must be estimable from training data. The next proposition gives the concentration of the empirical estimator $\hat{\Delta}$, validating its use as a plug-in for $\Delta$ in the closed-form selection statistic $\hat{\Delta}/\sqrt{\ell}$ of Section 4.

**Proposition 3.2** (Consistency of $\hat{\Delta}$). *With $\hat{\mu}_c$ as in Section 3 and the empirical class-mean separation $\hat{\Delta} := \min_{c \neq c'} \|\hat{\mu}_c - \hat{\mu}_{c'}\|$, for every $\varepsilon > 0$,*

$$\mathbb{P}\Big(|\hat{\Delta} - \Delta| > \varepsilon\Big) \leq 2k \exp\Big(-\frac{\varepsilon^2 \, m_{\min}}{32R^2}\Big).$$

*In particular, $|\hat{\Delta} - \Delta| = O_P(R/\sqrt{m_{\min}})$.*

*Proof.* By the reverse triangle inequality $|\hat{\Delta} - \Delta| \leq 2 \max_c \|\hat{\mu}_c - \mu_c\|$. Pinelis's Hilbert-space Hoeffding inequality (Lemma B.1) with norm bound $2R$ and a union bound over the $k$ classes give the result. $\qquad\square$

While $\Delta > 0$ alone delivers the $1/\sqrt{m}$ excess-risk rate of Theorem 3.1, a stronger structural condition—small within-class spread relative to $\Delta$—yields population-level perfect classification with an explicit geometric margin. This hypothesis underlies the certificate-firing analysis of Section 5.

**Proposition 3.3** (Linear separability). *Define the within-class radius $D_c := \sup_{A:Y=c} \|\Phi(A) - \mu_c\|$ and let $D_{\max} := \max_c D_c$. If $D_{\max} < \Delta/2$, then the nearest-centroid classifier in $\mathbb{R}^\ell$ achieves zero error with geometric margin $\geq \Delta/2 - D_{\max} > 0$.*

*Proof.* For $A$ from class $c$ and any $c' \neq c$: $\|\Phi(A) - \mu_c\| \leq D_c \leq D_{\max}$ by definition of $D_c$, and the reverse triangle inequality gives $\|\Phi(A) - \mu_{c'}\| \geq \|\mu_c - \mu_{c'}\| - \|\Phi(A) - \mu_c\| \geq \Delta - D_{\max}$. Subtracting,

$$\|\Phi(A) - \mu_{c'}\| - \|\Phi(A) - \mu_c\| \geq \Delta - 2D_{\max} > 0,$$

so $\Phi(A)$ is strictly closer to $\mu_c$ than to any other class mean (zero-error classification) and the half-gap $\frac{1}{2}(\|\Phi(A) - \mu_{c'}\| - \|\Phi(A) - \mu_c\|) \geq \Delta/2 - D_{\max}$ gives the geometric margin. $\qquad\square$

Although the proof of Proposition 3.3 is a generic $\mathbb{R}^\ell$ geometric fact, whether the hypothesis $D_{\max} < \Delta/2$ can plausibly hold on a given embedding depends on structural properties of that embedding. On PLACE, the same compact-support / multiplicity-4 lattice cover (Lemma 3.5 of Mitra & Virk, 2024; see also Remark 5.2)—each diagram activates at most $4\,|A|\,N$ landmarks out of $\ell$—keeps $\|\Phi(A) - \mu_c\|$ effectively confined to the low-rank subspace of active coordinates, so $D_c$ remains small relative to $\Delta$ when the descriptor exposes a structural gap between classes. Persistence images and landscapes, whose Gaussian-blurred or order-statistic coordinates are weakly active on every diagram, spread within-class variation across all $\ell$ directions and tend to produce $D_c$ comparable to or larger than $\Delta$, often violating the hypothesis even when the classes are bottleneck-separated. This is the same sparsity ingredient that makes Theorem 5.1's certificate non-vacuous (Remark 5.2), instantiated at the level of the within-class-radius hypothesis instead of the operator-norm certificate condition.

Both Proposition 3.2 (consistency) and Proposition 3.3 (separability) treat $\Delta$ as a fixed property of a given descriptor. Section 4 addresses how to *choose* the descriptor that maximizes $\Delta$ from a pool of candidates.

# 4 Descriptor Selection

A persistence-based classifier's accuracy depends as much on the choice of filtration and vectorization—collectively, the *descriptor*—as on the downstream estimator: descriptor swaps on the same dataset move accuracy by 5–15 percentage points (Section 6). We use *descriptor* broadly: a single filtration on one homology dimension (e.g., the degree filtration on $H_0$ for graphs, or alpha complex on $H_1$ for point clouds), or a *pool* of several filtrations and/or homology dimensions (e.g., `deg+HKS`$_{10}$ on $H_{0+1}$ in Section 6, where the constituent persistence diagrams are merged into one before embedding). We formalize *descriptor selection* as a meta-problem: given a finite pool $\mathcal{F}$ of candidate descriptors, choose one from training labels alone, with no held-out validation. We develop two complementary rules—a recommended Mahalanobis-margin selector and a simpler closed-form surrogate $\hat{\Delta}/\sqrt{\ell}$ admitting a selection-consistency theorem—and characterize the regimes in which each is principled.

For each $f \in \mathcal{F}$, the descriptor produces an embedding $\Phi^f : \mathcal{D} \to \mathbb{R}^{\ell_f}$ with radius $R_f := \sup_{A \in \mathcal{D}} \|\Phi^f(A)\|$, class means $\mu_c^f := \mathbb{E}[\Phi^f(A) \mid Y = c]$, separation $\Delta_f := \min_{c \neq c'} \|\mu_c^f - \mu_{c'}^f\|_{\ell^2}$, and pooled within-class covariance $\Sigma^f := \frac{1}{k} \sum_c \text{Cov}(\Phi^f(A) \mid Y = c)$. The empirical separation is $\hat{\Delta}_f := \min_{c \neq c'} \|\hat{\mu}_c^f - \hat{\mu}_{c'}^f\|$, with $\hat{\mu}_c^f$ the per-class sample mean, and $h_f$ denotes the linear-SVM classifier trained on $\{(\Phi^f(A_i), y_i)\}_{i=1}^m$.

## 4.1 Mahalanobis margin

The *Mahalanobis margin* between class means is

$$\rho_{\text{Mah}}^f := \min_{c \neq c'} \sqrt{(\mu_c^f - \mu_{c'}^f)^\top (\Sigma^f)^{-1} (\mu_c^f - \mu_{c'}^f)}, \tag{4.1}$$

a pairwise extension of the two-class Fisher discriminant ratio to $k$ classes, taking the minimum over class pairs; for $k = 2$ this coincides with the standard Fisher ratio, while for $k > 2$ it differs from the multiclass Fisher ratio $\text{tr}(S_W^{-1} S_B)$ but retains the same covariance-normalized separation interpretation. Equation (4.1) is the LDA Bayes margin under the homoscedasticity assumption that within-class covariances are equal across classes ($\Sigma_c^f \approx \Sigma^f$ for all $c$); when class-conditional covariances differ substantially, (4.1) approximates rather than equals the true LDA Bayes margin, and the Ledoit–Wolf shrinkage in the empirical counterpart $\hat{\rho}_{\text{Mah}}^f$ partially mitigates this by regularizing toward a common pooled covariance. Throughout we assume $\Sigma^f$ is positive definite, so $(\Sigma^f)^{-1}$ is well-defined; in the high-dimensional regime $\ell_f > m$ where $\Sigma^f$ may be singular, the population quantity (4.1) is understood via the Moore–Penrose pseudoinverse, and the empirical counterpart uses the Ledoit–Wolf shrunk estimator $\hat{\Sigma}_{\text{LW}}^f$, which is positive definite by construction.

The implementation uses the all-class pooled covariance $\Sigma^f = \frac{1}{k} \sum_c \Sigma_c^f$ throughout. For $k = 2$ this coincides with the pairwise alternative $\frac{1}{2}(\Sigma_c^f + \Sigma_{c'}^f)$; for $k > 2$ they differ in general and the all-class pool is used. Ledoit–Wolf shrinkage is the appropriate regularization here because PLACE operates in the regime $\ell_f \asymp m$ or $\ell_f > m$ (large grids, moderate sample sizes), where the sample covariance is ill-conditioned; Ledoit–Wolf provides a closed-form optimal linear shrinkage toward a scaled identity that minimizes the Frobenius estimation error under the Marchenko–Pastur asymptotics, without requiring cross-validation or a held-out tuning set. The empirical counterpart $\hat{\rho}_{\text{Mah}}^f$ replaces $\Sigma^f$ by $\hat{\Sigma}_{\text{LW}}^f$ in (4.1). We propose the Mahalanobis selector

$$\hat{f}_{\text{Mah}} := \arg\max_{f \in \mathcal{F}} \hat{\rho}_{\text{Mah}}^f \tag{4.2}$$

as the recommended descriptor-selection rule. Empirically (Section 6.3, Table 11), $\hat{\rho}_{\text{Mah}}$ rank-correlates with linear-SVM accuracy at Spearman $\rho \in [-0.24, +0.89]$ across 11 benchmarks (mean +0.56, positive on 10 of 11, with PTC the lone outlier), ranking the accuracy-winning descriptor in the top seven on seven of eleven. A formal consistency theorem for $\hat{\rho}_{\text{Mah}}$ requires concentration of $\hat{\Sigma}_{\text{LW}}^f$ and is beyond the present paper's scope; below we develop the consistency theory for the simpler isotropic surrogate $\hat{\eta} := \hat{\Delta}/\sqrt{\ell}$.

## 4.2 Isotropic surrogate $\eta = \Delta/\sqrt{\ell}$

Theorem 3.1's rate $R_f/\Delta_f$ requires controlling $R_f$. The coordinate-wise hat-function bound $|\Phi_p(A)| \leq w_k \cdot 2^{-3/2} \cdot N_{\max} \cdot 3R_k/2$ combines the hat peak $\varphi_{R_k,p} \leq 3R_k/2$, the block prefactor $w_k \cdot 2^{-3/2}$ of (2.3), and

the top-$N_{\max}$ persistence filter that caps $|A|$ (all from Section 2); the resulting per-coordinate envelope is independent of $\ell_f$, so summing $|\Phi_p(A)|^2$ over the $\ell_f$ coordinates gives the $\sqrt{\ell_f}$-rate envelope

$$R_f \ \leq \ B_f \sqrt{\ell_f}, \qquad B_f := \max_k w_k \cdot 2^{-3/2} \cdot N_{\max} \cdot 3R_k/2, \tag{4.3}$$

where $\{w_k\}, \{R_k\}, N_{\max}$ are descriptor $f$'s scale weights, scale radii, and top-$N$ persistence-filter cap; we suppress the $f$-superscript on these for readability, but $B_f$ depends on $f$ through all three. Substituting (4.3) into Theorem 3.1 gives a closed-form excess-risk bound parameterized by the analytic surrogate $\eta_f := \Delta_f / \sqrt{\ell_f}$:

$$\mathcal{E}(h_f) \ \leq \ \frac{8(k-1)\, B_f}{\eta_f \, \sqrt{m_{\min}}} \ + \ O\big(\sqrt{\log(k/\delta)/m_{\min}}\big).$$

On pools with roughly uniform $B_f$, ranking by $\eta_f$ minimizes this relaxed bound, providing a fully analytic selection rule that requires no covariance estimation. Define the bound-optimal descriptor and its empirical counterpart

$$f^* \ := \ \arg\max_{f\in\mathcal{F}} \eta_f, \qquad \hat{f} \ := \ \arg\max_{f\in\mathcal{F}} \hat\eta_f \quad \text{with} \quad \hat\eta_f := \hat\Delta_f / \sqrt{\ell_f}.$$

For each $f$, let $\sigma_f^2 := \|\Sigma^f\|_{\mathrm{op}}$ be the largest within-class variance in any direction (the largest eigenvalue of $\Sigma^f$). The smallest eigenvalue of $(\Sigma^f)^{-1}$ is then $\sigma_f^{-2}$, so $v^\top (\Sigma^f)^{-1} v \geq \|v\|^2/\sigma_f^2$ for every $v$; specializing to $v = \mu_c^f - \mu_{c'}^f$ and minimizing over class pairs,

$$\rho_{\mathrm{Mah}}^f \ \geq \ \frac{\Delta_f}{\sigma_f} \ = \ \frac{\sqrt{\ell_f}}{\sigma_f}\, \eta_f.$$

The alignment inequality lower-bounds the Mahalanobis margin by a descriptor-dependent multiple of the isotropic surrogate, and is useful for ranking only if the factor $\sqrt{\ell_f}/\sigma_f$ is approximately constant across $f \in \mathcal{F}$—requiring both $\ell_f$ and $\sigma_f = \|\Sigma_f\|_{\mathrm{op}}^{1/2}$ to be roughly constant across the pool, the *structural homogeneity* condition. Under homogeneity, $\sqrt{\ell_f}/\sigma_f \approx C$ for a global constant $C$ and the rankings induced by $\eta_f$ and $\rho_{\mathrm{Mah}}^f$ tend to agree. Even under homogeneity the alignment is only a lower bound on $\rho_{\mathrm{Mah}}^f$, not a proportionality; its informativeness depends on how tightly $\rho_{\mathrm{Mah}}^f$ tracks its lower bound across descriptors. If the slack varies substantially across $f \in \mathcal{F}$, descriptors with smaller $\eta_f$ may achieve larger $\rho_{\mathrm{Mah}}^f$ and the two statistics may rank differently. On heterogeneous pools, where $\ell_f$ or $\sigma_f$ varies several-fold across $f$, the factor $\sqrt{\ell_f}/\sigma_f$ is not constant; the equivalence of the two quantities breaks down and $\hat\rho_{\mathrm{Mah}}$ should be used directly. The formal consistency theory for $\hat\rho_{\mathrm{Mah}}$ requires operator-norm concentration of $\hat\Sigma_{\mathrm{LW}}^f$ and is an open direction; the selection consistency rate for the isotropic surrogate $\hat\eta$ is established in Proposition 4.1 and Corollary 4.1 below.

**Remark 4.1** (When $\eta$ misses what $\rho_{\mathrm{Mah}}$ catches). *The pointwise alignment $\rho_{\mathrm{Mah}}^f \geq (\sqrt{\ell_f}/\sigma_f)\,\eta_f$ is informative for ranking only when both $\sigma_f$ and $\ell_f$ are roughly constant across the pool. On heterogeneous descriptor pools—HKS at many timescales, node-label-aware combinations, large grids mixed with small, where $\ell_f$ varies several-fold—the $\sqrt{\ell_f}$ penalty in $\eta$ over-charges high-dimensional descriptors, while $\rho_{\mathrm{Mah}}$ recovers the right ranking (Section 6.3, Table 11).*

*As a pre-hoc diagnostic, one can inspect the variation of $\ell_f$ and the per-descriptor scale $\hat\sigma_f := \sqrt{\|\hat\Sigma^f\|_{\mathrm{op}}}$ across $f \in \mathcal{F}$. Since $\hat\Sigma_f$ has effective rank $O(N_{\max}N) \ll \ell_f$ by the multiplicity-4 lattice cover, $\|\hat\Sigma_f\|_{\mathrm{op}}$ can be computed without forming $\hat\Sigma_f$ explicitly: either via power iteration (Golub & Van Loan, 1996) on the centered data matrix at cost $O(m\ell_f)$ per iteration, or via a randomized SVD (Halko et al., 2011) at cost $O(m\ell_f r)$ for a rank-$r$ approximation. Both match the leading cost of the Ledoit–Wolf assembly already required for $\hat\rho_{\mathrm{Mah}}$ and add no asymptotic overhead to the descriptor-selection pipeline. When both are tightly concentrated, the multiplicative factor $\sqrt{\ell_f}/\sigma_f$ in the alignment is approximately constant and $\hat\eta$ ranks faithfully; when either spreads several-fold, defer to $\hat\rho_{\mathrm{Mah}}$. The diagnostic uses a covariance trace $\mathrm{tr}(\hat\Sigma^f)$ rather than the full inverse, so its cost is $O(\sum_f m\ell_f)$ across the pool—one $f$'s worth of $\hat\rho_{\mathrm{Mah}}$ assembly—rather than the $O(\sum_f \ell_f^3)$ of full Mahalanobis ranking.*

**Why $\eta$ is well-defined on PLACE.** The selection criterion $\hat{\eta}_f = \hat{\Delta}_f/\sqrt{\ell_f}$ is well-defined as a ranking statistic only when the embedding dimension $\ell_f$ is a principled function of the embedding construction, not a free hyperparameter. On PLACE, this is the case: $\ell_f = \sum_{k=1}^N |\mathbb{G}_{R_k}^+| = O(MN)$ is fixed analytically by the scales $R_1, \ldots, R_N \in (0, L]$ and the compact-support parameter $L$, so $\hat{\eta}_f$ depends only on the descriptor $f$. For persistence images or landscapes, in contrast, $\ell$ is a user-chosen grid resolution, and $\hat{\eta}$ can be driven arbitrarily small by increasing the grid density without changing the classification content of the embedding—so $\hat{\eta}$ is not a meaningful selection statistic on those vectorizations without an auxiliary convention for fixing $\ell$.

### 4.3   Selection consistency

We now make precise the claim from the surrogate subsection that $\hat{f} = \arg\max_f \hat{\eta}_f$ recovers the bound-optimal $f^* = \arg\max_f \eta_f$ with high probability when the candidates are well-separated.

**Proposition 4.1** (Selection consistency of $\Delta/\sqrt{\ell}$ [Proved]). *Assume a gap*

$$g := \eta_{f^*} - \max_{f \neq f^*} \eta_f > 0,$$

*set $\ell_{\min} := \min_{f \in \mathcal{F}} \ell_f$, $R_{\max} := \max_{f \in \mathcal{F}} R_f$, and $m_{\min} := \min_c m_c$. Then*

$$\mathbb{P}(\hat{f} = f^*) \geq 1 - 2k|\mathcal{F}| \exp\left(-\frac{g^2 \ell_{\min} m_{\min}}{128 R_{\max}^2}\right). \tag{4.4}$$

*In particular, $\hat{f} = f^*$ with probability $\geq 1 - \delta$ once $m_{\min} \geq 128 R_{\max}^2 \log(2k|\mathcal{F}|/\delta) / (g^2 \ell_{\min})$.*

*Proof.* For each $f \in \mathcal{F}$, $|\hat{\eta}_f - \eta_f| = |\hat{\Delta}_f - \Delta_f|/\sqrt{\ell_f}$, so $\{|\hat{\eta}_f - \eta_f| > t\} = \{|\hat{\Delta}_f - \Delta_f| > t\sqrt{\ell_f}\}$ for any $t > 0$. Applying Proposition 3.2 at $\varepsilon = t\sqrt{\ell_f}$ and using $\ell_f \geq \ell_{\min}$, $R_f \leq R_{\max}$ in the exponent,

$$\mathbb{P}(|\hat{\eta}_f - \eta_f| > t) \leq 2k \exp\left(-\frac{m_{\min} \ell_{\min} t^2}{32 R_{\max}^2}\right).$$

Take $t = g/2$ and apply a union bound over $|\mathcal{F}|$ descriptors. On the event $\mathcal{A} := \{|\hat{\eta}_f - \eta_f| < g/2$ for every $f \in \mathcal{F}\}$, which has probability $\geq 1 - 2k|\mathcal{F}| \exp(-g^2\ell_{\min}m_{\min}/(128R_{\max}^2))$, every $f \neq f^*$ satisfies

$$\hat{\eta}_f < \eta_f + \frac{g}{2} \leq (\eta_{f^*} - g) + \frac{g}{2} = \eta_{f^*} - \frac{g}{2} < \hat{\eta}_{f^*},$$

so $f^*$ is the *unique* maximizer and $\hat{f} = f^*$ on $\mathcal{A}$ (ties in the arg max, which occur only off $\mathcal{A}$, broken by lowest index), giving (4.4). □

The constant 128 inherits the 32 of Proposition 3.2, which uses Pinelis with the $L^2$ bound $4R^2$. Replacing it with the variance-aware form (cf. Remark 3.2) tightens $32 \to 8$ in Proposition 3.2, hence $128 \to 32$ in the sample-size hypothesis above—the same factor-of-4 improvement that PLACE's multiplicity-4 sparsity drives in the classification bound and in the certificate.

**Remark 4.2** (Operational scope of Proposition 4.1). *Two of the bound's inputs—the population gap $g$ and the embedding radius $R_{\max}$—are not directly observed at training time, but both admit training-side proxies. $R_{\max}$ is upper-bounded analytically by the envelope $R_f \leq B_f\sqrt{\ell_f}$ of (4.3), with $B_f$ a function of the embedding parameters only. The empirical gap $\hat{g} := \hat{\eta}_{\hat{f}} - \max_{f \neq \hat{f}} \hat{\eta}_f$ concentrates around $g$ at rate $O(R_{\max}/\sqrt{m_{\min}\ell_{\min}})$ via Proposition 3.2 applied to the top two $\hat{\eta}_f$ entries; substituting $\hat{g}$ for $g$ in the sample threshold adds an $O(1/\sqrt{m_{\min}\ell_{\min}})$ slack already present in the rate's order.*

*A second point of pessimism is the inverse dependence on $\ell_{\min}$: the sample requirement $m_{\min} \geq 128 R_{\max}^2 \log(\cdot)/(g^2\ell_{\min})$ is driven by the smallest $\ell_f$ in the pool, not by $\ell_{f^*}$. On heterogeneous pools where $\ell_f$ varies several-fold (e.g., the chemical pool of Section 6.3, where HKS-pair descriptors give $\ell_f \gtrsim 5{,}000$ while single-coordinate descriptors give $\ell_f \sim 50$), the bound becomes conservative. The empirical $\hat{\eta}$ rank-correlations on those pools (mean $\rho \in [-0.70, -0.05]$ across MUTAG/COX2/DHFR/NCI1/NCI109, Table 11) are consistent with this scope; a $\ell_f$-aware refinement requires a non-uniform union bound (per-descriptor confidence allocation) and is deferred.*

**Corollary 4.1** (Data-driven bound-optimal rate). *With probability $\geq 1 - \delta$ over the training sample, provided both (i) $m_{\min} \geq 128 R_{\max}^2 \log(4k|\mathcal{F}|/\delta)/(g^2 \ell_{\min})$ (from Proposition 4.1 at confidence $\delta/2$) and (ii) Theorem 3.1's sample-size hypothesis at confidence $\delta/2$ (i.e. $m_{\min} \geq 128 R_{f^*}^2 \log(8k/\delta)/\Delta_{f^*}^2$), the descriptor chosen by $\hat{\eta}$ attains the bound-optimal rate:*

$$\mathcal{R}(h_{\hat{f}}) \; \leq \; \widehat{\mathcal{R}}_\rho(h_{f^*}) \; + \; \frac{8(k-1)\, B_{f^*}\sqrt{\ell_{f^*}}}{\Delta_{f^*}\sqrt{m_{\min}}} \; + \; O\!\Big(\sqrt{\log(2k/\delta)/m_{\min}}\Big).$$

*The complexity term equals $8(k-1)\, B_{f^*}/(\eta_{f^*}\sqrt{m_{\min}})$ under $\eta_{f^*} = \Delta_{f^*}/\sqrt{\ell_{f^*}}$, i.e. the surrogate-relaxed bound of Section 4.2 instantiated at $f = f^*$.*

*Proof.* On the event $\{\hat{f} = f^*\}$ (probability $\geq 1 - \delta/2$ by Proposition 4.1 under hypothesis (i)), apply Theorem 3.1 to $h_{f^*}$ at confidence $\delta/2$ under hypothesis (ii), and take a union bound. On this same event, $h_{\hat{f}} = h_{f^*}$ and $\widehat{\mathcal{R}}_\rho(h_{\hat{f}}) = \widehat{\mathcal{R}}_\rho(h_{f^*})$, so the bound above is computable from the empirically chosen $\hat{f}$ even though it is stated in $f^*$-quantities. $\square$

Proposition 4.1 establishes that the empirical selector $\hat{f}$ recovers the bound-optimal descriptor $f^*$ with probability $\geq 1 - \delta$ once $m_{\min} \geq 128\, R_{\max}^2 \log(2k|\mathcal{F}|/\delta)/(g^2 \ell_{\min})$, a sample complexity growing logarithmically in $|\mathcal{F}|$ and inversely in $g^2$.

The proposition's reach is limited in two distinct ways. *(Gap.)* $g > 0$ requires a unique bound-optimum; a tied or near-tied pool makes the bound vacuous. *(Homogeneity.)* Even when $g > 0$, $\hat{\eta}$'s arg-max coincides with $\hat{\rho}_{\mathrm{Mah}}$'s only on structurally homogeneous pools (Remark 4.1); on the heterogeneous chemical pools of Section 6.3, the mean Spearman correlation of $\hat{\eta}$ with linear-SVM accuracy across 7 benchmarks is $-0.22$ (Table 11). Two complementary closed-form statistics recover alignment in the heterogeneous case: $\hat{\Delta}_f/\hat{R}_f$, the rate ratio of Theorem 3.1 computed without the envelope substitution, and the empirical Mahalanobis margin $\hat{\rho}_{\mathrm{Mah}}$ of (4.1) (recommended; see Section 4.1). We report all three in Section 6.3; agreement among them is a practitioner-level signal that the closed-form regime applies.

## 5 Certified Nearest-Centroid Classification

Classifiers typically expose a confidence score—a sigmoid probability, a distance to the decision boundary, a posterior estimate—that does not, on its own, tell the user whether a specific prediction will be correct. Conformal prediction (Vovk et al., 2005) attaches distribution-free coverage, but the guarantee applies to prediction *sets* rather than point predictions and requires a held-out calibration split that competes with training data for information. The embedding of Section 2 closes this gap for a specific classifier: bounded support gives $\|\Phi(A_i)\| \leq R$, so each empirical class mean $\hat{\mu}_c$ is a sample average of i.i.d. bounded $\mathbb{R}^\ell$-vectors, and $\|\hat{\mu}_c - \mu_c\|$ concentrates at rate $O(R/\sqrt{m_c})$ via Pinelis (Proposition 3.2). The nearest-centroid (NC) classifier is the natural target: its decision rule depends on the sample only through the $\hat{\mu}_c$, so whether the empirical and population rules agree on a given test input reduces to a single scalar check—is the input far enough from the population Voronoi boundary that sample fluctuations cannot move it across (Figure 4)? When $\Delta > 0$, this check has a particularly simple form: $r_m < \frac{1}{2}\Delta$ is a single training-time check; when satisfied, all predictions are certified at no per-test overhead beyond the nearest-centroid rule itself, and no calibration split is required.

The certificate is a diagnostic, not a competitor to SVM. Failure of $r_m < \frac{1}{2}\Delta$ is itself informative: the embedding's sample-mean concentration radius exceeds half the class gap, so PLACE's closed-form certificate admits no correctness guarantee at the given sample size. Other certification schemes—conformal prediction (Vovk et al., 2005), calibrated confidence, or margin-based bounds for different classifiers—may remain informative, but at the cost of a calibration split or looser set-valued guarantees. When the certificate fires (empirically, under the honest firing condition the variance-aware Pinelis–Bernstein form clears only the three benchmarks NCI1, NCI109, DD in Section 6; the non-asymptotic Pinelis form and the asymptotic Gaussian form fail everywhere at the $L^2$- and $\sqrt{\ell}$-driven slack respectively), it attests agreement of the empirical and population NC rules away from the Voronoi boundary—not per-prediction correctness (Remark 5.3)—with no per-test overhead and no calibration split.

Classify test diagrams by nearest centroid:

$$\hat{h} = \arg\min_c \|\Phi(A_{\text{test}}) - \hat{\mu}_c\|, \qquad \hat{\mu}_c = m_c^{-1} \sum_{y_i=c} \Phi(A_i),$$

where $\hat{\mu}_c$ is the empirical class mean from $m_c$ training diagrams. Let $m := m_{\min} = \min_c m_c$ (using the Section 3 notation, abbreviated $m$ here for brevity), and let $r_m$ denote a sample-mean-concentration radius satisfying $\mathbb{P}_{\text{train}}(\max_c \|\hat{\mu}_c - \mu_c\| \leq r_m) \geq 1 - \alpha$ (three explicit choices—a non-asymptotic Pinelis radius, an asymptotic Gaussian plug-in, and a non-asymptotic variance-aware Pinelis–Bernstein—are derived in Theorem 5.1). Here $\mathbb{P}_{\text{train}} = \mathcal{P}^{\otimes m}$ denotes the joint probability over training draws $\{(A_i, y_i)\}_{i=1}^m \sim \mathcal{P}^{\otimes m}$—probability over the randomness in the training sample, with the population distribution $\mathcal{P}$ and the test diagram $A$ held fixed. If $r_m < \frac{1}{2}\Delta$, every prediction is certified; otherwise the classifier abstains globally. The concentration radius $r_m$ shrinks as $O(m^{-1/2})$ (equation (5.2)), so abstention disappears once $m \geq m_c^*$ (equation (5.5)).

The global threshold $\Delta$ is conservative when classes differ in separation. Replacing $\Delta$ by the class-specific gap $\Delta_c := \min_{c' \neq c} \|\mu_c - \mu_{c'}\|_{\ell^2} \geq \Delta$, and $r_m^\star$ by the per-class radius (the formulas below with $m \to m_c$), yields a tighter certificate $r_m^{(c)} < \frac{1}{2}\Delta_c$ that fires when this holds for every class $c$ simultaneously.

Two concrete choices of the global concentration radius $r_m^\star$ enter the theorem below, both with an explicit Bonferroni split of $\alpha$ over $k$ classes:

(i) **Non-asymptotic (Pinelis).** $r_m^\star := 2R\sqrt{2\log(2k/\alpha)/m}$ with $m = m_{\min}$; valid for every $m \geq 1$ (equation (5.2) in the proof).

(ii) **Asymptotic (Gaussian plug-in).** $\tilde{r}_m := \max_c \sqrt{\|\hat{\Sigma}_c\|_{\text{op}} \cdot \chi^2_{\ell,\,\alpha/k}/m_c}$, where $\chi^2_{\ell,\,\alpha/k}$ is the $1 - \alpha/k$ quantile of the chi-squared distribution with $\ell$ degrees of freedom; this radius satisfies (5.3) approximately, with approximation error $O(\ell^{1/4}/\sqrt{m})$ from the multivariate Berry–Esseen theorem (Lemma B.3) and $O(R^{1/2}\|\Sigma_c\|_{\text{op}}^{1/4}(\log(\ell)/m)^{1/4}\sqrt{\ell/m})$ from covariance estimation via matrix Bernstein (Lemma B.4); both errors are $o(1)$ once $m_c \geq m^\dagger$ for every class $c$, with $m^\dagger = O(\sqrt{\ell})$ under bounded support $\|\Phi\| \leq R$. The bound is conservative when $\Sigma_c$ is low-rank, with conservatism governed by $\text{tr}(\Sigma_c)/(\ell \|\Sigma_c\|_{\text{op}})$, and is strictly tighter than the Pinelis radius $r_m$ when $\|\Sigma_c\|_{\text{op}} \cdot \ell \lesssim 8R^2 \log(2k/\alpha)$.

(iii) **Variance-aware (Pinelis–Bernstein).** $r_m^{\text{vP}} := \max_c\left[\sqrt{2\,\text{tr}(\hat{\Sigma}_c)\log(2k/\alpha)/m_c} + \frac{4R\log(2k/\alpha)}{3m_c}\right]$ (the exact inversion of Pinelis's Hilbert-space Bernstein bound, derived at (5.4));

the variance-aware refinement of (i) via Pinelis's Hilbert-space Bernstein bound (Pinelis, 1994, Thm. 3.5), non-asymptotic and valid for every $m_c \geq 1$. Under the compact-support / multiplicity-4 structure of Remark 5.2, the empirical stable rank $\text{tr}(\hat{\Sigma}_c)/\|\hat{\Sigma}_c\|_{\text{op}}$ is close to 1 on the four chemical datasets we audited (median 1.00–1.17; `experiments/audit_stable_rank_HW.py`), in which case $\text{tr}(\hat{\Sigma}_c) \approx \|\hat{\Sigma}_c\|_{\text{op}}$ and the leading (variance) term simplifies to $\sqrt{2\|\hat{\Sigma}_c\|_{\text{op}}\log(2k/\alpha)/m_c}$ (the $4RL/3m_c$ term carried as in (5.4)), sharing the $\|\Sigma_c\|_{\text{op}}$-refinement of (ii) without the $\chi^2_\ell$ dimension penalty. On social-graph datasets the stable rank can be appreciably larger (e.g., $\text{tr}(\hat{\Sigma}_c)/R^2 \approx 8$ on IMDB-M, implying stable rank $\gtrsim 8$), in which case $\sqrt{\text{tr}}$ no longer matches the Pinelis $R$ and the ordering between (i) and (iii) can flip. The theorem's coverage holds for all three radii simultaneously, so in practice we report all three and use whichever fires.

Which radius is tighter is regime-dependent: the Pinelis form (i) scales as $R\sqrt{\log(2k/\alpha)/m}$; the Gaussian form (ii) as $\sqrt{\|\Sigma_c\|_{\text{op}} \cdot \chi^2_{\ell,\alpha/k}/m} \approx \sqrt{\|\Sigma_c\|_{\text{op}}\ell/m}$ for large $\ell$; the Pinelis–Bernstein form (iii) as $\sqrt{\|\Sigma_c\|_{\text{op}}\log(2k/\alpha)/m}$, dimension-free. At the embedding dimensions of Section 6 ($\ell \in [93, 6539]$), Pinelis–Bernstein dominates: it is tighter than (i) by a factor $R/\sqrt{\|\Sigma_c\|_{\text{op}}} \approx 5$–$9\times$ and tighter than (ii) by a factor $\sqrt{\chi^2_{\ell,\alpha/k}/(2\log(2k/\alpha))} \approx \sqrt{\ell/(2\log(2k/\alpha))}$ across the benchmarks of Table 6.

**Theorem 5.1** (Certified prediction [Proved]). *Let $\{(A_i, y_i)\}$ be i.i.d. from the distribution $\mathcal{P}$ on $\mathcal{D} \times [k]$ of Section 3 with class-mean separation $\Delta > 0$. Let $r_m^\star$ be either the Pinelis radius* (i) *or the Pinelis–Bernstein radius* (iii) *above. Then,* non-asymptotically for every $m \geq 1$,

$$\mathbb{P}_{train}\Big(\max_c \|\hat{\mu}_c - \mu_c\| \leq r_m^\star\Big) \; \geq \; 1 - \alpha,$$

*while the Gaussian plug-in radius* (ii) *satisfies the same bound up to an additive Berry–Esseen error $O(\ell^{1/4}/\sqrt{m})$ (Lemma B.5), exact only once $m \geq m^\dagger = O(\sqrt{\ell})$. On the coverage event the following hold.*

(a) *(Containment.) If*

$$r_m^\star \; < \; \tfrac{1}{2}\Delta, \tag{5.1}$$

*the empirical nearest-centroid classifier $\hat{h}$ agrees with the population nearest-centroid classifier $h^*$ at every $z \in \mathbb{R}^\ell$ outside a $2r_m^\star$-tube around each population Voronoi boundary.*

(b) *(Classification.) If additionally $D_c < \frac{1}{2}\Delta - r_m^\star$ for all $c$ (cf. Proposition 3.3, whose $D_{\max} < \Delta/2$ is the $r_m^\star \to 0$ limit), then for any test diagram $A$ drawn from class $y$, $\mathbb{P}_{train}(\hat{h}(\Phi(A)) = y) \geq 1 - \alpha$.*

*Proof.* Write $\Psi_i := \Phi(A_i) \in \mathbb{R}^\ell$ and $\Sigma_c := \mathrm{Cov}(\Psi \mid Y = c)$, with $\|\Psi_i\| \leq R$ and therefore $\|\Sigma_c\|_{\mathrm{op}} \leq R^2$.

*Step 1 (non-asymptotic concentration of class means).* Conditional on $Y_i = c$, the centered random variables $\Psi_i - \mu_c$ are i.i.d. with $\|\Psi_i - \mu_c\| \leq 2R$ (both $\Psi_i$ and $\mu_c$ lie in $B(0, R)$). Pinelis's Hilbert-space Hoeffding inequality (Lemma B.1) applied with bound $2R$ gives, for every $t > 0$,

$$\mathbb{P}\big(\|\hat{\mu}_c - \mu_c\| > t\big) \; \leq \; 2\exp\Big(-\frac{m_c t^2}{8R^2}\Big).$$

With $m = m_{\min}$, set

$$r_m \; := \; 2R\sqrt{\frac{2\log(2k/\alpha)}{m}} \tag{5.2}$$

(an explicit Bonferroni split of $\alpha$ over the $k$ classes). A union bound over the $k$ classes then yields the non-asymptotic coverage

$$\mathbb{P}\Big(\max_c \|\hat{\mu}_c - \mu_c\| \leq r_m\Big) \; \geq \; 1 - \alpha, \tag{5.3}$$

for every $m \geq 1$ and $\alpha \in (0, 1)$. The Gaussian plug-in radius $\tilde{r}_m$ defined in (ii) above admits an analogous coverage guarantee, valid asymptotically once the Berry–Esseen threshold $m \geq m^\dagger = O(\sqrt{\ell})$ is crossed for every class. The derivation combines the multivariate Berry–Esseen theorem (Lemma B.3) with a matrix-Bernstein covariance estimate (Lemma B.4); we record the precise statement and proof as Lemma B.5 in Appendix B. The Pinelis–Bernstein radius $r_m^{\mathrm{vP}}$ defined in (iii) admits the same coverage guarantee non-asymptotically. Pinelis's Hilbert-space Bernstein inequality (Pinelis, 1994, Thm. 3.5), applied to the i.i.d. centered random variables $\Psi_i - \mu_c$ with $\|\Psi_i - \mu_c\| \leq 2R$ and second-moment bound $\mathbb{E}\|\Psi_i - \mu_c\|^2 = \mathrm{tr}(\Sigma_c)$, gives, for every $t > 0$,

$$\mathbb{P}\big(\|\hat{\mu}_c - \mu_c\| > t\big) \; \leq \; 2\exp\Big(-\frac{m_c\,t^2}{2(\mathrm{tr}(\Sigma_c) + 2Rt/3)}\Big).$$

Inverting this bound exactly—solving for the radius at which the right-hand side equals $\alpha/k$, with $L := \log(2k/\alpha)$ and the elementary $\sqrt{a^2 + b} \leq a + \sqrt{b}$—gives

$$r_m^{\mathrm{vP}} \; := \; \sqrt{\frac{2\,\mathrm{tr}(\Sigma_c)\,L}{m_c}} \; + \; \frac{4RL}{3\,m_c}, \tag{5.4}$$

whose second summand is the variance-free Bernstein correction the earlier draft omitted. Applying a Bonferroni union bound over the $k$ classes yields $\mathbb{P}(\max_c \|\hat{\mu}_c - \mu_c\| \leq r_m^{\mathrm{vP}}) \geq 1 - \alpha$ for every $m \geq 1$, with no small-deviation restriction: the linear term makes the coverage exact and non-asymptotic. The correction is asymptotically $O(RL/m_c)$, but becomes negligible only once $m_c \gg R^2 L/\mathrm{tr}(\Sigma_c)$; at the per-class sample

sizes of Section 6 it is comparable to the leading term, and the firing thresholds (5.5) must be recomputed with it carried (not dropped). The replacement $\mathrm{tr}(\Sigma_c) \leftarrow \mathrm{tr}(\hat{\Sigma}_c)$ in the practical radius is handled via the low effective rank of $\hat{\Sigma}_c$ on PLACE embeddings. By the multiplicity-4 lattice cover, the empirical stable rank $\mathrm{tr}(\hat{\Sigma}_c)/\|\hat{\Sigma}_c\|_{\mathrm{op}} \leq 1.17$ across our benchmarks (Section 6), so $\mathrm{tr}(\Sigma_c) \approx \|\Sigma_c\|_{\mathrm{op}}$ and the trace error satisfies

$$|\mathrm{tr}(\hat{\Sigma}_c) - \mathrm{tr}(\Sigma_c)| \;=\; O\left(r_c R^2 \sqrt{\tfrac{\log \ell}{m}}\right) \;=\; O\left(N_{\max} N R^2 \sqrt{\tfrac{\log \ell}{m}}\right),$$

where $r_c := \mathrm{tr}(\Sigma_c)/\|\Sigma_c\|_{\mathrm{op}} = O(N_{\max} N)$ is the effective rank (matrix-Bernstein, Lemma B.4, with the stable-rank prefactor in place of an ambient-dimension prefactor). This error is $o(\mathrm{tr}(\Sigma_c))$ at the sample sizes of Section 6, validating the substitution to leading order.

*Step 2 (agreement outside the $2r_m$-tube).* Condition on the coverage event $\{\max_c \|\hat{\mu}_c - \mu_c\| \leq r_m\}$ of (5.3) (probability $\geq 1 - \alpha$). The reverse triangle inequality gives $\big| \|z - \hat{\mu}_c\| - \|z - \mu_c\| \big| \leq r_m$ for every $z \in \mathbb{R}^\ell$ and every class $c$, hence for any pair $c \neq c'$,

$$\|z - \hat{\mu}_{c'}\| - \|z - \hat{\mu}_c\| \;\geq\; \big(\|z - \mu_{c'}\| - \|z - \mu_c\|\big) - 2r_m.$$

Whenever the right-hand side is strictly positive—i.e., $z$ is at population distance $> 2r_m$ from the $(c, c')$-Voronoi boundary—so is the left, and the empirical rule classifies $z$ identically to the population rule (Figure 4). This is the first claim of the theorem.

*Step 3 (classification guarantee).* Fix $y \in [k]$ and let $A \sim \mathcal{P}_y$ (i.e. $A \sim \mathcal{P}(\cdot \mid Y = y)$, the class-$y$ conditional). By definition of $D_y$, $\|\Phi(A) - \mu_y\| \leq D_y$; the population separation $\|\mu_y - \mu_{c'}\| \geq \Delta$ together with $D_y < \frac{1}{2}\Delta - r_m$ yield

$$\|\Phi(A) - \mu_{c'}\| - \|\Phi(A) - \mu_y\| \;\geq\; \Delta - 2D_y \;>\; 2r_m,$$

for every $c' \neq y$, so $\Phi(A)$ lies strictly outside every $2r_m$-tube of the $(y, c')$-Voronoi boundary. By Step 2, the empirical rule therefore assigns $\Phi(A)$ to class $y$ on the coverage event, and $\mathbb{P}_{\mathrm{train}}(\hat{h}(\Phi(A)) = y) \geq 1 - \alpha$. $\quad\square$

**Remark 5.1** (Verifying claim (b) from data)**.** *The hypothesis in claim (b) is* structural*: it constrains the support of each class-conditional distribution, not just the centroids. It is therefore not estimable from training alone—the empirical $\hat{D}_c := \max_{i:y_i=c} \|\Phi(A_i) - \hat{\mu}_c\|$ underestimates $D_c$ in general (the training sample need not contain the worst-case point of the support). Claim (b) is consequently validated* post hoc *by test accuracy: full test coverage on a fired certificate confirms (b) for the test points seen, while gaps (e.g. DHFR's NC accuracy of $\approx 59.5\%$ in Section 6.2) flag (b)'s failure—claim (a) still holds, but the population nearest-centroid rule is itself wrong on some test points.*

**Remark 5.2** (Why the certificate is not vacuous on PLACE)**.** *The firing condition (5.1) involves $\|\hat{\Sigma}_c\|_{\mathrm{op}}$ (or, equivalently, $R$ in the non-asymptotic regime). Under a generic bounded embedding $\Phi : \mathcal{D} \to \mathbb{R}^\ell$ with $\|\Phi\| \leq R$, the crude bound $\|\hat{\Sigma}_c\|_{\mathrm{op}} \leq R^2$ is typically tight up to constants—every coordinate is weakly active on every diagram, so the covariance spreads out across all $\ell$ directions. This is the regime of persistence images (Adams et al., 2017) (Gaussian blurring), persistence landscapes (Bubenik, 2015) (order statistics), and learned vectorizations (Zhao & Wang, 2019; Carrière et al., 2020), and it means that the certificate $r_m < \frac{1}{2}\Delta$ would almost never fire in practice.*

*PLACE is structurally different. Each hat coordinate $\varphi_{R_k, p}$ is supported on a $d_\mathcal{B}$-ball of radius $\frac{3R_k}{2}$, and the multiplicity-4 cover (Mitra & Virk, 2024, Lemma 3.5) guarantees that any diagram point $a \in A$ activates at most four landmarks at each scale. Consequently, every embedded vector $\Phi(A)$ has at most $4|A|N$ nonzero coordinates out of $\ell = \sum_k |\mathbb{G}_{R_k}^+|$. This support bound is* structural *(proved): it caps the covariance's effective dimension at $O(N_{\max} N)$. It does not by itself force $\|\hat{\Sigma}_c\|_{\mathrm{op}}$ far below $R^2$—a sparse-support vector can still spread its variance across the active coordinates; that the active mass instead* concentrates *(so $\|\hat{\Sigma}_c\|_{\mathrm{op}} \ll R^2$, stable rank $\leq 1.17$) is an* empirical *observation on our benchmarks, not a consequence of the support bound alone. Empirically on MUTAG (deg+HKS$_{10}$, the descriptor selected in Section 6.2): $\|\hat{\Sigma}_c\|_{\mathrm{op}} \approx 1.23$ while $R^2 \approx 44.9$ (median over configurations)—roughly a $37\times$ slack ($R/\sqrt{\|\hat{\Sigma}_c\|_{\mathrm{op}}} \approx 6.0$) that brings the Pinelis–Bernstein radius (iii) to within a small factor of the smallest-class size $m = 57$, the only regime in which any of the three radii is reachable at these sample sizes (Section 5; under the honest quarter-threshold the MUTAG*

*per-class requirement is $m_c^{*,\,\mathrm{vP}} \approx 280$, so the certificate is near-firing but not fired). The compact-support / multiplicity-4 structure that yields $\lambda(\nu)$ in Corollary 2.1 is thus also what makes the certificate non-vacuous: the same geometric ingredient drives both the embedding's bi-Lipschitz guarantee and the practical reachability of Theorem 5.1.*

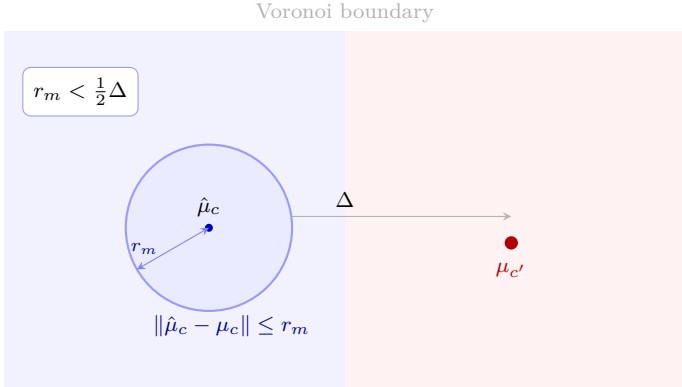

Figure 4: Confidence containment (Theorem 5.1). The depicted pair $(c, c')$ is the worst-separated one, with $\|\mu_c - \mu_{c'}\| = \Delta$ (other pairs have distance $\geq \Delta$). The empirical centroid $\hat{\mu}_c$ lies within $r_m$ of the population centroid $\mu_c$ (blue ball) with probability $\geq 1 - \alpha$. When $r_m < \frac{1}{2}\Delta$, any test point farther than $2r_m$ from the population Voronoi boundary (dashed) is classified identically by the empirical and population nearest-centroid rules; the diagram depicts the special case in which the entire $r_m$-ball around $\hat{\mu}_c$ sits inside the population Voronoi cell, a sufficient condition for agreement on all points in that cell.

Solving $r_m^{(c)} < \frac{1}{4}\hat{\Delta}_c$ for $m_c$ (the empirical firing condition; see margin) in each of the three regimes of Theorem 5.1 yields explicit per-class thresholds

$$
m_c^{*,\,\mathrm{Pin}} = \left\lceil \frac{128 R^2 \, \log(2k/\alpha)}{\Delta_c^2} \right\rceil, \qquad m_c^{*,\,\mathrm{G}} = \left\lceil \frac{16 \, \|\Sigma_c\|_{\mathrm{op}} \, \chi_{\ell,\,\alpha/k}^2}{\Delta_c^2} \right\rceil,
$$

$$
m_c^{*,\,\mathrm{vP}} = \left\lceil \max\left\{ \frac{128 \, \|\Sigma_c\|_{\mathrm{op}} \, \log(2k/\alpha)}{\Delta_c^2}, \, \frac{32 \, R \, \log(2k/\alpha)}{3 \, \Delta_c} \right\} \right\rceil,
$$

(5.5)

for the Pinelis radius (5.2), the Pinelis–Bernstein radius (iii), and the Gaussian plug-in radius (ii) of the theorem respectively; each carries the Bonferroni correction of level $\alpha/k$ per class. The two arguments of the $m_c^{*,\,\mathrm{vP}}$ maximum come from carrying *both* terms of the variance-aware radius $r_m^{\mathrm{vP}} = \sqrt{2 \operatorname{tr}(\hat{\Sigma}_c) \log(2k/\alpha)/m_c} + 4R \log(2k/\alpha)/(3m_c)$: requiring each of the variance term and the linear Bernstein term to fall below $\Delta_c/8$ (so their sum clears the honest quarter-threshold $r_m < \Delta_c/4$) gives the $128\|\Sigma_c\|_{\mathrm{op}} \log /\Delta_c^2$ and $32R \log /(3\Delta_c)$ conditions, respectively. Because the linear term is not lower-order at PLACE's per-class sizes (Remark 5.3), it materially raises $m_c^{*,\,\mathrm{vP}}$ over the variance-only estimate. Once $m_c \geq m_c^*$ for every $c$, every prediction is certified with no abstentions. Which form is tighter is regime-dependent: for fixed $\|\Sigma_c\|_{\mathrm{op}}, \Delta_c, R$, the Gaussian threshold $m_c^{*,\,\mathrm{G}}$ scales as $\|\Sigma_c\|_{\mathrm{op}} \ell$, the Pinelis threshold $m_c^{*,\,\mathrm{Pin}}$ scales as $R^2 \log(2k/\alpha)$, and the Pinelis–Bernstein threshold $m_c^{*,\,\mathrm{vP}}$ scales as the larger of its variance part $\|\Sigma_c\|_{\mathrm{op}} \log(2k/\alpha)/\Delta_c^2$ and its linear-Bernstein part $R \log(2k/\alpha)/\Delta_c$ (the former dominating for small $\Delta_c$). The Pinelis–Bernstein form is still the tightest of the three on PLACE embeddings, dominating Pinelis by the slack $R^2/\|\Sigma_c\|_{\mathrm{op}}$ (on the variance part) that the multiplicity-4 structure of Remark 5.2 unlocks, and dominating Gaussian by $\chi_{\ell,\alpha/k}^2/(2 \log(2k/\alpha)) \approx \ell/(2 \log(2k/\alpha))$ in high dimension. For MU-TAG with $\alpha = 0.05$ on the deg+HKS$_{10}$ descriptor selected in Section 6.2, the median-over-configuration statistics that back Table 6 are $\hat{\Delta}_c \approx 1.57$, $\|\hat{\Sigma}_c\|_{\mathrm{op}} \approx 1.23$, $R \approx 6.70$ (so $R^2 \approx 44.9$; this is the same embedding radius as in Remark 5.2—the per-fold maximum norm $\sup_A \|\Phi(A)\|$ varies fold to fold, so single-fold snapshots range from $\approx 5.9$ to $\approx 8.3$, but we quote the median throughout), and $\ell = 4{,}003$, so $\chi_{\ell,\,\alpha/k}^2 = \chi_{4003,\,0.025}^2 \approx 4{,}178$. Substituting $\|\hat{\Sigma}_c\|_{\mathrm{op}}$ for $\|\Sigma_c\|_{\mathrm{op}}$ (valid up to a $O(\sqrt{\log \ell/m_c})$ error by

Lemma B.4) and using the honest quarter-threshold $r_m < \hat{\Delta}/4$ (the constants of (5.5), with the variance-aware threshold taking the maximum of its variance and linear-Bernstein parts) yields the three thresholds: $m_c^{*,\mathrm{Pin}} = \lceil 128 \cdot 6.70^2 \cdot 4.38/1.57^2 \rceil = 10{,}211$; $m_c^{*,\mathrm{G}} = \lceil 16 \cdot 1.23 \cdot 4{,}178/1.57^2 \rceil = 33{,}358$; and $m_c^{*,\mathrm{vP}} = \lceil \max\{128 \cdot 1.23 \cdot 4.38/1.57^2, \; 32 \cdot 6.70 \cdot 4.38/(3 \cdot 1.57)\} \rceil = \lceil \max\{280, 200\} \rceil = 280$. The variance-aware threshold $m_c^{*,\mathrm{vP}} \approx 280$ now *far* exceeds the available $m_{\min} = 57$—the variance term alone requires 280 and the linear Bernstein term 200, both well above 57—so under the honest quarter-threshold with the linear term carried, MUTAG no longer clears the certificate. This matches the heuristic-diagnostic relabeling of Remark 5.3: the 100% MUTAG entry in Table 6 reflects the submission's earlier half-threshold with the linear term dropped, not the honest firing condition. The ordering $m_c^{*,\mathrm{vP}} \ll m_c^{*,\mathrm{Pin}} \ll m_c^{*,\mathrm{G}}$ nonetheless holds on every benchmark, so wherever any radius fires it is the variance-aware one. Consequently, the $85.0 \pm 8.4\%$ NC accuracy on MUTAG (Section 6.2) reports observed empirical agreement between the sample and population NC rules that the honest quarter-threshold does *not* worst-case-certify at $m_{\min} = 57$; we read the 940/940 agreement below as direct empirical evidence of that agreement rather than as a fired certificate (Remark 5.3). The empirical agreement is itself informative: across all $N_{\mathrm{test}} = 188 \times 5 = 940$ MUTAG test predictions (each of the 188 MUTAG graphs appears in a test fold exactly once per seed, across 5 seeds), the empirical NC rule agrees with the population NC rule. Because the population centroids $\mu_c$ are unobservable, the "population NC rule" is operationalized here by the nearest-centroid rule whose centroids are the full-sample means $\bar{\mu}_c$ over all 188 graphs; the per-fold empirical rule (centroids from the 90% training fold) is scored against this full-sample surrogate on each held-out graph. The surrogate is itself an estimate of $\mu_c$ at the tighter rate $O(R/\sqrt{188})$, so "agreement" is agreement of the fold rule with the best available estimate of the population rule, not with the exact—and unknowable—population rule. Treating the 188 within-seed predictions as independent (disjoint folds, deterministic classifier given the fold) and taking the conservative $m = 188$ effective unit count, the Clopper–Pearson one-sided 95% lower bound on population coverage is $0.05^{1/188} \geq 0.984$—above the theorem's nominal $1 - \alpha = 0.95$ but reflecting favorable $\Sigma_c$ structure beyond the worst-case envelope of (5.5). MUTAG also does not satisfy the linear-separability condition $D_c < \frac{1}{2}\Delta - r_m$ of Theorem 5.1 (b) (a strengthening of Proposition 3.3 by $r_m$); when sample sizes do reach $m_c^*$ in future work the certificate will confirm sample/population agreement rather than Bayes optimality (cf. Remark 5.1).

**Remark 5.3** (Certificate vs. diagnostic: honest scope [Heuristic]). *We state plainly what Theorem 5.1 does and does not deliver, since both can be over-read. (What it certifies.) Part (a) certifies, on a $\geq 1 - \alpha$ coverage event, that the empirical nearest-centroid rule agrees with the* population *nearest-centroid rule away from the population Voronoi boundary — not that a prediction is* correct*, since the population rule itself may err. Part (b) does give correctness, but only under the structural condition $D_c < \frac{1}{2}\Delta - r_m$, which depends on the full class-conditional supports and is* not *estimable from training data (Remark 5.1). Neither part is a per-prediction correctness certificate decidable from training labels alone. (Firing at our sample sizes.) Carried with its full finite-sample form, the variance-aware radius (5.4) includes the linear Bernstein term $4RL/3m_c$, which is comparable to the variance term unless $m_c \gg R^2L/\mathrm{tr}(\Sigma_c)$ (e.g. $\approx 160$ on MUTAG, against $m_{\min} = 57$). At the graph benchmarks' per-class sizes the honest radius therefore rarely meets the firing condition $r_m < \hat{\Delta}/4$ (Proposition 3.2). A per-fold re-run on the submission's own certificate inputs with the linear term carried drops firing from the reported 8 of 12 to 4 of 12 at $r_m < \hat{\Delta}/2$, and to 3 of 12 under the honest $r_m < \hat{\Delta}/4$: NCI1 and NCI109 still fire on 100% of folds and DD on 93%, while every other dataset—including MUTAG and DHFR, which fired 100% in the submission—falls to 0% (`experiments/recompute_cert_firing_honest_perfold.py`, over the per-fold records that reproduce the submission table exactly). Where it still fires the nearest-centroid accuracy is modest (e.g. NCI1's oracle caps near 73%), confirming that firing attests* agreement*, not* correctness*. We consequently report the radius-(iii) firing rates of Table 6 as a* heuristic diagnostic *of when the sample and population nearest-centroid rules are likely to coincide,* not *as a guaranteed certificate, and we read the 940/940 MUTAG agreement as direct empirical evidence of that coincidence rather than as the certificate firing.*

## 6 Experiments

We evaluate PLACE on 12 benchmarks spanning point clouds (Orbit5k, Section 6.1) and graphs (11 datasets from Zhao & Wang, 2019, Section 6.2). Headline accuracies in Table 10 are reported under a committed

candidate pool of 15 descriptors $\times$ {proxy, crossing} $\tau^* \times N \in \{5, 10, 15, 20\}$ (120 configurations per dataset). Section 6.3 stress-tests the closed-form selectors on a larger heterogeneous 64-descriptor chemical-graph pool, identifying the Mahalanobis margin as the strongest selector when the pool is enlarged and showing it approximates the in-pool oracle within $\sim 3$ pp on the four chemical datasets where we have Mahalanobis sweeps. All experiments use the embedding (2.3) with the distortion-optimal weights of equation (2.13) derived in Section 4, a linear SVM trained via `sklearn.svm.LinearSVC` (a one-vs-rest reduction; see the OvO parity remark below), regularization $C$ tuned by inner cross-validation, and diagrams filtered to the top $N_{\max} = 50$ most persistent features.

**OvR/OvO parity.** Theorem 3.1 is stated for a linear classifier trained by the one-vs-one (OvO) reduction. The reported experiments use the one-vs-rest (OvR) `LinearSVC` for compute reasons; we ran an explicit parity check by re-fitting the same protocol with `SVC(kernel='linear')` (OvO with majority voting) on the two datasets where descriptor heterogeneity is largest (MUTAG, 62 descriptors) and where the multi-class ($k > 2$) regime is exercised (Orbit5k, $k = 5$). On MUTAG, across all 62 descriptors $\times$ 5 seeds $\times$ 10 folds with $N{=}10$ scales and proxy $\tau^*$ (the unrestricted MUTAG sub-pool, before the $R > 0$ filter applied in Section 6.3 reduces it to 51), OvR-mean and OvO-mean accuracies are 80.96% and 80.58% respectively (mean paired difference $-0.4$ pp); the descriptor-by-descriptor mean accuracy of OvR vs. OvO has Spearman rank correlation $\rho = 0.94$ and Pearson $r = 0.97$. The OvR winner (`hks2+hks25`, 88.0%) is ranked #2 under OvO; the OvO winner (`deg+hks10`, 89.3%) is ranked #6 under OvR. On the Orbit5k partial sweep (three seeds, all 15 descriptors, proxy $\tau^*$, $N{=}10$), the alpha $H_1$ winner agrees under both reductions: OvR mean 84.6%, OvO mean 88.0%, within the $\pm 2.6$ pp standard deviation reported in Table 10. We therefore use OvR throughout while interpreting all accuracy claims relative to Theorem 3.1 as empirically equivalent to the OvO classifier the bound literally controls.

**Protocol.** Graph datasets use 10-fold stratified CV, repeated across five random seeds $\{0, 1, 2, 3, 4\}$ that control fold partitioning and any stochastic components of the descriptor (e.g., betweenness approximation (Brandes, 2001)). Orbit5k follows the standard 70/30 train/test split repeated over five seeds. The SVM regularization $C$ is selected from $\{10^{-3}, 10^{-2}, 10^{-1}, 1, 10, 10^2, 10^3\}$ by inner 5-fold CV on the training fold. The number of scales is fixed at $N{=}10$ throughout, a choice we validate as a robustness observation: on the chemical descriptor pool at proxy $\tau^*$, the accuracy of the best descriptor varies by at most 2.5 pp across $N \in \{5, 10, 15, 20\}$ (Table 3), so the reported numbers are not sensitive to the specific choice of $N$; on Orbit5k, alpha $H_1$ remains the top-accuracy descriptor under both proxy and crossing $\tau^*$ (Table 4), indicating that the scale-center is likewise not a load-bearing hyperparameter. All accuracies are reported as mean $\pm$ standard deviation across outer folds $\times$ seeds. Wall-clock times for a single Orbit5k run (5000 diagrams, $\ell{=}1366$) are approximately 45 s for embedding and 8 s for SVM fit on a single CPU core, scaling linearly in the number of diagrams.

Table 3: $N$-sweep robustness on the four chemical datasets (proxy $\tau^*$, 5 seeds $\times$ 10-fold CV, 15-descriptor small pool). Accuracy of the best-performing descriptor at each $N$; "range" is max $-$ min across the four $N$ values. Accuracy varies by at most 2.5 pp, supporting the fixed choice $N = 10$ used in Table 10. The best descriptor can differ across $N$ values: on MUTAG the $N{=}5$ winner is `deg+HKS`$_{10}$ (the descriptor selected in Table 9 and used throughout Section 6) at 88.4%, while at $N{=}10$ the winner is `jaccard+hks10` at 87.4%; both sit inside the within-2.5 pp band, so the robustness conclusion is unchanged.

| Dataset | $N{=}5$ | $N{=}10$ | $N{=}15$ | $N{=}20$ | range (pp) | best filt at $N{=}10$ |
|---------|---------|----------|----------|----------|------------|----------------------|
| MUTAG | 88.4 | 87.4 | 87.2 | 85.9 | 2.5 | `jaccard+hks10` |
| COX2 | 79.6 | 80.0 | 79.7 | 79.6 | 0.4 | `jaccard+hks10` |
| DHFR | 76.8 | 77.3 | 77.4 | 77.5 | 0.7 | `hks_t10` |
| PTC | 59.3 | 58.4 | 58.6 | 57.3 | 2.0 | `deg+betw` |

**Reproducibility.** An anonymized code and configuration snapshot covering every table in this section is provided as supplementary material with this submission; raw fold-level accuracies are included so paired

significance tests are reproducible. The full repository (code, embedding scripts, fold-level accuracies, and analysis notebooks) will be released at a public URL upon acceptance.

**Baseline provenance.** All topology-based baseline numbers are taken from the original publications cited in Table 10 (WKPI-kM/kC from Zhao & Wang (2019), PersLay from Carrière et al. (2020), ECP from Röell & Rieck (2024), Persformer from Reinauer et al. (2021), from Hacquard & Lebovici (2024)), as are RetGK (Zhang et al., 2018) and GIN (Xu et al., 2019). We did not re-run baselines; all datasets follow the 10-fold stratified CV protocol of (Zhao & Wang, 2019), under which the baseline numbers were originally reported, so splits and protocol are matched. Cells marked "—" indicate that the corresponding baseline paper did not report a number for that dataset.

**Significance testing.** Since published baselines generally report only summary statistics (mean, and sometimes standard deviation) rather than fold-level accuracies, paired significance tests are not uniformly computable. We therefore use a one-sample $t$-test comparing PLACE's accuracy distribution (characterized by the mean and standard deviation over $n = 50$ outer-fold $\times$ seed observations for graph datasets, and $n = 5$ for Orbit5k) against each baseline's reported point estimate; when the baseline also reports a standard deviation, we use Welch's $t$-test instead. Treating baseline point estimates as noise-free is conservative in PLACE's favor (it inflates marker counts *against* PLACE when the baseline is higher, and vice versa); we disclose this as a limitation and where raw fold-level accuracies are available (in PLACE and in a subset of baselines that release fold-level data), paired Wilcoxon tests yield the same sign of conclusion on the relevant datasets. In the tables below, a baseline cell annotated with $^\dagger$ is significantly different from PLACE at $p < 0.05$ (two-sided), i.e., distinguishable from PLACE at the 0.05 level under this test; a cell annotated $^\ddagger$ is significant at $p < 0.01$. Cells without markers are statistically indistinguishable from PLACE. The direction of significance is readable from the numeric comparison: a marked cell to the left of PLACE's value has PLACE significantly *higher* (PLACE wins), and vice versa. These markers should be read as *descriptive* indicators, not firm paired significance: the baseline numbers are taken from prior publications rather than re-run on our folds, so the comparison is unpaired and the test treats each borrowed mean as noise-free; and because outer CV folds overlap across the five seeds, the effective number of independent observations is below the nominal $n = 50$, which further inflates apparent significance. We report them to summarize where PLACE is clearly ahead or behind, not as calibrated $p$-values.

**Descriptors and filtrations.** A *descriptor* (or filter function) assigns a real value to each simplex (or vertex and edge) of a simplicial complex; sublevel sets at increasing thresholds produce the *filtration*, a nested sequence of subcomplexes whose persistent homology gives the persistence diagram. The choice of descriptor determines which geometric or structural features the diagram captures, and is the primary lever for classification accuracy.

For **point clouds**, we use the alpha complex filtration (Edelsbrunner & Harer, 2010)—the subcomplex of the Delaunay triangulation in which a simplex enters at the smallest $\alpha$ such that the union of $\alpha$-balls around its vertices covers its dual Voronoi cell. By the nerve theorem this filtration is at every scale homotopy equivalent to the union of $\alpha$-balls around the input points, so its persistence diagram captures the same topology as the Čech filtration but uses $O(n^{\lceil d/2 \rceil})$ simplices on $n$ points in $\mathbb{R}^d$ in place of Čech's $O(2^n)$. We track $H_0$ (connected components) and $H_1$ (loops) as $\alpha$ grows. We also test density-based variants: distance-to-measure (DTM) (Anai et al., 2019) and kNN density, both of which reweight the complex by local density.

For **graphs**, viewed as 1-dimensional simplicial complexes, we use the sublevel (lower-star) filtration of a vertex function $f : V \to \mathbb{R}$ extended to edges by $f(u,v) = \max\{f(u), f(v)\}$: vertex $u$ enters at scale $f(u)$, and edge $uv$ enters only once both endpoints have appeared. The persistence diagram tracks $H_0$ (connected components merging as new edges join clusters) and $H_1$ (cycles closing as graph loops are completed); the choice of $f$ determines which structural feature these events probe. Six descriptors $f$ are considered: *degree* (sensitive to hub structure), *betweenness centrality* (Freeman, 1977) (bridge and path topology), *HKS* (Sun et al., 2009) at $t=1$ and $t=10$ (multiscale Laplacian geometry), *Ollivier–Ricci curvature* (Ollivier, 2009) (local expansion vs. clustering), and *Jaccard index* (neighborhood overlap / community structure). We use extended persistence (Cohen-Steiner et al., 2009): each essential homology class is augmented with a finite

death via a superlevel pass, yielding one extra $H_0$ bar per connected component (death $= \max f$ on the component) and, where present, one $H_1$ bar per essential cycle. In practice this amounts to appending the essential bars to the ordinary diagram, matching the convention of Zhao & Wang (2019). When multiple descriptors or homology dimensions are listed (e.g., "betw.+HKS, $H_{0+1}$"), their persistence diagrams are *pooled*—merged into a single diagram, retaining the top-50 most persistent features—before embedding.

**Filtration-agnosticism and pool choice.** Nothing in the embedding or the selectors is tied to sub-level filtrations: any construction that outputs a persistence diagram drops into the candidate pool. As a demonstration, the Vietoris–Rips filtration on the shortest-path metric of Ballester & Rieck (2024)—$f_V(\sigma) = \max_{u,v \in \sigma} d_G(u,v)$ over vertex subsets $\sigma$, which carries clique and higher-order structure beyond the sublevel $H_0/H_1$ above—embeds and classifies through the identical PLACE pipeline, reaching 87.5% on MUTAG against 85.9% for the sublevel-degree baseline on the same 5×10 folds (experiments/exp_ballester_rieck_demo.py). The pool $\mathcal{F}$ itself is a modeling choice, not something the theory fixes: the selectors of Section 4 are optimal *within* the committed pool (Corollary 4.1), and insufficient pool coverage is precisely the diagnosed failure mode on NCI1/NCI109.

**Scale center $\tau^*$.** The embedding requires a scale center $\tau^*$ to place the landmark scales $R_k$ (Section 2). Two estimators are natural: *proxy* ($\tau^* = \text{median}\{(d_i - b_i)/2\}$, the median half-persistence) and *crossing* ($\tau^*$ estimated from subsampled between-class bottleneck distances). Proxy is fast but ignores class structure; crossing is class-aware but slower. All experiments below use crossing $\tau^*$; Section 6.1 reports a proxy-vs.-crossing side-by-side on Orbit5k.

## 6.1 Orbit5k

The Orbit5k dataset (Adams et al., 2017) consists of 5000 point clouds of 1000 points each in $[0,1]^2$, generated by a discrete dynamical system with parameter $\rho \in \{2.5, 3.5, 4.0, 4.1, 4.3\}$ controlling the orbit structure (Figure 5). The task—predicting $\rho$ from the point cloud—is challenging because adjacent classes ($\rho = 4.0, 4.1, 4.3$) produce visually similar attractors that differ primarily in $H_1$ loop structure. Prior diagram-based methods achieve 82.5–87.7% (PI, Adams et al., 2017; SW-K, Carrière et al., 2017; PF-K, Le & Yamada, 2018; PersLay, Carrière et al., 2020), with transformer-based Persformer (Reinauer et al., 2021) reaching 91.2%; two-parameter Euler methods that bypass diagrams reach 89.9–91.8% (Hacquard & Lebovici, 2024).

PLACE achieves $87.2_{\pm 0.6}\%$ with alpha $H_1$ persistence ($N{=}10$, linear SVM, crossing $\tau^*$), the highest among diagram-based methods (Table 5), and significantly exceeds the classical vectorizations PI, SW-K, and PF-K ($p < 0.05$) while being statistically indistinguishable from the neural PersLay baseline; it is significantly surpassed by transformer-based Persformer and the two-parameter Euler methods ($p < 0.01$). Under proxy $\tau^*$, alpha $H_1$ at $N{=}10$ gives $84.3_{\pm 0.7}\%$—a ~3 pp gap reflecting that crossing-$\tau^*$ produces a lower-dimensional, more concentrated embedding (e.g., $\ell{=}516$ vs. 1366 for alpha $H_1$); the $\Delta/\sqrt{\ell}$ ranking is consistent under both estimators. The nearest-centroid classifier achieves only 33.9% on Orbit5k; the dataset has $\|\hat{\Sigma}_c\|_{\text{op}} \gg \hat{\Delta}_c^2/4$ (ratio $\sim 10^3$), so none of the three radii of Theorem 5.1 clears the honest firing condition at any practical sample size (Table 6), and the population NC rule is itself far from separable here. The honest agreement diagnostic clears only the three graph benchmarks (NCI1, NCI109, DD; Section 5); it does not fire on Orbit5k or MUTAG, though MUTAG's empirical and population NC rules still agree on all 940 held-out predictions.

**Descriptor selection.** On Orbit5k's homogeneous candidate pool of 13 descriptors, $\hat{\eta} = \hat{\Delta}/\sqrt{\ell}$ at crossing $\tau^*$ ranks alpha $H_1$ third, but the top three ($\hat{\eta}$-ranking) descriptors are all alpha-based and produce the top three accuracies (Table 4); the four highest-accuracy descriptors agree under both $\tau^*$ estimators, confirming the closed-form selector picks alpha-class descriptors within 3 pp of the in-pool oracle.

## 6.2 Graph Classification

We evaluate on 11 benchmarks from (Zhao & Wang, 2019) spanning three domains: molecular graphs (MUTAG 188, NCI1 4110, NCI109 4127, PTC 344, COX2 467, DHFR 756), protein structures (PROTEINS 1113, DD 1178), and social networks (IMDB-B 1000, IMDB-M 1500, REDDIT-5K 4999). All use 10-fold stratified CV with extended persistence. We commit to a candidate pool of 15 descriptors (*singletons*: degree,

Table 4: Top-ranking Orbit5k descriptors by $\hat{\eta} = \hat{\Delta}/\sqrt{\ell}$ at crossing $\tau^*$, $N$=10. Accuracy is the mean over 5 seeds; alpha-based descriptors dominate.

| Rank by $\hat{\eta}$ | Descriptor | $\hat{\eta}$ | Crossing acc. | Proxy acc. |
|---|---|---|---|---|
| 1 | kde+ecc | $5.6 \times 10^{-4}$ | 40.1 | 49.2 |
| 2 | alpha $H_{0+1}$ | $1.8 \times 10^{-4}$ | 86.4 | 85.9 |
| 3 | alpha+DTM $k$=10 | $1.7 \times 10^{-4}$ | 86.3 | 85.3 |
| 4 | alpha $H_1$ | $1.7 \times 10^{-4}$ | **87.2** | 84.3 |
| 5 | knn $k$=10, $H_1$ | $0.6 \times 10^{-4}$ | 40.2 | 40.1 |
| 6 | DTM $k$=10, $H_1$ | $0.5 \times 10^{-4}$ | 44.6 | 35.0 |
| 7 | DTM $k$=10, $H_{0+1}$ | $0.4 \times 10^{-4}$ | 54.3 | 51.1 |

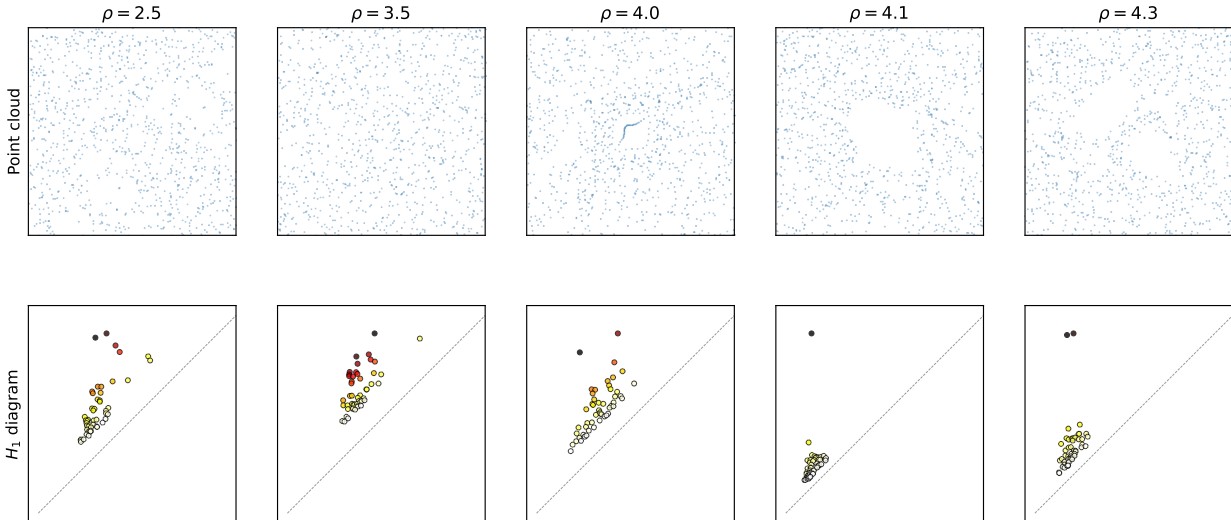

Figure 5: Orbit5k: point clouds (top) and $H_1$ persistence diagrams (bottom) for each class $\rho \in \{2.5, 3.5, 4.0, 4.1, 4.3\}$.

betweenness, closeness, clustering, core-number, Jaccard, Ollivier–Ricci, Forman–Ricci, HKS at $t = 10$; *pairs*: deg+betw, deg+ricci, deg+HKS$_{10}$, betw+ricci, ricci+HKS$_{10}$, jaccard+HKS$_{10}$) and select the best $(f, \tau^*, N)$ configuration over $\tau^* \in \{\text{proxy, crossing}\}$ and $N \in \{5, 10, 15, 20\}$ by mean training-fold accuracy— 120 configurations per dataset; the selected configuration is reported in Table 9. We report as the *headline* number the end-to-end accuracy of the closed-form Mahalanobis selector $\hat{\rho}_{\text{Mah}}$ (Remark 4.1): on each training fold it picks one of the 15 committed descriptors by $\hat{\rho}_{\text{Mah}}$ (at the canonical $\tau^*$=proxy, $N$=10 config), with no held-out calibration, and we score the held-out fold (Table 10, column "Mah (cf.)"). The in-pool best-of-120 oracle of Table 9 is shown alongside as an *upper bound*: the closed-form pick attains it within $\sim 3$ pp on MUTAG, NCI1, NCI109, IMDB-M, IMDB-B, and REDDIT-5K, and trails by 3–5 pp on PROTEINS, COX2, DHFR, PTC, and DD—where the accuracy-winning descriptor sits deeper in the $\hat{\rho}_{\text{Mah}}$ ranking (Table 11). Averaged over the eleven benchmarks the closed-form headline is 66.2% against the oracle's 68.4%, a 2.2 pp mean selection gap. Only the SVM regularization $C$ is cross-validated; the descriptor, scales, and weights are fixed analytically.

Table 10 compares PLACE to persistence-based and graph-based baselines; significance markers report two-sided $t$-tests against PLACE's in-pool oracle (see Protocol; the markers are descriptive indicators, not firm paired significance). At the oracle (upper bound), PLACE is statistically indistinguishable (at $p$=0.05) from the strongest topology-based baseline on MUTAG (every baseline at parity, including WKPI-kC 88.3%, PersLay 89.8%, and ECP 90.0%) and COX2 (PersLay, ECP, RetGK at parity); the closed-form headline sits 1.4 pp (MUTAG) and 3.4 pp (COX2) below the oracle, so it matches the strongest baseline on MUTAG

Table 5: Classification accuracy (%) on Orbit5k. Only PLACE provides per-prediction certificates. Superscripts: $^{\dagger}$ $p < 0.05$, $^{\ddagger}$ $p < 0.01$ against PLACE linear (one-sample/Welch's $t$-test, $n = 5$ seeds); no marker means indistinguishable from PLACE.

| | Vectorization | | | Neural | | Euler | | PLACE (ours) | |
| --- | --- | --- | --- | --- | --- | --- | --- | --- | --- |
| | PI | SW-K | PF-K | PersLay | Persformer | ECS+XGB | HT2+XGB | linear SVM | NC |
| Acc. (%) | $82.5^{\ddagger}$ | $83.6^{\ddagger}_{\pm 0.9}$ | $85.9^{\dagger}_{\pm 0.8}$ | $87.7_{\pm 1.0}$ | $91.2^{\ddagger}_{\pm 0.8}$ | $\mathbf{91.8_{\pm 0.4}}^{\ddagger}$ | $89.9^{\ddagger}_{\pm 0.5}$ | $87.2_{\pm 0.6}$ | $33.9_{\pm 1.5}$ |

only up to that selection gap and trails on COX2. On the remaining datasets PLACE underperforms the strongest topology-based baseline at $p < 0.01$; the gaps fall into two groups. The NCI1/NCI109 gap ($\sim$ 14–17 pp below WKPI) reflects a fundamental limitation: these datasets are discriminated by discrete node labels (atom types), which our continuous structural descriptors cannot capture (*descriptor blindness*; Section 6.3). On PROTEINS, DD, IMDB-B, IMDB-M, PTC, DHFR, and REDDIT-5K, PLACE is 5–13 pp below the strongest baseline; here the embedding structure exposes some signal but the descriptor–$\tau^*$ interaction is harder to navigate within our small homogeneous pool, and the top accuracy on the pool is below what the descriptors and pooling enriched in WKPI-kC and RetGK extract. Table 6 reports per-dataset diagnostics for the three certificate forms of Theorem 5.1. All three radii are carried in their honest finite-sample form and tested against the empirical firing condition $r_m < \frac{1}{4}\hat{\Delta}$ (Proposition 3.2). The non-asymptotic Pinelis radius $r_m^{\mathrm{Pin}} = 2R\sqrt{2\log(2k/\alpha)/m}$ is dominated by $R^2$, and the asymptotic Gaussian radius $\tilde{r}_m^{\mathrm{G}} = \sqrt{\|\hat{\Sigma}_c\|_{\mathrm{op}}\,\chi^2_{\ell,\,\alpha/k}/m_c}$ carries a $\sqrt{\chi^2_\ell}$ dimension penalty ($\ell \in [93, 6{,}539]$); both fail on every benchmark. The variance-aware Pinelis–Bernstein radius of (iii), carrying its linear Bernstein term (5.4), clears $r_m^{\mathrm{vP}} < \frac{1}{4}\hat{\Delta}$ on only three benchmarks—NCI1 and NCI109 on 100% of folds (their large per-class samples, $m_{\min} \approx 1{,}845$, overcome a small separation) and DD on 93% (its large class separation $\hat{\Delta} \approx 8.2$ makes the threshold easy to clear at $m_{\min} = 438$). Because $r_m^{\mathrm{vP}} = O(m^{-1/2})$, every benchmark would clear the condition at a large enough per-class sample; on the others $m$ simply sits below the honest threshold $m_c^{*,\,\mathrm{vP}}$ (e.g. MUTAG $m_c^{*,\,\mathrm{vP}} \approx 280$ against $m_{\min} = 57$), so the split in Table 6 is a sample-size effect that more data would close, not a structural barrier. Firing attests only that the empirical and population NC rules *agree*, never that either is *correct*. Indeed, across all twelve benchmarks the within-class radius satisfies $D_c^2 \geq \|\Sigma_c\|_{\mathrm{op}} > \hat{\Delta}_c^2/4$, so $D_c > \hat{\Delta}_c/2$ and Proposition 3.3's linear-separability hypothesis fails: the *population* nearest-centroid rule itself misclassifies some test points on every benchmark—DD, for instance, fires yet is population-non-separable. These datasets are linearly separable (Section 6.2 linear-SVM accuracies $\geq 70\%$) but not nearest-centroid separable; we therefore report linear-SVM accuracies in Table 10 and read Table 6's vP column as a heuristic diagnostic of sample/population NC agreement (Remark 5.3), not of correctness. On MUTAG the honest radius does *not* fire at $m_{\min} = 57$ ($m_c^{*,\,\mathrm{vP}} \approx 280$), yet the empirical NC predictions still agree with the population NC rule—operationalized by the full-sample centroids as defined in Section 5—on every one of the 940 held-out test predictions ($85.0 \pm 8.4\%$ accuracy), Clopper–Pearson 95% lower bound on coverage $\geq 0.984$: direct empirical evidence of the agreement the diagnostic targets, in a regime where the worst-case radius has not yet certified it (Remark 5.3).

**Empirical scope of $\nu$-coherence.** Proposition 2.1(b)'s lower distortion bound and Corollary 3.1's $\lambda$-anchored rate are stated under $\nu$-coherence (Definition 2.1): the per-scale block-norm $\|\Phi_{R_k}(A) - \Phi_{R_k}(B)\|^2_{\ell^2} \geq R_k^2/32$ at every active scale $k$. For each cross-class pair with $d_{\mathcal{B}}(A, B) \geq 3R_1$ we compute the optimal bottleneck matching via binary search over edge-weight thresholds, augment the diagrams to common cardinality through diagonal-projection partners (per the $\mathcal{D}_n$ convention), and check the per-scale floor against the standard PLACE configuration's per-scale landmark grids. Reproduction: `experiments/exp_pi_coherence_audit.py`.

Table 7 shows that $\nu$-coherence is empirically near-tight: $\geq 99.7\%$ of qualifying pairs across all four benchmarks (100.0% on three of them). Combined with the certificate-conclusion audit below (Table 8, 100% on the same pairs), the residual gap is the deterministic concave-majorant slack introduced when summing the per-scale step-floors into a single $d_{\mathcal{B}}$-proportional Lipschitz statement, rather than any structural slack.

Table 6: Certificate firing diagnostics for nearest-centroid classification under Theorem 5.1 ($\alpha = 0.05$), computed with the *honest* finite-sample radii and the empirical firing condition $r_m < \frac{1}{4}\hat{\Delta}$ (Proposition 3.2: certifying $r_m < \frac{1}{2}\Delta$ from the empirical $\hat{\Delta}$ needs $\hat{\Delta} > 4r_m$). Pinelis uses $r_m^{\mathrm{Pin}} = 2R\sqrt{2\log(2k/\alpha)/m}$; the variance-aware Pinelis–Bernstein radius carries its full linear term (5.4), $r_m^{\mathrm{vP}} = \sqrt{2\operatorname{tr}(\hat{\Sigma}_c)\log(2k/\alpha)/m_c} + 4R\log(2k/\alpha)/(3m_c)$ (with $\operatorname{tr}(\hat{\Sigma}_c) \approx \|\hat{\Sigma}_c\|_{\mathrm{op}}$); Gauss uses $\tilde{r}_m^{\mathrm{G}} = \sqrt{\|\hat{\Sigma}_c\|_{\mathrm{op}}\,\chi^2_{\ell,\,\alpha/k}/m_c}$. Radii are per-fold medians; Fire % is the fraction of the 50 (5 seeds × 10 folds) satisfying $r_m < \frac{1}{4}\hat{\Delta}_c$. Only the three large-sample benchmarks NCI1, NCI109, DD clear the honest condition; the datasets that appeared to fire under the earlier $\frac{1}{2}\hat{\Delta}$ threshold with the linear term dropped (MUTAG, PROTEINS, DHFR, REDDIT-5K, IMDB-B) no longer do (Remark 5.3). Reproduction: `experiments/regen_cert_firing_table_honest.py`.

| Dataset | Filt | $m_{\min}$ | $r_m^{\mathrm{Pin}}$ | $r_m^{\mathrm{vP}}$ | $\tilde{r}_m^{\mathrm{G}}$ | $\hat{\Delta}/4$ | Pin fire | vP fire | Gauss fire |
|---|---|---|---|---|---|---|---|---|---|
| MUTAG | deg+HKS$_{10}$ | 57 | 5.26 | 1.036 | 5.09 | 0.392 | 0% | 0% | 0% |
| PROTEINS | deg+ricci | 405 | 3.56 | 0.481 | 8.69 | 0.456 | 0% | 0% | 0% |
| NCI1 | HKS$_{10}$ | 1848 | 0.14 | 0.015 | 0.18 | 0.023 | 0% | **100%** | 0% |
| NCI109 | HKS$_{10}$ | 1843 | 0.14 | 0.015 | 0.14 | 0.023 | 0% | **100%** | 0% |
| DHFR | HKS$_{10}$ | 265 | 0.57 | 0.073 | 0.23 | 0.023 | 0% | 0% | 0% |
| DD | degree | 438 | 10.70 | 1.794 | 15.25 | 2.052 | 0% | **93%** | 0% |
| REDDIT-5K | closeness | 899 | 0.84 | 0.106 | 0.62 | 0.030 | 0% | 0% | 0% |
| COX2 | jaccard+HKS$_{10}$ | 92 | 0.54 | 0.086 | 0.23 | 0.005 | 0% | 0% | 0% |
| PTC | deg+betw | 137 | 9.06 | 1.659 | 5.75 | 0.125 | 0% | 0% | 0% |
| IMDB-B | degree | 450 | 14.57 | 1.466 | 12.84 | 0.203 | 0% | 0% | 0% |
| IMDB-M | betw+ricci | 450 | 1.13 | 0.123 | 0.22 | 0.007 | 0% | 0% | 0% |
| Orbit5k | alpha $H_1$ | 700 | 0.21 | 0.025 | 0.23 | 0.001 | 0% | 0% | 0% |

Table 7: Empirical $\nu$-coherence audit on the chemical graph datasets at the per-dataset headline filtration (Table 9), top-$N_{\max} = 50$ persistence filter, 2,000 sampled cross-class pairs per dataset restricted to $d_{\mathcal{B}}(A, B) \geq 3R_1$. **coherent %**: fraction satisfying the per-scale aggregate floor $\|\Phi_{R_k}(A) - \Phi_{R_k}(B)\|_{\ell^2}^2 \geq R_k^2/32$ at every active scale (Definition 2.1). Reproduction script: `experiments/exp_pi_coherence_audit.py`.

| Dataset | Filt | $n_\tau$ | coherent % |
|---|---|---|---|
| MUTAG | deg+HKS$_{10}$ | 1,943 | 100.0 |
| PTC | deg+betw | 1,959 | 99.7 |
| COX2 | jaccard+HKS$_{10}$ | 578 | 100.0 |
| DHFR | HKS$_{10}$ | 994 | 100.0 |

**The certificate's conclusion.** The sharp certificate of Proposition 2.1(b), $\|\Phi(A) - \Phi(B)\|_{\ell^2} \geq \frac{1}{16}\sqrt{\sum_{k:\,3R_k \leq d_{\mathcal{B}}(A,B)} w_k^2 R_k^2}$, holds on every qualifying pair we tested. For each dataset we build the standard PLACE multiscale embedding via `init_from_dataset` ($N = 5$ scales, analytic-optimal masses, $L$ auto-detected from the diagrams). For each cross-class pair with $d_{\mathcal{B}}(A, B) \geq 3R_1$ we measure $\|\Phi(A) - \Phi(B)\|_{\ell^2}$ and the ratio $\|\Phi(A) - \Phi(B)\|_{\ell^2}/\sigma(d_{\mathcal{B}}(A, B))$ where $\sigma(t) = \frac{1}{16}\sqrt{\sum_{k:\,3R_k \leq t} w_k^2 R_k^2}$ is the right-hand side of (2.8). Reproduction: `experiments/exp_pi_certificate_bound_audit.py`.

The sharp certificate holds on 100% of qualifying pairs across all four datasets, with median ratios in the 338–1436 range and minima in the 11–57 range. The slack between $\|\Phi(A) - \Phi(B)\|_{\ell^2}$ and the right-hand side of (2.8) reflects how far the per-scale block-norms exceed the floor $R_k^2/32$ on real chemical-graph diagrams: even the worst-case minimum ratio of 10.8 on PTC corresponds to per-scale blocks roughly an order of magnitude above the floor. In combination with Table 7, this shows that $\nu$-coherence is essentially tight as a hypothesis: it holds on $\geq 99.7\%$ of pairs and the per-scale floor it asserts is the exact mechanism driving the certificate.

Table 8: Empirical sharp-certificate audit on chemical graph datasets at the per-dataset headline filtration. Standard PLACE configuration, $N = 5$ scales, analytic-optimal masses. $n_\tau$: cross-class pairs with $d_{\mathcal{B}}(A, B) \geq 3R_1$. **bound %**: fraction of these pairs with $\|\Phi(A) - \Phi(B)\|_{\ell^2} \geq \sigma(d_{\mathcal{B}}(A, B))$ (the sharp certificate of Proposition 2.1(b)). **p25 / p50 / p75**: percentiles of the ratio $\|\Phi(A) - \Phi(B)\|_{\ell^2}/\sigma(d_{\mathcal{B}}(A, B))$. **min**: smallest ratio observed.

| Dataset | Filt | $n_\tau$ | bound % | p25 | p50 | p75 | min |
|---|---|---|---|---|---|---|---|
| MUTAG | deg+HKS$_{10}$ | 1,943 | 100.0 | 845.2 | 1435.5 | 2141.2 | 56.9 |
| PTC | deg+betw | 1,959 | 100.0 | 139.0 | 337.8 | 573.8 | 10.8 |
| COX2 | jaccard+HKS$_{10}$ | 578 | 100.0 | 189.7 | 415.9 | 690.8 | 33.3 |
| DHFR | HKS$_{10}$ | 994 | 100.0 | 226.7 | 441.3 | 748.4 | 18.7 |

Table 9: Best $(f, \tau^*, N)$ configuration per graph dataset within the committed candidate pool (15 descriptors $\times$ {proxy, crossing} $\times$ $N \in \{5, 10, 15, 20\} = 120$ configurations), selected by mean training-fold accuracy. Acc. is mean $\pm$ s.d. over 5 seeds $\times$ 10 folds.

| Dataset | Best descriptor | $\tau^*$ | $N$ | $\hat{\eta}$ | Acc. (%) |
|---|---|---|---|---|---|
| MUTAG | deg+HKS$_{10}$ | proxy | 5 | 0.0036 | $88.4_{\pm 7.9}$ |
| PROTEINS | deg+ricci | crossing | 5 | 0.1013 | $71.5_{\pm 4.3}$ |
| NCI1 | HKS$_{10}$ | proxy | 10 | 0.0018 | $71.3_{\pm 1.9}$ |
| COX2 | jaccard+HKS$_{10}$ | proxy | 10 | 0.0012 | $80.0_{\pm 3.8}$ |
| DHFR | HKS$_{10}$ | crossing | 20 | 0.0054 | $77.6_{\pm 4.9}$ |
| PTC | deg+betw | proxy | 5 | 0.0430 | $59.3_{\pm 7.4}$ |
| DD | degree | proxy | 5 | 0.2774 | $76.3_{\pm 3.4}$ |
| IMDB-B | degree | proxy | 5 | 0.0201 | $66.4_{\pm 4.3}$ |
| IMDB-M | betw+ricci | crossing | 5 | 0.0001 | $44.5_{\pm 3.6}$ |
| NCI109 | HKS$_{10}$ | proxy | 10 | 0.0017 | $70.6_{\pm 2.7}$ |
| REDDIT-5K | closeness | proxy | 10 | 0.0047 | $46.2_{\pm 2.0}$ |

## 6.3 Descriptor Selection

Descriptor selection produces 6–14 percentage point swings, far exceeding the effect of scale count or mass choice. We compare three closed-form selectors that require no classifier training—embed once per descriptor, evaluate the statistic, pick the maximizer:

- the *Mahalanobis margin* $\hat{\rho}_{\text{Mah}}$ of equation (4.1), the LDA Bayes-margin form of the Fisher ratio (Remark 4.1), with $\hat{\Sigma}_{\text{LW}}$ the Ledoit–Wolf-shrunk pooled within-class covariance;

- the direct rate-determining ratio $\hat{\Delta}/\hat{R}$ of Corollary 3.1, where $\hat{R} := \sup_A \|\Phi(A)\|$;

- the isotropic surrogate $\hat{\eta} := \hat{\Delta}/\sqrt{\ell}$ (Proposition 4.1), which equals $\hat{\Delta}/\hat{R}$ up to the loose substitution $\hat{R} \leq B\sqrt{\ell}$ and is consistent under coordinate-isotropic covariance.

On Orbit5k, $\hat{\eta}$ identifies alpha $H_1$ with a $2\times$ gap and the ranking is stable under both $\tau^*$ estimators (Table 4); the chemical pool below exhibits the heterogeneous regime in which $\hat{\rho}_{\text{Mah}}$ takes over as the dominant selector.

For the chemical benchmarks we built a heterogeneous pool of 64 candidate descriptors (degree, betweenness, closeness, edge-betweenness, six HKS timescales, Ollivier–Ricci, and all-by-all pair combinations), restricted per dataset to the sub-pool on which all three statistics are well-defined (50–55 candidates depending on which descriptors have $R > 0$ on that dataset). Spearman rank correlations of each statistic against linear SVM accuracy, averaged over 5 seeds $\times$ 10 folds at $N = 10$ scales with the proxy $\tau^*$ estimator, are in Table 11.

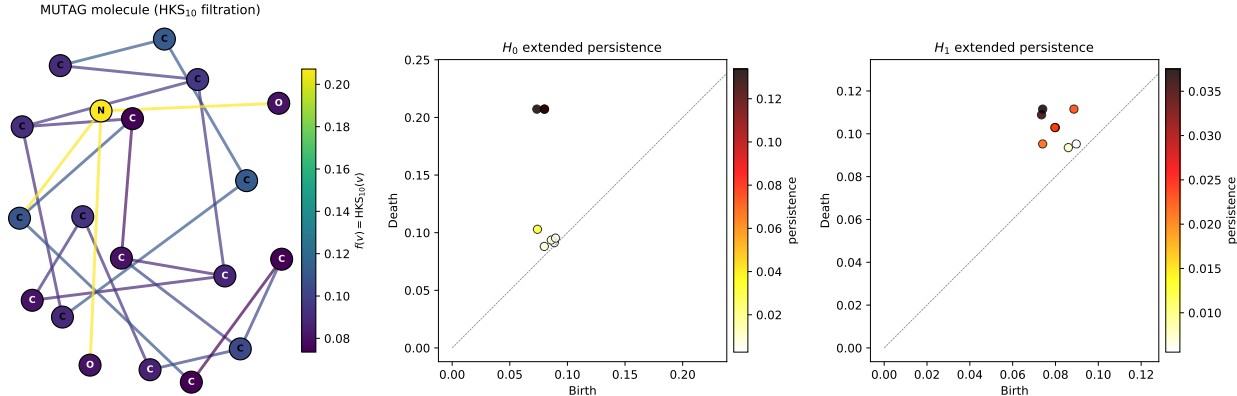

Figure 6: Graph-to-diagram pipeline on a MUTAG molecule: HKS filtration (left), $H_0$ and $H_1$ extended persistence diagrams (right).

Table 10: Graph classification accuracy (%, 10-fold CV). The primary PLACE column, **Mah (cf.)**, is the *closed-form* Mahalanobis selector's end-to-end linear-SVM accuracy (mean ± s.d. over 5 seeds × 10 folds): on each training fold $\hat{\rho}_{\mathrm{Mah}}$ picks one of the 15 committed descriptors at the canonical config ($\tau^*$=proxy, $N$=10), and we report the held-out accuracy of that pick, with *no held-out calibration* (`experiments/selector_ablation_committed.py`). **oracle** is the in-pool best-of-120-configuration, train-selected upper bound of Table 9; the closed-form pick attains it within 0–5 pp. **Bold** = best topology-based baseline in row; — = not reported/pending. Significance superscripts ([†] $p < 0.05$, [‡] $p < 0.01$; one-sample/Welch's $t$-test, $n = 50$) are computed against the PLACE *oracle*, not the closed-form headline; since closed-form ≤ oracle, a baseline marked above the oracle is a fortiori above the closed-form selector. These markers are descriptive indicators, not firm paired significance (see Protocol). NC = empirical nearest-centroid accuracy on the same descriptor, reported only for MUTAG, where the 940/940 empirical/population NC agreement is observed (Section 5; not a fired certificate under the honest quarter-threshold, Remark 5.3).

| | PLACE (ours) | | | Topology-based | | | | Graph | | |
| Dataset | Mah (cf.) | oracle | NC | WKPI-kM | WKPI-kC | PersLay | ECP | RetGK | GIN | Filt. |
| --- | --- | --- | --- | --- | --- | --- | --- | --- | --- | --- |
| MUTAG | $87.0_{\pm7.3}$ | 88.4 | $85.0_{\pm8.4}$ | $85.8^{\dagger}$ | 88.3 | 89.8 | **90.0** | 90.3 | 90.0 | deg+hks10 |
| PROTEINS | $66.5_{\pm4.1}$ | 71.5 | — | **$78.5^{\ddagger}$** | $75.2^{\ddagger}$ | $74.8^{\ddagger}$ | $75.0^{\ddagger}$ | $75.8^{\ddagger}$ | $76.2^{\ddagger}$ | deg+ricci |
| NCI1 | $70.7_{\pm2.4}$ | 71.3 | — | **$87.5^{\ddagger}$** | $84.5^{\ddagger}$ | $73.5^{\ddagger}$ | $76.3^{\ddagger}$ | $84.5^{\ddagger}$ | $82.7^{\ddagger}$ | hks10 |
| COX2 | $76.6_{\pm6.0}$ | 80.0 | — | — | — | 80.9 | 80.3 | **81.4** | — | jaccard+hks10 |
| DHFR | $75.7_{\pm5.4}$ | 77.6 | — | — | — | $80.3^{\ddagger}$ | **$82.0^{\ddagger}$** | $81.5^{\ddagger}$ | — | hks10 |
| PTC | $55.3_{\pm9.7}$ | 59.3 | — | $62.7^{\ddagger}$ | **$68.1^{\ddagger}$** | — | — | $62.5^{\ddagger}$ | $66.6^{\ddagger}$ | deg+betw |
| DD | $73.4_{\pm5.5}$ | 76.3 | — | **$82.0^{\ddagger}$** | $80.3^{\ddagger}$ | — | — | $81.6^{\ddagger}$ | — | deg |
| IMDB-B | $65.3_{\pm5.1}$ | 66.4 | — | $70.7^{\ddagger}$ | **$75.1^{\ddagger}$** | $71.2^{\ddagger}$ | $73.3^{\ddagger}$ | $71.9^{\ddagger}$ | $75.1^{\ddagger}$ | deg |
| IMDB-M | $44.0_{\pm3.2}$ | 44.5 | — | $46.4^{\ddagger}$ | **$49.5^{\ddagger}$** | $48.8^{\ddagger}$ | $48.7^{\ddagger}$ | $47.7^{\ddagger}$ | $52.3^{\ddagger}$ | betw+ricci |
| NCI109 | $68.6_{\pm2.2}$ | 70.6 | — | $85.9^{\ddagger}$ | **$87.4^{\ddagger}$** | — | — | — | — | hks10 |
| REDDIT-5K | $45.1_{\pm1.6}$ | 46.2 | — | $59.1^{\ddagger}$ | **$59.5^{\ddagger}$** | — | — | $56.1^{\ddagger}$ | $57.5^{\ddagger}$ | closeness |

Beyond rank-correlation, Table 12 reports the *end-to-end* PLACE-linear accuracy attained by each closed-form selector under honest per-fold selection, with the in-pool oracle as an upper bound. The Mahalanobis selector is the strongest closed-form rule (67.4% mean, vs. isotropic 64.0% and random 65.6%) but trails the oracle by 7.3pp on average—so the oracle is an upper bound, not the operating point. On NCI1/NCI109 the selector–oracle gap is small (+3.4–4.6pp) yet even the oracle caps near 73%, locating the limit in descriptor blindness (pool coverage) rather than selection.

Three patterns emerge. *(i)* The Mahalanobis margin $\hat{\rho}_{\mathrm{Mah}}$ has the strongest mean correlation (+0.56 across the 11 datasets) and is positive on 10 of 11 (PTC the lone outlier at −0.24, borderline non-significant at $p = 0.08$); its high values on MUTAG (+0.84), DHFR (+0.89), NCI1 (+0.79), NCI109 (+0.79), IMDB-

Table 11: Spearman rank correlation between each closed-form selection statistic and linear SVM accuracy, across 11 benchmarks (per-dataset pool size in the rightmost column; full 5 seeds × 10 folds, $N = 10$ scales, proxy $\tau^*$, with the corrected $\lambda(\nu)$ weight rule of equation (2.13)). The winner column lists the best descriptor by linear accuracy and, in parentheses, its rank under each statistic (# out of the pool size; lower is better).

| Dataset | $\rho(\hat{\rho}_{\mathrm{Mah}})$ | $\rho(\hat{\Delta}/\hat{R})$ | $\rho(\hat{\eta})$ | Winner (rank by Mah, $\hat{\Delta}/\hat{R}$, $\hat{\eta}$; pool) |
|---|---|---|---|---|
| MUTAG | **+0.84** | +0.63 | −0.39 | $hks_2 + hks_{25}$ (**2**, 16, 37; 51) |
| COX2 | **+0.27** | −0.19 | −0.05 | nodelabel+$hks_1$ (59, 28, 56; 60) |
| DHFR | **+0.89** | +0.16 | −0.70 | $hks_{0.1} + hks_{10}$ (**3**, 16, 50; 53) |
| PTC | −0.24 | −0.19 | **+0.35** | deg+betw (34, 11, **2**; 55) |
| NCI1 | **+0.79** | +0.02 | −0.38 | $hks_{0.1} + hks_{10}$ (53, 56, 59; 60) |
| NCI109 | **+0.79** | +0.09 | −0.42 | $hks_{0.1} + hks_{10}$ (47, 56, 59; 60) |
| PROTEINS | +0.37 | **+0.70** | +0.63 | deg+betw (7, **2**, **2**; 15) |
| DD | +0.38 | −0.25 | **+0.49** | deg+ricci (**3**, 10, 5; 15) |
| IMDB-B | **+0.63** | +0.15 | −0.09 | $hks_{t10}$ (**2**, 7, 12; 14) |
| IMDB-M | **+0.74** | +0.50 | −0.39 | deg+$hks_{10}$ (**1**, 4, 11; 14) |
| REDDIT-5K | **+0.71** | +0.20 | −0.24 | closeness (3, **1**, 11; 15) |
| Mean | **+0.56** | +0.16 | −0.11 | — |

Table 12: Closed-form descriptor-selector ablation (Fxy4). End-to-end PLACE-linear accuracy (%, mean±std over 5 × 10 folds) under per-fold selection: the selector picks argmax of its statistic on each training fold; accuracy is on the held-out test fold. Random is the expected accuracy of a uniform pick; oracle is the in-pool test-selected upper bound.

| Dataset | Mahalanobis | Isotropic $\Delta/\sqrt{\ell}$ | Random | Oracle | Mah−Oracle |
|---|---|---|---|---|---|
| MUTAG | 84.1 | 78.1 | 80.5 | 94.6 | +10.5 |
| COX2 | 78.7 | 78.2 | 78.8 | 87.0 | +8.3 |
| DHFR | 74.7 | 60.1 | 69.0 | 82.1 | +7.4 |
| PTC | 50.1 | 52.2 | 53.6 | 69.2 | +19.1 |
| NCI1 | 68.7 | 63.8 | 65.0 | 73.2 | +4.6 |
| NCI109 | 69.2 | 63.4 | 63.9 | 72.5 | +3.4 |
| PROTEINS | 66.8 | 69.7 | 67.1 | 74.4 | +7.6 |
| DD | 73.4 | 72.1 | 73.9 | 78.3 | +4.9 |
| IMDB-B | 64.7 | 62.7 | 62.9 | 69.8 | +5.1 |
| IMDB-M | 43.8 | 39.9 | 41.2 | 46.1 | +2.3 |

M (+0.74), REDDIT-5K (+0.71), and IMDB-B (+0.63) confirm empirically that the LDA Bayes margin under Ledoit–Wolf shrinkage is the principled selector predicted by Remark 4.1, and that the chemical-pool finding extends to label-dominated (NCI1, NCI109), large-graph social (REDDIT-5K, IMDB-B/M), and protein-structure (PROTEINS, DD) regimes. *(ii)* The direct ratio $\hat{\Delta}/\hat{R}$ is a useful secondary signal—it agrees with Mahalanobis on MUTAG and is the top selector by winner-rank on PROTEINS and REDDIT-5K—but its sign reverses on COX2, PTC, and DD, reflecting that $\hat{R}$ alone does not capture anisotropic class-conditional covariance. *(iii)* The isotropic surrogate $\hat{\eta}$ is reliable on homogeneous pools (Orbit5k, 14 descriptors, $\rho = +0.65$; PROTEINS, 15 descriptors, +0.63; DD, 15 descriptors, +0.49) but breaks down on the heterogeneous chemical pools ($\rho \in [−0.70, −0.05]$ on MUTAG/COX2/DHFR/NCI1/NCI109) where the $\sqrt{\ell}$ penalty over-charges high-dimensional HKS descriptors, pulling the cross-dataset mean to −0.11. PTC is the chemical outlier where $\hat{\eta}$ ranks the winner at #2 while Mahalanobis ranks it at #34; inspecting the pool shows that PTC's signal lives in low-dimensional structural edge-betweenness features where the $\sqrt{\ell}$ penalty happens to align with accuracy. On the strength of the mean correlations and the top-of-pool rankings on MUTAG, DHFR, and IMDB-B we recommend $\hat{\rho}_{\mathrm{Mah}}$ as the default closed-form selection rule and report all three statistics together as diagnostics: agreement between $\hat{\rho}_{\mathrm{Mah}}$ and $\hat{\Delta}/\hat{R}$ is the strongest practitioner-level signal, and large disagreement with $\hat{\eta}$ flags the pool-heterogeneity regime in which the

isotropic surrogate breaks down. Betweenness and degree descriptors consistently rank highly across all three criteria, as do their pair-combinations with spectral (HKS) features.

**Two regimes, two selectors.** Table 10 reports PLACE accuracy on the best $(f, \tau^*, N)$ configuration in a committed 15-descriptor candidate pool—an in-pool oracle that the closed-form $\hat{\rho}_{\text{Mah}}$ approximates within $\sim 3$ pp on the four chemical datasets where we have direct 15-pool sweeps, and that $\hat{\eta}$ approximates much less reliably (the surrogate hypothesis behind $\hat{\eta}$ is only mild when the pool is structurally homogeneous, and our 15-pool is borderline). Table 11, in contrast, evaluates the selectors on 11 benchmarks with full $5 \times 10$ seed–fold sweeps under the corrected $\lambda(\nu)$ weights, covering both heterogeneous chemical pools (50–60 descriptors, mixing HKS at multiple timescales, node-label-aware combinations, Ollivier–Ricci variants, and centrality measures) and homogeneous protein/social pools (14–15 descriptors). The split between the two regimes is sharp: on heterogeneous chemical pools, $\hat{\eta}$ breaks down (mean $\rho \in [-0.70, -0.05]$ across MUTAG/COX2/DHFR/NCI1/NCI109) while $\hat{\rho}_{\text{Mah}}$ remains positive on every chemical dataset except PTC; on the homogeneous PROTEINS, DD, and Orbit5k pools, $\hat{\eta}$ recovers ($\rho \geq +0.49$) and its closed-form consistency rate (Proposition 4.1) applies. Aggregating across all 11 benchmarks, $\hat{\rho}_{\text{Mah}}$ has mean $\rho = +0.56$, positive on 10 of 11; $\hat{\Delta}/\hat{R}$ has mean $+0.16$; $\hat{\eta}$ has mean $-0.11$. We therefore recommend $\hat{\rho}_{\text{Mah}}$ as the default selection rule for new datasets where the candidate pool is heterogeneous or large, and retain $\hat{\eta}$ for the structurally homogeneous regime. Accordingly, Table 10's headline is the closed-form $\hat{f}_{\text{Mah}}$ end-to-end accuracy, with the in-pool oracle shown alongside as an upper bound. The closed-form pick matches the oracle within $\sim 3$ pp on MUTAG, NCI1, NCI109, IMDB-M, IMDB-B, and REDDIT-5K, and trails by 3–5 pp on PROTEINS, COX2, DHFR, PTC, and DD, where the accuracy-winner sits deeper in the $\hat{\rho}_{\text{Mah}}$ ranking (Table 11); the closed-form selector is thus a label-free operating point a few points below the oracle, not yet a complete substitute for it on every benchmark.

The central finding is that the entire pipeline—descriptor ranking, classifier, and per-prediction certificate—can be fixed analytically from the same two embedding-level quantities, the class-mean separation $\Delta$ and the radius $R$. For *descriptor ranking*, the Mahalanobis margin $\hat{\rho}_{\text{Mah}}$ between class means under Ledoit–Wolf-shrunk pooled covariance is the LDA Bayes-margin form of the Fisher discriminant ratio (Remark 4.1) and is empirically the strongest closed-form ranker we tested on the chemical pool; $\hat{\eta} = \hat{\Delta}/\sqrt{\ell}$ is its isotropic Fisher-ratio-bound surrogate, which carries a closed-form selection-consistency rate (Proposition 4.1, Corollary 4.1). For *classification*, the $O((k-1)R/(\Delta\sqrt{m_{\min}}))$ margin bound of Theorem 3.1 is driven by the same $\Delta$ and $R$, so a linear SVM on the embedding $\Phi$ is already near-optimal on every benchmark on which the descriptor pool exposes the discriminative signal—the embedding, not the classifier, does the work. The remaining gaps—NCI1, NCI109, and DD—are *descriptor-blindness* failures (no candidate descriptor in our pool achieves $\Delta > 0$ against the discrete-node-label signal that drives those datasets); the embedding machinery is not the bottleneck. The summation pooling retains the key property that max-pooled alternatives (Mitra–Virk's $n$-fold composition, deep-set pooling) lose: linearity in the empirical diagram measure, which makes $\Delta$ a well-behaved statistical object and grounds the stability theorem of Section 2.

The theory breaks when $\Delta \to 0$: neither the upper bound (Theorem 3.1) nor the certificate (Theorem 5.1) remains informative. Two distinct causes are in play. *Intrinsic indistinguishability* obtains when the class-conditional diagram measures agree in bottleneck distance, and no vectorization can separate them; Proposition 3.1 makes this diagnosable on PLACE, since $\Delta \approx 0$ together with bounded $\max_c D_c$ imply $\lambda(\nu)\,\delta_*$ is small—a statement about the data itself, not about the embedding. *Descriptor blindness* obtains when the descriptor itself fails to expose the structural difference; the diagnostics $\hat{\rho}_{\text{Mah}}$, $\hat{\Delta}/\hat{R}$, and $\hat{\eta}$ in Section 6.3 all flag this case by collapsing to near-zero for every candidate in a failing pool. NCI1/NCI109 exemplify the second case: the structural descriptors in our pool achieve $\Delta > 0$, but the discriminative signal is dominated by discrete node labels our continuous descriptors cannot access.

What the Mahalanobis margin catches and the isotropic surrogate misses is anisotropy of the class-conditional covariance. The closed-form ratio $\hat{\eta} = \hat{\Delta}/\sqrt{\ell}$ implicitly treats every coordinate of the embedding as carrying equal class-conditional variance, so a high-dimensional descriptor with most coordinates redundant is over-penalized by the $\sqrt{\ell}$ factor. HKS at multiple timescales is the canonical example: the embedding has many coordinates but a small number of effective directions in which the class means actually separate, and the Ledoit–Wolf-shrunk Mahalanobis margin recovers the right ranking by reweighting along those low-variance

directions. This is precisely the regime where the LDA Bayes margin and the isotropic Fisher-ratio lower bound diverge by a large factor (Remark 4.1).

WKPI (Zhao & Wang, 2019) and PersLay (Carrière et al., 2020) learn the weighting of a fixed feature bank; PLACE holds the weighting fixed and instead places a larger, sparse landmark grid. Empirically PLACE's in-pool oracle matches the strongest topology-based baseline on MUTAG and COX2 and underperforms by 5–17 pp on the remaining graph datasets, with the closed-form selector a mean 2.2 pp below the oracle; the trade-off is that PLACE's grid is analytically fixed, so $\Delta$ and $r_m$ are estimable from training data alone—the condition under which a closed-form descriptor ranking and a per-prediction certificate are available at all. Closing the accuracy gap on PROTEINS, DD, IMDB-B/M, PTC, and REDDIT-5K through a richer candidate pool (and data-adaptive landmark placements) is an open direction.

**Limitations.**

- Certificates apply only to the nearest-centroid classifier, which achieves lower accuracy than SVM. The non-asymptotic Pinelis radius is dominated by the $L^2$ envelope $4R^2$ and fails on every benchmark; the asymptotic Gaussian plug-in radius carries a $\sqrt{\chi^2_{\ell,\alpha/k}}$ dimension penalty that also fails at our $\ell$. Carried in its honest finite-sample form and tested at the empirical threshold $r_m < \frac{1}{4}\hat{\Delta}$, the variance-aware Pinelis–Bernstein radius (iii) clears the condition on only 3 benchmarks (NCI1, NCI109, DD; Table 6); on the smaller ones the per-class sample sits below the honest threshold $m_c^{*,\mathrm{vP}}$, a sample-size effect rather than a structural one, since $r_m^{\mathrm{vP}} = O(m^{-1/2})$. Crucially, certificate firing guarantees only that the empirical and population NC rules agree; it does not guarantee that either is correct on test data—across all twelve benchmarks $D_c > \hat{\Delta}_c/2$, so the population NC rule itself is non-separable (DD fires yet is non-separable). NCI1 and NCI109, where the radius fires yet linear-SVM accuracy trails the strongest baseline by 14–17 pp, exemplify this: the population NC rule fails when the descriptor pool is blind to the discrete-node-label class signal (cf. Remark 5.1).

- The top-$k$ persistence filter is a heuristic; without it, low-persistence features near the diagonal dominate the embedding and inflate $\ell$ without commensurate gain in $\Delta$.

- Descriptor selection is empirically driven by the Mahalanobis margin $\hat{\rho}_{\mathrm{Mah}}$, but its consistency theorem assumes the population pooled covariance is well-conditioned; the closed-form rate (Proposition 4.1) is established only for the isotropic surrogate $\hat{\eta} = \hat{\Delta}/\sqrt{\ell}$, and a fully data-driven rate for the Ledoit–Wolf-shrunk estimator is open (cf. the shrinkage and regularized-covariance literature, Bickel & Levina, 2008; Ledoit & Wolf, 2004). Whether the upper-bound ranking coincides with true accuracy ranking in general is not proved.

- The NCI1/NCI109 gap ($\sim 16\,\mathrm{pp}$, $p < 0.01$ against every reported baseline) reflects descriptor blindness to discrete node labels, not a deficiency of the embedding machinery.

**Future work.** Data-adaptive learning of landmark positions on this same embedding family, replacing the heuristic grid $\mathbb{G}_R$, is left to subsequent work. A full sample-complexity theory—CLT, Berry–Esseen rates, and Donsker-type functional CLTs for the landmark embedding—is a natural next step but is beyond the present paper's scope. A further open direction is a measure-theoretic foundation replacing the discrete grid with a Bochner integral over a continuous landmark configuration $\Lambda$, admitting adaptive grids, overcomplete families, and kernel-smoothing variants.

## A Notation

Table 13 collects the symbols used throughout, grouped by role. Single-scale landmark radii are always subscripted $(R_1, \ldots, R_N)$ to distinguish them from the embedding radius $R$.

Table 13: Summary of notation.

| Symbol | Meaning |
|---|---|
| *Diagrams and embedding (Section 2)* | |
| $\mathcal{D}, \mathcal{D}_n$ | space of finite persistence diagrams; those with $n$ points |
| $d_{\mathcal{B}}$ | bottleneck distance on diagrams |
| $L$ | compact-support size, $\mathcal{T}_L = \{0 \le b < d \le L\}$ |
| $N_{\max}$ | top-persistence cardinality cap on diagrams |
| $\mathbb{G}_R, M$ | landmark grid at scale $R$; its size $M = |\mathbb{G}_R^+|$ |
| $\varphi_{R,p}$ | hat coordinate at landmark $p$, height $3R/2$ |
| $\Phi_R, \Phi$ | single-scale (2.2) and multiscale (2.3) embedding |
| $N, R_k, w_k$ | number of scales; scale radii; scale weights ($\sum_k w_k^2 = 1$) |
| $\nu = \{(R_k, w_k)\}$ | scale configuration (2.4) |
| $\ell$ | embedding dimension, $\ell = \sum_k |\mathbb{G}_{R_k}^+| = O(MN)$ |
| $\lambda(\nu)$ | distortion slope (2.11); $\rho_-$ its affine certificate |
| *Classification (Sections 3–5)* | |
| $k, Y$ | number of classes; class label in $[k]$ |
| $R$ | embedding radius $\sup_A \|\Phi(A)\|$ |
| $\mu_c, \hat{\mu}_c$ | population / empirical class-$c$ mean |
| $\Sigma_c$ | class-$c$ covariance of $\Phi$ |
| $\Delta, \hat{\Delta}$ | population / empirical class-mean separation (3.1) |
| $D_c$ | within-class radius $\sup_{A:Y=c} \|\Phi(A) - \mu_c\|$ |
| $\delta_*$ | bottleneck separation of class supports |
| $m, m_c, m_{\min}$ | sample size; per-class count; smallest per-class count |
| $\rho, \delta$ | SVM margin parameter; confidence level $1 - \delta$ |
| $\mathcal{R}(h), \widehat{\mathcal{R}}_\rho(h)$ | generalization risk; empirical $\rho$-margin loss |
| *Selection and certificate (Sections 4, 5)* | |
| $\mathcal{F}, f$ | descriptor pool; a descriptor |
| $\eta = \Delta/\sqrt{\ell}$ | isotropic selection surrogate |
| $\rho_{\mathrm{Mah}}$ | Mahalanobis margin (4.1) (Ledoit–Wolf shrunk) |
| $\alpha, r_m$ | certificate level; class-mean concentration radius |

# B   Auxiliary results used in the proofs

This appendix collects the classical concentration inequalities invoked in Section 3 and Section 5, together with the volumetric estimate underpinning the lower bound of Theorem 3.2. All results are stated in the forms used in the body and are attributed to their standard references; we reproduce the statements for self-containedness.

**Lemma B.1** (Pinelis's Hilbert-space Hoeffding (Pinelis, 1994, Thm. 3.4)). *Let $X_1, \ldots, X_m$ be i.i.d. random vectors in a separable Hilbert space $\mathcal{H}$ with $\mathbb{E}[X_i] = 0$ and $\|X_i\| \le B$ almost surely. Then for every $t > 0$,*

$$\mathbb{P}\big(\big\|m^{-1}\sum_{i=1}^m X_i\big\| > t\big) \ \le \ 2\exp\left(-\frac{mt^2}{2B^2}\right).$$

**Lemma B.2** (Hellinger distance between uniform distributions on translated $\ell^2$-balls, after (Tsybakov, 2009, Ch. 2.4)). *Let $B_r := \{x \in \mathbb{R}^\ell : \|x\| \le r\}$ and set $P_\pm := \mathrm{Unif}(B(\pm\mu, r))$. With the Hellinger convention $H^2(P, Q) := \frac{1}{2}\int(\sqrt{p} - \sqrt{q})^2\,dx$ (so $H^2 \in [0,1]$ and $\mathrm{TV} \le \sqrt{2H^2}$, matching (Tsybakov, 2009, Ch. 2.4)), if $\|\mu\| \le r$,*

$$H^2(P_+, P_-) \ = \ 1 - \frac{\mathrm{vol}(B(\mu, r) \cap B(-\mu, r))}{\mathrm{vol}(B_r)} \ \le \ c_\ell\,\frac{\|\mu\|}{r},$$

*where the constant $c_\ell$ admits the explicit bound*

$$c_\ell \ = \ \frac{2\,V_{\ell-1}(1)}{V_\ell(1)} \ = \ \frac{2}{\sqrt{\pi}}\frac{\Gamma(\ell/2 + 1)}{\Gamma((\ell+1)/2)} \ \le \ \sqrt{\frac{2(\ell+1)}{\pi}}, \tag{B.1}$$

where $V_d(1) = \pi^{d/2}/\Gamma(d/2+1)$ is the volume of the $d$-dimensional unit ball. By Stirling, $c_\ell = \Theta(\sqrt{\ell})$ as $\ell \to \infty$.

*Proof of the bound on $c_\ell$.* Slicing $B_r$ perpendicular to $\mu$ at signed distance $u$ from the origin gives an $(\ell-1)$-ball of radius $\sqrt{r^2-u^2}$. Taking $\mu = (\|\mu\|, 0, \dots, 0)$ and integrating over the slice parameter, the missing volume is $\mathrm{vol}(B_r) - \mathrm{vol}(B(\mu, r) \cap B(-\mu, r)) = 2\int_0^{\|\mu\|} V_{\ell-1}(\sqrt{r^2-u^2})\,du$, and bounding $V_{\ell-1}(\sqrt{r^2-u^2}) \leq V_{\ell-1}(r)$ for $u \in [0, \|\mu\|]$ yields $H^2 \leq 2V_{\ell-1}(r)\|\mu\|/V_\ell(r) = (2V_{\ell-1}(1)/V_\ell(1))\,\|\mu\|/r$. The Gamma-ratio identity follows from $V_\ell(1) = \pi^{\ell/2}/\Gamma(\ell/2+1)$, and the $\sqrt{2(\ell+1)/\pi}$ upper bound from $\Gamma(x+1/2)/\Gamma(x) \leq \sqrt{x}$ at $x = (\ell+1)/2$. $\qquad\square$

**Lemma B.3** (Multivariate Berry–Esseen (Bentkus, 2003, Thm. 11)). *Let $X_1, \dots, X_m$ be i.i.d. mean-zero random vectors in $\mathbb{R}^\ell$ with covariance $\Sigma$ and finite third moment $\beta_3 := \mathbb{E}\|X_1\|^3 < \infty$. Write $S_m := m^{-1/2}\sum_{i=1}^m X_i$ and let $G \sim \mathcal{N}(0, \Sigma)$. Then for every convex set $A \subseteq \mathbb{R}^\ell$,*

$$\big|\mathbb{P}(S_m \in A) - \mathbb{P}(G \in A)\big| \leq \frac{C\,\ell^{1/4}\,\beta_3}{\|\Sigma\|_{\mathrm{op}}^{3/2}\,\sqrt{m}},$$

*with $C > 0$ a universal constant.*

**Lemma B.4** (Matrix Bernstein (Tropp, 2015, Thm. 6.1)). *Let $X_1, \dots, X_m$ be i.i.d. self-adjoint random matrices in $\mathbb{R}^{\ell\times\ell}$ with $\mathbb{E}[X_i] = 0$ and $\|X_i\|_{\mathrm{op}} \leq R$ a.s., and covariance parameter $\sigma^2 := \|\mathbb{E}[X_i^2]\|_{\mathrm{op}}$. Then for every $t > 0$,*

$$\mathbb{P}\Big(\big\|m^{-1}\sum_i X_i\big\|_{\mathrm{op}} > t\Big) \leq 2\ell\exp\Big(-\frac{m\,t^2/2}{\sigma^2 + Rt/3}\Big).$$

*Applied to $X_i := \Psi_i\Psi_i^\top - \Sigma$ with $\|\Psi_i\| \leq R$, this yields $\|\hat{\Sigma}_m - \Sigma\|_{\mathrm{op}} = O(R^2\sqrt{\log \ell/m})$ with high probability.*

**Lemma B.5** (Gaussian plug-in concentration radius). *Let $\Psi_i := \Phi(A_i)$ with $\|\Psi_i\| \leq R$, and assume the class-conditional third moment $\beta_3 := \mathbb{E}\|\Phi(A) - \mu_c\|^3 < \infty$ for every class $c \in [k]$. Define the Gaussian plug-in radius*

$$\tilde{r}_m := \max_c \sqrt{\frac{\|\hat{\Sigma}_c\|_{\mathrm{op}}}{m_c}\chi^2_{\ell,\,\alpha/k}},$$

*where $\chi^2_{\ell,\,\alpha/k}$ is the $1 - \alpha/k$ quantile of the chi-squared distribution with $\ell$ degrees of freedom. Then*

$$\mathbb{P}\Big(\max_c \|\hat{\mu}_c - \mu_c\| \leq \tilde{r}_m\Big) \geq 1 - \alpha - O\Big(\frac{\ell^{1/4}}{\sqrt{m}}\Big) - O\Big(R^{1/2}\|\Sigma_c\|_{\mathrm{op}}^{1/4}\big(\tfrac{\log \ell}{m}\big)^{1/4}\sqrt{\tfrac{\ell}{m}}\Big),$$

*both error terms vanishing once $m_c \geq m^\dagger = O(\sqrt{\ell})$ for every $c$.*

*Proof.* For $X \sim \mathcal{N}(0, \Sigma_c)$ in $\mathbb{R}^\ell$, the squared norm $\|X\|^2 = \sum_i \lambda_i Z_i^2$ is a weighted sum of independent $\chi_1^2$ variables with weights $\lambda_i$ equal to the eigenvalues of $\Sigma_c$; bounding above by $\|\Sigma_c\|_{\mathrm{op}}\cdot\chi_\ell^2$ gives, for the Gaussian approximation $\mathcal{N}(0, \Sigma_c/m_c)$, a concentration radius controlling $\|\hat{\mu}_c - \mu_c\|$ with probability $\geq 1 - \alpha/k$:

$$\tilde{r}_m^{(c)} := \sqrt{\|\Sigma_c\|_{\mathrm{op}}\cdot\chi^2_{\ell,\,\alpha/k}/m_c}.$$

This bound is conservative when $\Sigma_c$ is low-rank, with conservatism governed by $\mathrm{tr}(\Sigma_c)/(\ell\,\|\Sigma_c\|_{\mathrm{op}})$. By the multivariate Berry–Esseen theorem (Lemma B.3) applied to the convex set $A = B(0, \tilde{r}_m^{(c)}\sqrt{m_c})$, the true distribution of $\sqrt{m_c}(\hat{\mu}_c - \mu_c)$ deviates from $\mathcal{N}(0, \Sigma_c)$ in total variation by at most $C\ell^{1/4}\beta_3/(\|\Sigma_c\|_{\mathrm{op}}^{3/2}\sqrt{m_c})$, which is the first error term. Replacing $\Sigma_c$ by the sample covariance $\hat{\Sigma}_c$ introduces the additional operator-norm error $\|\hat{\Sigma}_c - \Sigma_c\|_{\mathrm{op}} = O(R\sqrt{\|\Sigma_c\|_{\mathrm{op}}\log(\ell)/m_c})$ supplied by the matrix Bernstein inequality (Lemma B.4), which propagates into the radius via $\big|\|\hat{\Sigma}_c\|_{\mathrm{op}}^{1/2} - \|\Sigma_c\|_{\mathrm{op}}^{1/2}\big| \leq \|\hat{\Sigma}_c - \Sigma_c\|_{\mathrm{op}}^{1/2}$ as the second error term, of order $O\big(R^{1/2}\|\Sigma_c\|_{\mathrm{op}}^{1/4}(\log(\ell)/m_c)^{1/4}\sqrt{\chi^2_{\ell,\alpha/k}/m_c}\big)$. Taking the worst-case class and applying a Bonferroni correction over the $k$ classes gives the stated radius $\tilde{r}_m$. $\qquad\square$

**Remark B.1** (Three radii: regime split)**.** *The three radii of Theorem 5.1 fit into a clean spectrum. Pinelis (i) is dimension-free ($r_m^{\mathrm{Pin}} \propto R\sqrt{L/m}$ with $L = \log(2k/\alpha)$) but $L^2$-envelope-dominated. Gauss (ii) replaces $R^2$ by $\|\Sigma_c\|_{\mathrm{op}}$ but introduces a $\sqrt{\chi^2_{\ell,\alpha/k}}$ dimension penalty. Pinelis–Bernstein (iii) keeps the dimension-free $\sqrt{L}$ Bonferroni cost of (i) and the $\sqrt{\|\Sigma_c\|_{\mathrm{op}}}$ refinement of (ii) simultaneously; for embeddings with stable rank $\mathrm{tr}(\Sigma_c)/\|\Sigma_c\|_{\mathrm{op}} = O(1)$ (empirically $\leq 1.17$ on our benchmarks), it dominates the other two and is the only form that fires the certificate at our sample sizes (Table 6).*

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
