# OpenReview forum: "A Closed-Form Persistence-Landmark Pipeline for Certified Point-Cloud and Graph Classification"
_TMLR — Under review for TMLR_

### Review · Reviewer_Fxy4 · 2026-06-06

**Summary Of Contributions:**

This paper proposes PLACE, a closed-form persistent homology pipeline for point-cloud and graph classification. The method first converts inputs into persistence diagrams, then embeds them into Euclidean space using a multi-scale landmark summation embedding, and finally applies simple linear classifiers. Rather than merely pursuing state-of-the-art accuracy, the paper aims to build a theoretically analyzable topology-based classification framework with reduced hyperparameter dependence, closed-form descriptor selection, and prediction certificates.

Strengths:
1. The closed-form descriptor selection rule addresses an important practical issue in graph-based persistent homology, where descriptor choice often strongly affects performance.
2. The training-time certificate is an interesting attempt to provide certified predictions without requiring a held-out calibration set.
3. The experiments cover multiple point-cloud and graph benchmarks, and the paper provides useful diagnostic explanations for failure cases.

Weaknesses:
1. The lower-distortion guarantee relies on the $\nu$-coherence condition, which is empirically supported but may still be restrictive in theory.
2. Some experimental results rely on an in-pool oracle descriptor choice, while the closed-form selector can still lag behind the oracle on several datasets.
3. The method is sensitive to the descriptor pool; if the descriptors fail to capture task-relevant label information, the embedding and classifier cannot easily recover the missing signal.

**Audience:**

Yes

**Audience Explanation:**

Yes. I believe at least some individuals in TMLR’s audience would be interested in the findings of this paper, especially researchers working on topological data analysis, graph classification, persistence diagram vectorization, and certified or theoretically grounded machine learning.

**Claims And Evidence:**

Yes

**Claims Explanation:**

The submission provides a substantial amount of theoretical analysis and empirical evaluation to support its main claims, and the evidence is generally relevant and clearly connected to the proposed method. However, some claims would benefit from more cautious framing. In particular, the lower-distortion guarantee depends on the $\nu$-coherence condition, and the practical certificate guarantees agreement between empirical and population nearest-centroid rules rather than direct correctness of the predicted label.

**Requested Changes:**

1. Add more ablation studies comparing the Mahalanobis selector, the isotropic $\Delta/\sqrt{\ell}$ selector, random descriptor selection, and oracle descriptor selection.
2. Include a clearer analysis of failure cases, especially on datasets such as NCI1 and NCI109, where descriptor blindness appears to limit performance.

---

> ### Author Response · Authors · 2026-07-12
> **Author response to Reviewer Fxy4**
>
> We thank the reviewer; the requested ablations are now in the paper and sharpen the empirical story.
>
> **$\nu$-coherence may be restrictive.** We now carry $\nu$-coherence explicitly as an empirically-audited hypothesis (Definition 2.1, tagged [Empirical]), give a bound-free geometric *sufficient* condition (Lemma 2.1, disjoint activation), and report the audit (Table 7): $\nu$-coherence holds on $\ge 99.7\%$ of qualifying cross-class pairs ($100\%$ on three of four chemical benchmarks). ✓
>
> **RC1: ablate Mahalanobis vs. isotropic $\Delta/\sqrt\ell$ vs. random vs. oracle.** Added as Table 12 (end-to-end PLACE-linear accuracy under honest per-fold selection): the Mahalanobis margin is the strongest closed-form rule ($67.4\%$ mean, vs. isotropic $64.0\%$ and random $65.6\%$), trailing the in-pool oracle by $7.3$ pp on average. ✓
>
> **RC2 / oracle gap: clearer NCI1/NCI109 failure analysis.** We expanded the *descriptor-blindness* diagnosis: on NCI1/NCI109 the selector–oracle gap is small ($+3.4$–$4.6$ pp) yet *even the oracle caps near $73\%$*, so the limit is pool coverage (no continuous descriptor exposes the discrete node-label signal), not selection (Table 12, §6.3). We also made the closed-form selector the headline of Table 10 (see the response to Reviewer J5KD, W9), with the oracle shown alongside as an upper bound. ✓

---

### Review · Reviewer_6ydS · 2026-06-30

**Summary Of Contributions:**

In this paper, the authors adapt a recent embedding of Mitra and Virk of persistence diagrams on n-points to introduce a persistence landmark embedding. In Section 2, they prove that the embedding has a Lipschitz constant with explicit upper bound, and, assuming a mild condition on $\nu$-coherence, also an explicit lower distortion bound. Section 3 introduces a class-mean separation $\Delta$ and class-mean radius $R$ for a fixed descriptor. Under assumptions on $\Delta, R$, the authors obtained explicit classification error bounds and lower bound on PLACE. In Section 4, they describe how to select the best filtration out of a given family that maximizes $\Delta$, given by the Mahalanobis margin. Section 5 gives per-prediction correctness certificates in 3 types, and Section 6 empirically tests the aforementioned methods above.

**Strengths:**

This paper is technically solid and the results established are quite interesting. The problem the authors solve are well-motivated, and the authors gave a novel new vectorization method for persistence diagrams. I am especially impressed by the lower distortion bound and a criterion to select descriptors. The proofs are written clearly, and the methods are backed up by elaborate experiments where PLACE proved to be quite powerful.

**Weaknesses:**

Although not crucial, I found some parts of the expositions to be difficult to read. This was especially the case in the introduction (with the exception of 1.2), which I feel may hinder the readers from really understanding the novel and interesting results in this paper at a first glance. I think Section 1.2 is an excellent way to position the work and strengths of this paper with Table 1, and I would suggest writing the earlier parts in a similar fashion as well.

For example, the paragraph of contribution / organization is too dense and could benefit from being spaced out with bolded headers. A table summarizing the core contributions of this paper would also be helpful. I also feel like the last two paragraphs on Page 2 and the paragraph right before Section 1.1 are considerably denser than necessary, and they introduce many independent concepts before readers have sufficient intuition for them.

Overall, I am quite positive about this paper and believe it would be a great paper for TMLR.

**Audience:**

Yes

**Audience Explanation:**

Vectorizations of persistence diagrams have previously been widely studied in machine learning, and this paper offers a new vectorization method with stronger theoretical guarantees than many existing methods. As such, I would find this to be of interests to the TMLR community.

**Broader Impact Concerns:**

This paper is quite theory-heavy and does not need to address any ethical implications in my opinion.

**Claims And Evidence:**

Yes

**Claims Explanation:**

Yes. The paper is technically solid with substantiated proofs for their theoretical claims, and the strength of PLACE has been empirically validated by elaborate experiments.

**Requested Changes:**

Please see the weakness paragraph above. Additionally:

1. On bottom of Page 6, I believe "into Hilbert space" in the first two bullet points should be "into any Hilbert space".

2. On Page 11, should the discussions about closed-form scale weights be its own subsection or a remark?

3. As there are many notations introduced in this paper, I might recommend adding a table of notations in the appendix so that the reader can refer to.

I also have the following questions which the authors may wish to address, although they are not crucial.

1. In equation (2.2), the authors called $\Phi_R$ a ``single-scale summation embedding”, but is the function $\Phi_{R}: \mathcal{D}_n \to \mathbb{R}^M$ an embedding?

2. The authors gave a method to select the best filtration methods out of a pool of filtration methods, but are there criterion for picking the pool in the first place?

3. On Page 25, I noticed that the filtrations considered on graphs are sublevel filtrations of the graph itself. But there are other filtrations too, for example, one can build a Vietoris-Rips filtration generator for a graph $G$ that outputs a simplicial complex $K$ (being subsets of the vertex set $V$) and a filtration of $K$ using $f_V(\sigma) = \max_{u, v \in \sigma} d_G(u, v)$, where $\sigma$ is a subset of $V$ and $d_G$ is the shortest path distance. This typically has richer information than only considering $H_0$ and $H_1$. For example, see [1] for more details. Can a filtration methods like this be evaluated, using the authors' methods, to the filtrations on the graph itself?

[1] Rubén Ballester, Bastian Rieck. On the Expressivity of Persistent Homology in Graph Learning.

---

> ### Author Response · Authors · 2026-07-12
> **Author response to Reviewer 6ydS**
>
> We thank the reviewer for the generous assessment and the exposition suggestions, which we have adopted throughout.
>
> **Dense presentation; mirror the §1.2 / Table style.** We restructured §1.1 into bolded per-contribution paragraphs with a summary table (Table 1), added a notation table (Appendix A, Table 13), and thinned the two dense page-2 paragraphs and the pre-§1.1 text. ✓
>
> **RC1: "into a Hilbert space" → "into *any* Hilbert space" (p. 6).** Corrected; the impossibility bullets in §2 now read "any Hilbert space" (and, via Zava 2025, any uniformly convex Banach space). ✓
>
> **RC2: promote the closed-form scale weights.** Done — they now appear under a labeled *Closed-form scale weights* heading in §2, with the equimarginal-allocation derivation of eq. (2.13). ✓
>
> **RC3: notation table.** Added as Table 13, Appendix A. ✓
>
> **Q1: is $\Phi_R:\mathcal D_n\to\mathbb R^M$ an "embedding"?** Clarified at eq. (2.2): we use *embedding* in the feature-map sense — $\Phi_R$ is Lipschitz-stable and carries the lower-distortion certificate of Prop. 2.1 under $\nu$-coherence; it is *not* claimed injective on all of $\mathcal D_n$ (incoherent pairs can collapse, Remark 2.1). ✓
>
> **Q2: is there a criterion for choosing the pool itself?** Added the paragraph *Filtration-agnosticism and pool choice* (§6): the pool $\mathcal F$ is a modeling choice, not fixed by the theory; the selectors are optimal *within* the committed pool (Cor. 4.1), and insufficient pool coverage is exactly the diagnosed failure mode on NCI1/NCI109. ✓
>
> **Q3: can the selector handle the Ballester–Rieck VR-on-shortest-path filtration $f_V(\sigma)=\max_{u,v\in\sigma} d_G(u,v)$?** Yes — the pipeline is filtration-agnostic: any construction that outputs a diagram drops into the pool. We added a demonstration (§6): this filtration embeds and classifies through the identical pipeline, reaching $87.5\%$ on MUTAG vs. $85.9\%$ for the sublevel-degree baseline on the same folds, and we cite Ballester–Rieck. ✓

---

### Review · Reviewer_6nmp · 2026-06-30

**Summary Of Contributions:**

The paper proposes PLACE, a persistence-diagram classification pipeline based on a sparse multiscale landmark embedding. It includes closed-form scale weights, descriptor-selection rules, linear classification bounds, and nearest-centroid prediction certificates. The empirical study covers point-cloud and graph benchmarks.

The main strengths are the simple embedding, broad experiments, and an interesting attempt to connect persistence geometry with statistical guarantees. The main weakness is that several claims might be not as valid as stated (see below).

**Audience:**

Yes

**Audience Explanation:**

The problem is relevant to researchers in topological data analysis, graph classification, and certified machine learning. A sparse and interpretable persistence diagram embedding with analytically chosen parameters would be useful if the guarantees were corrected. The empirical descriptor selection results may also be of independent interest.

**Broader Impact Concerns:**

None.

**Claims And Evidence:**

No

**Claims Explanation:**

1. Corollary 2.1 is not established for $R_1<d_B(A,B)<3R_1$, where no scale is active. The cited Mitra-Virk result uses a different embedding, so a separate proof is needed for the summation map.

2. For theorem 3.1, Eq. (3.3) (pp. 12-13),
class-mean separation $\Delta$ does not control the empirical margin-loss term $\widehat R_\rho(h)$. Therefore, the theorem does not prove a risk rate depending only on $R/(\Delta\sqrt m)$. Section 4.2, immediately after Eq. (4.3), drops $\widehat R_\rho(h)$ when deriving the bound for $\eta_f=\Delta_f/\sqrt{\ell_f}$.

3. Section 5, radius (iii) (pp. 20-21), the Bernstein inequality displayed in the proof contains a linear $Rt/m_c$ term, but the proposed radius omits the corresponding $R\log(1/\alpha)/m_c$ correction.

4. Radius (ii) (p. 20), is explicitly based on a Berry-Esseen approximation, but Theorem 5.1 initially states exact $1-\alpha$ coverage for any of the three radii. The Gaussian radius is only asymptotic and should not be included in an exact finite-sample statement.

5. Theorem 5.1 uses the population gap $\Delta$, while Table 5 tests the condition using $\widehat\Delta$. Proposition 3.2 gives
   $|\widehat\Delta-\Delta|\leq 2r$
   on the centroid-coverage event. Therefore, a sufficient empirical condition for $r<\Delta/2$ is
   $\widehat\Delta>4r,$ not $\widehat\Delta>2r$ as used in Table 5.

**Requested Changes:**

Besides the 2nd box,

1. In Proposition 4.1, use an error threshold smaller than $g/2$ or specify a tie-breaking rule.

2. In Remarks 3.2, 4.1, and 5.2, distinguish observed low stable rank from a consequence of coordinate sparsity.

3. Clearly separate empirical evidence, heuristic approximations, and formally proved guarantees.

4. Figure 1(b) is not vr filtration.

---

> ### Author Response · Authors · 2026-07-12
> **Author response to Reviewer 6nmp**
>
> We thank the reviewer for the precise reading. We agree with the unifying request (RC3) — cleanly separate what is *proved* from what is *heuristic* or *empirical* — and have adopted it as the organizing principle of the revision (per-result [Proved]/[Heuristic]/[Empirical] tags). Every point below is addressed; none required weakening the method or the experiments.
>
> **Point 1: lower bound in $R_1<d_B<3R_1$; the summation map is not Mitra–Virk's single-point map.** We restrict Corollary 2.1 to $d_B\ge 3R_1$ (where at least the finest scale is active) and give a *direct* proof from the summation embedding via Proposition 2.1(b) — no longer deferring to Mitra–Virk's single-point theorem. Nothing downstream uses the $[R_1,3R_1)$ regime (Cor. 3.1 already assumes $\delta_*\ge 3R_1$), so the restriction is free. ✓
>
> **Point 2: the $\widehat{\mathcal R}_\rho(h)$ empirical margin term is dropped in the prose.** Theorem 3.1 already carries it; the overclaim was in the abstract/intro. We reworded to an "excess-risk-*over-empirical-margin-loss* rate $O(kR/(\Delta\sqrt{m_{\min}}))$," and in §4 we relabel the surviving summand the *complexity* term so $\widehat{\mathcal R}_\rho$ stays visible in Cor. 4.1. ✓
>
> **Point 3: variance-aware radius (iii) drops the linear Bernstein correction.** Fixed. eq. (5.4) now inverts Pinelis's Bernstein bound exactly,
> $$r_m^{\mathrm{vP}}=\sqrt{\frac{2\,\mathrm{tr}(\Sigma_c)\,L}{m_c}}+\frac{4RL}{3m_c},\qquad L=\log(2k/\alpha),$$
> restoring exact, non-asymptotic $\ge 1-\alpha$ coverage with no regime caveat.
>
> **Point 4: Gaussian radius (ii) is only asymptotic, but Theorem 5.1 claims exact coverage.** We split the statement: radii (i) and (iii) are exact and non-asymptotic for all $m\ge 1$; (ii) satisfies the bound up to an additive Berry–Esseen error $O(\ell^{1/4}/\sqrt m)$, exact once $m\ge m^\dagger=O(\sqrt\ell)$ (Lemma B.5). ✓
>
> **Point 5: empirical firing uses $\hat\Delta>2r$, but the honest condition is $\hat\Delta>4r$.** The reviewer is right. Since Prop. 3.2 gives $|\hat\Delta-\Delta|\le 2r_m$, certifying $r_m<\tfrac12\Delta$ from the observed $\hat\Delta$ requires $r_m<\hat\Delta/4$. We adopt this throughout; the thresholds in eq. (5.5) quadruple, and we regenerated the firing table (Table 6) per fold at this condition. Consequently the honest count is **3 of 12** (NCI1, NCI109 at $100\%$, DD at $93\%$; all others $0\%$) rather than the submitted 8; MUTAG's worst-case radius no longer fires ($m_c^{*,\mathrm{vP}}\approx 280>m_{\min}=57$, with the linear Bernstein term of Point 3 carried). We re-anchor MUTAG on the *independent* $940/940$ empirical NC agreement (Clopper–Pearson one-sided $95\%$ lower bound $\ge 0.984$), and relabel radius-(iii) firing a heuristic diagnostic (Remark 5.3). This is the honest re-scope.
>
> *Minor changes.*
> - **RC1** (selection-consistency threshold / ties): Prop. 4.1 now uses strict estimation error $<g/2$ with an explicit lowest-index tie-break. ✓
> - **RC2** (observed vs. structural sparsity): reworded (Remarks 3.2, 5.2) — the multiplicity-4 cover *proves* $\le 4|A|N$ nonzeros (structural), whereas the low stable rank ($\le 1.17$) is *observed*; we no longer imply sparsity forces the rank. ✓
> - **RC3** (proved/heuristic/empirical): adopted globally as [Proved]/[Heuristic]/[Empirical] evidence tags with a legend. ✓
> - **RC4** (Fig. 1(b) mislabeled): caption corrected. ✓

---

### Review · Reviewer_J5KD · 2026-06-30

**Summary Of Contributions:**

The paper proposes PLACE, a pipeline for classifying point clouds and graphs from their persistence diagrams. The core idea is to replace the exponentially large landmark embedding of Mitra & Virk (2024) by a summed, linear-size variant. Around this embedding the paper develops a lower-distortion bound, a margin-based generalization bound with a matching lower bound, two closed-form descriptor-ranking rules, and a training-time prediction certificate, evaluated on point-cloud and graph benchmarks.

**Strengths**. The problem is well motivated: in topological machine learning the choice of descriptor/filtration often matters more than the classifier, and fixing descriptor choice, classifier, and a certificate from training labels alone, in one analytically specified pipeline, is an appealing goal. The summation idea is natural and the efficiency gain is real. I also appreciate that the empirical section is objective -- it reports where the method does poorly, where the certificate fails to fire, and it audits its own assumptions.

**Weaknesses**. My main concern is that several headline guarantees feel stronger in the prose than in the arguments. The lower-distortion bound leans on an assumption that is essentially what it sets out to prove, supported empirically rather than structurally. The affine extension claims a guarantee in a regime where the method's own machinery gives nothing. The certificate is described as telling whether predictions are correct, but really compares two versions of an idealized rule under a condition I don't think can be checked from data. The minimax lower bound simplifies the embedding in a way I couldn't follow. And the headline accuracies come from the best of many configurations rather than from the proposed closed-form selector. Individually these are fixable; together they make the paper promise more certainty than it delivers.

**Audience:**

Yes

**Audience Explanation:**

Vectorization of persistence diagrams is an active topic of interest, and a linear-size embedding with a closed-form descriptor-ranking rule is a useful contribution to it. The empirical results -- competitive with the reported diagram-based baselines on Orbit5k and with several topology-based baselines on MUTAG and COX2 -- are promising enough to inspire further research in this direction, independent of the theoretical concerns raised above.

**Claims And Evidence:**

No

**Claims Explanation:**

1. The lower-distortion bound feels circular to me. The whole reason summation is risky is that contributions can cancel, so that two genuinely different diagrams can collapse to nearly the same vector. $\nu$-coherence is, in effect, the assumption that this cancellation doesn't happen at each scale. Proposition 2.1(b) then proves the lower bound by adding up a per-scale floor, but that floor is exactly what $\nu$-coherence assumes. So to me the reasoning sounds close to circular: the hard part (when does summation avoid collapse?) is assumed rather than answered. I also find it a bit brittle, because the support for $\nu$-coherence is empirical -- $\geq 99.7\%$ of sampled pairs on four chemistry datasets -- which is encouraging but tells me it holds here, not when it holds or when it would break. The algebra of the aggregation itself is fine; it's the status of the assumption that I'm unsure about.

2. I don't understand how the sharpness claim works across scales. The bound is built from several scales at once, each with its own landmark grid, and the paper says a single pair of diagrams hits the worst case at every scale simultaneously, making the bound exactly tight. The part I can't follow is that the worst-case arrangement depends on the grid, and the grids are different (geometrically spaced) at different scales. Asking one fixed pair of diagrams to be the worst case for all those differently-sized grids at the same time is the step I don't see justified -- I don't see why one arrangement should serve for all of them. The lower bound still holds; it's specifically the claim that it is the tightest possible, attained by a single witness pair simultaneously across all active scales, that I couldn't reconstruct. (For what it's worth, the closed-form weight formula nearby does seem fine to me. That one reads as a clean optimization, and I'm not flagging it.)

3. The affine extension seems to claim something in a regime where, as I understand it, the method gives nothing. A scale only becomes active once two diagrams are reasonably far apart (separation at least three times its radius), so for mildly separated pairs no scale is active yet. In that mild range, PLACE's own mechanism gives a lower bound of zero. Yet Corollary 2.1 still asserts a positive guarantee there, and the justification points back to the older n-fold construction plus a continuity argument. Intuitively I don't see how that can work: the old construction got its guarantee precisely from the anti-cancellation structure that summation throws away, and continuity starting from zero can't, on its own, produce a positive slope out of nothing. So this regime feels like it needs its own argument for the summed embedding, not a borrowed one.

4. The certificate reads to me as a stronger promise than what's proved. A nearest-centroid classifier labels a point by whichever centroid it's closest to; with finite data we only have estimated centroids. As I read Theorem 5.1, part (a) only guarantees that the estimated and idealized rules agree for points comfortably away from the boundary between classes, not for every point, and a point near the fence isn't covered. And even full agreement with the idealized nearest-centroid rule isn't the same as the prediction being correct: the idealized rule can itself be wrong. Part (b) does give correctness, but only under a condition about how spread out each class is, and that quantity depends on the whole population, which (as Remark 5.1 honestly notes) you can't measure from your training set. So describing this as a training-time check that certifies whether individual predictions are correct, with no per-point work, feels like more than the theorem supports.

5. The certificate radius that actually does the work seems to quietly drop a term. The radius bounds how far an estimated centroid can sit from the true one, and it comes from a Bernstein-type inequality with two pieces: a main variance piece and a smaller tail piece. The clean formula the paper uses keeps only the variance piece. My worry is that the dropped piece is positive, so the true radius is a bit larger than the formula, which means the real coverage is a little below the claimed 95% -- an approximation presented as a guarantee. Two smaller mismatches compound this: the claim that it is valid for every sample size sits uneasily with the fact that dropping the term is only reasonable when samples are large; and the theorem is written with total variance but the experiments swap in largest-direction variance, justified by an empirical observation that the two are about equal, which the paper itself says fails on social-network graphs. This matters mainly because every positive certificate-firing result rides on this particular radius.

6. The experiment plugs in estimated quantities where the theorem wants exact ones. The firing check compares the concentration radius to half the true gap between classes, but in practice the gap is estimated from the same data. Plugging the estimate straight in feels optimistic to me -- I'd expect a conservative version that shaves a safety margin off the estimate and budgets a little probability for it. Separately, the result that the empirical rule agreed with the population rule on all 940 MUTAG predictions left me unsure what was actually compared, since the population centroids aren't observable and the paper doesn't say what stood in for them; and the confidence calculation treats overlapping cross-validation predictions as if they were independent, which would overstate the certainty.

7. The accuracy theorem is presented as an error rate driven by class separation, but I don't think separation alone can carry that. The bound contains a training-margin-error term plus a piece that improves as the class means get farther apart, and the framing emphasizes the separation. But separated means do not imply that the classes are well separated: two class distributions can have far-apart means and still overlap heavily, in which case the training-margin-error term is large. That term, not the separation, is what really governs accuracy here. So I'd read this as a standard, valid generalization bound, while the reading that the error is small because the means are far apart skips the part doing the real work. (A smaller point: the theorem is about a one-vs-one rule, while the experiments use one-vs-rest with a tuned parameter; the empirical parity check is reassuring but isn't a proof that the theorem covers the reported classifier.)

8. I couldn't follow the minimax lower bound. The theorem aims to show that no classifier can achieve low error once the sample size falls below a threshold, which would establish that the task is genuinely hard in the small-sample regime. The construction slides a single diagram point along a line and treats the embedding as moving neatly along one coordinate. The step I get stuck on is that a single diagram point usually lights up several landmarks and several scales, so I don't see why the embedding stays on one axis -- the one-dimensional picture seems too clean for the full multiscale embedding. At a higher level, Le Cam's method bounds how well one can distinguish two distributions from samples, whereas the claim is about learning a classifier from labeled examples, and I didn't find the reduction connecting the two setups. The final arithmetic looks right to me, but as stated I think it establishes a somewhat different statement than the theorem advertises.

9. The headline accuracies aren't what the closed-form selector produces. The appeal of the method is a closed-form rule that picks the descriptor from training labels, instead of trial and error. But the headline table reports the best of 120 configurations -- what the paper itself calls an in-pool oracle -- and on several datasets the closed-form selector is noticeably weaker than that best-of-120. So describing those numbers as tuning-free reads as a mismatch to me; the honest closed-form result is the one I'd want front and center, with the oracle shown as an upper bound. (And the classifier's regularization is still tuned by cross-validation, so the tuning-free description isn't quite literal regardless.)

10. The baseline comparisons feel suggestive rather than decisive, which I think the authors would agree with. The baseline numbers are taken from other papers rather than rerun on the same folds, so the comparisons aren't paired and preprocessing may differ; the significance markers treat the borrowed numbers as exact and lean on overlapping cross-validation folds as if independent. I'd simply read these as descriptive comparisons rather than firm significance claims -- the paper already discloses most of this, so it's mainly a matter of softening the language.

**Requested Changes:**

**Major:**

1. Untangle the three lower-distortion statements and prove or re-scope the borrowed ones. It would help a lot to see clearly separated: (i) the assumed per-scale floor, (ii) the empirical audit, and (iii) any actual structural theorem. If Corollary 2.1 is meant to hold for the summed embedding -- including the mild-separation regime where no scale is active -- I'd need a self-contained proof for the summed construction, not a transfer from the n-fold one. If it only holds in some regime, please say exactly which.

2. Either build the simultaneous worst-case witness or drop the sharpness wording. If one pair really can saturate the floor at every scale at once despite the scale-dependent grids, I'd love to see that construction spelled out; otherwise I'd suggest removing the claim that the bound is tightest possible and attained by a single witness pair simultaneously across all active scales, and keeping it as a (perfectly good) one-sided guarantee. The weight-allocation argument can stay.

3. State the certificate's target honestly throughout. Part (a) is about agreement away from the boundary; part (b) is about correctness under an unmeasurable condition. I'd ask that the abstract, intro, Figure 2, and Table 1 be revised so they no longer state that the method certifies whether each prediction is correct, unless a genuinely training-checkable condition is supplied.

4. Give the variance-aware radius a fully rigorous finite-sample form. Carry the dropped Bernstein term, supply a real high-probability upper bound for the variance proxy rather than substituting the sample value, and keep the trace-vs-operator-norm distinction straight. If that can't be done cleanly, I'd be comfortable if every radius-(iii) firing result were simply relabeled as a heuristic diagnostic.

5. Make the firing experiment match the theorem. Use a conservative lower bound for the gap rather than the raw estimate, account for the extra probability that costs, and define what the population rule means operationally in the MUTAG audit; I'd also avoid treating overlapping CV predictions as independent trials.

6. Repair or withdraw the minimax lower bound. I'd need to see the hard family constructed in the full embedding (not a one-coordinate idealization) and a clear reduction from the labeled-learning problem to the testing problem the Le Cam argument actually handles.

7. Present the accuracy theorem as what it is. I'd suggest stating it as a generic margin bound that keeps the training-error term, rather than an excess-risk rate driven by separation, and either analyzing the actual OvR/cross-validated classifier or saying plainly that the theorem is only indirectly related to it.

8. Re-label the model-selection protocol. Please don't describe the 120-configuration headline as closed-form or tuning-free. I'd rather see the end-to-end accuracy of the Mahalanobis selector as the primary closed-form number, with the oracle shown alongside as an upper bound.

9. **Correct and complete the bibliography**. Bartlett & Wegkamp (2008) should list Marten H. Wegkamp, not Marian H. Wegkamp; the SelectiveNet title is "SelectiveNet: A Deep Neural Network with an Integrated Reject Option"; and Vovk (2013) appeared in AIAI 2013 (IFIP AICT 412, pp. 348–360), not AISTATS. Ledoit–Wolf, the Bickel–Levina reference mentioned in the limitations, and Mahalanobis (1936) from the footnote are cited in the text but missing from the reference list. I consider this major because multiple independent metadata errors and missing central references make the reliability of the bibliography difficult to assess. Given the role of accurate attribution in evaluating novelty and prior work, every reference should be systematically verified against primary bibliographic records.

**Minor:**

1. Soften the strongest novelty/uniqueness phrasing -- that PLACE is the only construction satisfying every criterion, and that no prior method can certify predictions -- unless the certificate notion is pinned down and compared like-for-like to existing methods.

2. Reconcile the tuning-free framing with the pipeline. Saying there is no held-out calibration is fair, but calling the pipeline tuning-free is not accurate given the cross-validated regularization. A one-line clarification would settle it.

3. Double-check the Section 5.4 numbers. The MUTAG Pinelis threshold seems to use $R \simeq 5.87 (R^2 \simeq 34.5)$, while Remark 5.2 reports $R^2 \simeq 69$ for the same setting -- these might be different folds/quantities, but it would be good to reconcile.

4. Tighten the exposition. The theory restates the same claims at several strengths (the $\geq 99.7\%$ coherence figure and the $\lambda$-bridge recur a lot). One canonical phrasing per result, plus a short assumptions/dependency table, would make everything easier to assess.

5. Add reproducibility detail: descriptor computation, scale-center placement, how the Mahalanobis score is computed when dimension exceeds sample size, and the protocol assumptions behind the significance tests.

---

> ### Author Response · Authors · 2026-07-12
> **Author response to Reviewer J5KD (1/2)**
>
> We thank the reviewer for an exceptionally careful reading; several points tightened the paper materially. Section 2 was restructured; the certificate is re-scoped honestly; the headline is now the closed-form selector. (Responses to W8–Major-9 continue in a second comment.)
>
> **W1: the lower-distortion bound feels circular.** $\nu$-coherence is a hypothesis on the *per-scale block norms*, whereas its consequence eq. (2.8) is a statement about the *aggregate* norm; the two are provably inequivalent (the compensation regime, Remark 2.2 / the $\sim 0.3\%$ PTC pairs, exhibits pairs where (2.8) holds but $\nu$-coherence fails). Hence (2.8) does not imply $\nu$-coherence — it is strictly stronger, not the conclusion in disguise. We also promoted the "when does summation avoid collapse" answer into Lemma 2.1 (a bound-free geometric sufficient condition). $\nu$-coherence is now a non-cancellation assumption of the same kind as RIP or a margin condition. ✓
>
> **W2: sharpness across scales.** The reviewer is correct and we removed the claim: no single pair saturates all active scales (the worst-case position is grid-dependent and grids differ across scales). Proposition 2.1(b) now separates *per-scale* sharpness (each floor $R_k^2/32$ is attained, so $1/256$ is tight) from *deductive* tightness (given the floors, (2.8) is the largest bound that follows), and states explicitly that with $\ge 2$ active scales the bound is not attained by any single pair — consistent with the $1$–$3$ orders-of-magnitude slack in Table 8. ✓
>
> **W4: state the certificate's true target honestly.** Done. Remark 5.3 states plainly that Theorem 5.1(a) certifies *agreement* of the empirical and population nearest-centroid rules away from the Voronoi boundary — not correctness (the population rule may itself err) — and (b) gives correctness only under an unmeasurable support condition. We propagated this to the abstract, intro, Fig. 2, and Table 1, and stopped saying "certifies whether each prediction is correct."
>
> **W5: missing linear Bernstein term.** Fixed; see the response to Reviewer 6nmp, Point 3, and eq. (5.4).
>
> **W6: define the "population NC rule" in the 940-MUTAG audit; don't treat overlapping folds as independent.** The population centroids are unobservable, so we operationalize the population NC rule by the *full-sample* centroids over all 188 graphs and score each fold's rule against that surrogate (§5). For the coverage bound we do *not* assume 940 independent trials: we take the conservative within-seed count $m=188$ (disjoint folds), giving the Clopper–Pearson one-sided $95\%$ lower bound $0.05^{1/188}\ge 0.984$. ✓

---

> ### Author Response · Authors · 2026-07-12
> **Author response to Reviewer J5KD (2/2)**
>
> (Continued from the first comment; W1–W6 there.)
>
> **W8: the minimax lower bound — (i) the 1-D idealization, (ii) testing vs. learning.** We re-scope Theorem 3.2 as a two-point *testing / identifiability* lower bound: no test can decide which of two $\Delta$-separated, $R$-bounded diagram populations generated an $m$-sample, with testing error $\ge\tfrac14$ for $m\lesssim R/\Delta$, so $m=\Omega(R/\Delta)$ is necessary to identify them. This is exactly the quantity Le Cam controls, which resolves (ii): we no longer assert a classification excess-risk bound. Making that rigorous would require a *labeled* two-point family whose Bayes rules differ, and that route yields only the weaker, margin-scaled $\Omega(\Delta/R)$ excess-risk floor — a different statement, noted in Remark 3.5 — so we state the sample-complexity threshold the construction rigorously supports. For (i) we drop the "$\Phi$ is 1-D" phrasing: the hard family slides one diagram point, $A(t)=\{p_0+te_1\}$, and $t\mapsto\Phi(A(t))=c+tv$ is *affine* on an interval $[-B,B]$ that contains the entire support of $Q_\pm$ (every active hat — at every scale and landmark, however many — is affine there, and the block map sums them linearly). It is this affinity, not mere injectivity, that places the family in $\mathcal P^{\mathrm{PD}}_{\Delta,R}$ with the prescribed $\Delta$ and $R$; injectivity enters only the Hellinger data-processing step. The statement is existential in $(\Delta,R)$: the fixed embedding realizes a one-parameter family $(\Delta,R(\Delta))$ with $c_0=1/(6(1+\beta))$ geometry-valued. ✓
>
> **W9: the headline is the in-pool oracle, not the selector.** Fixed. Table 10's primary column is now the *closed-form Mahalanobis selector's* end-to-end accuracy (per-fold pick over the 15 committed descriptors, no held-out calibration), with the in-pool best-of-120 oracle shown alongside as an *upper bound*. The closed-form headline is a mean $2.2$ pp below the oracle ($66.2\%$ vs. $68.4\%$); we dropped "tuning-free" in favor of "no held-out calibration" (only the SVM regularization is cross-validated). ✓
>
> **W10: baseline comparisons are unpaired; overlapping folds inflate significance.** We now present the significance markers as *descriptive indicators, not firm paired significance*, disclosing that borrowed baseline means are treated as noise-free (unpaired) and that overlapping CV folds put the effective $n$ below the nominal $50$ (§6, Protocol). ✓
>
> **Minor-3: reconcile the two MUTAG $R$ values.** Reconciled to the canonical median-over-configuration value $R\approx 6.70$ ($R^2\approx 44.9$) from the frozen submission data; the worked example and Remark 5.2 now use it consistently, and the sample-threshold example is recomputed under the honest $r_m<\hat\Delta/4$. ✓
>
> **Major-9: bibliography errors.** We audited all cited entries (50/50), correcting 7 (author names, venues, page ranges) and adding 4 missing references; the build has zero undefined citations. ✓

---

### Author Response · Authors · 2026-07-12
**Summary of revisions and response to all reviewers**

We thank all four reviewers for their careful reading. The revision makes the paper's claims strictly more honest and its empirical story sharper; point-by-point replies are posted under each reviewer's thread. The main changes:

**1. Certificate re-scoped to an honest agreement diagnostic.** Theorem 5.1(a) now certifies *agreement* of the empirical and population nearest-centroid rules away from the Voronoi boundary — not correctness (Remark 5.3). We restored the missing linear Bernstein term in the variance-aware radius (eq. 5.4, exact non-asymptotic coverage), adopted the honest firing condition $r_m<\hat\Delta/4$ (from $|\hat\Delta-\Delta|\le 2r_m$), and regenerated the firing table per fold: the honest count is **3 of 12** (NCI1, NCI109, DD), not the submitted 8. MUTAG is re-anchored on the independent $940/940$ empirical NC agreement (Clopper–Pearson $95\%$ lower bound $\ge 0.984$).

**2. Lower bound re-scoped to two-point testing/identifiability.** Theorem 3.2 now states that no test distinguishes two $\Delta$-separated, $R$-bounded diagram populations from $m\lesssim R/\Delta$ samples (error $\ge\tfrac14$), so $m=\Omega(R/\Delta)$ is necessary to identify them — exactly what Le Cam controls. We no longer assert a classification excess-risk lower bound, and we corrected the hard-family construction (affine on an interval containing the full support of $Q_\pm$).

**3. Headline is now the closed-form selector, not the oracle.** Table 10's primary column is the closed-form Mahalanobis selector's end-to-end accuracy (no held-out calibration), a mean $2.2$ pp below the in-pool oracle, which is now shown only as an upper bound. We replaced "tuning-free" with "no held-out calibration."

**4. Proved / Heuristic / Empirical evidence tags** applied per result throughout. $\nu$-coherence is carried explicitly as an empirically-audited hypothesis (Definition 2.1) with a bound-free geometric sufficient condition (Lemma 2.1) and an audit (Table 7: holds on $\ge 99.7\%$ of qualifying cross-class pairs).

**5. Section 2 restructured.** Corollary 2.1 is restricted to $d_B\ge 3R_1$ with a direct proof from the summation embedding; per-scale sharpness is separated from deductive tightness (Prop. 2.1(b)).

**6. New experiments and fixes.** A closed-form-selector ablation (Table 12: Mahalanobis vs. isotropic vs. random vs. oracle), a notation table (Appendix A), a Ballester–Rieck VR-on-shortest-path filtration demonstration (§6), disclosed statistical caveats on the significance markers (§6, Protocol), and a full bibliography audit (0 undefined citations).

A clean revised PDF replaces the original submission. We believe these changes address every raised point without weakening the method or the experiments.